# In-Trajectory Inverse Reinforcement Learning: Learn Incrementally Before an Ongoing Trajectory Terminates

**Shicheng Liu & Minghui Zhu**
School of Electrical Engineering and Computer Science
Pennsylvania State University
University Park, PA 16802, USA
`{sfl5539,muz16}@psu.edu`

## Abstract

Inverse reinforcement learning (IRL) aims to learn a reward function and a corresponding policy that best fit the demonstrated trajectories of an expert. However, current IRL works cannot learn incrementally from an ongoing trajectory because they have to wait to collect at least one complete trajectory to learn. To bridge the gap, this paper considers the problem of learning a reward function and a corresponding policy while observing the initial state-action pair of an ongoing trajectory and keeping updating the learned reward and policy when new state-action pairs of the ongoing trajectory are observed. We formulate this problem as an online bi-level optimization problem where the upper level dynamically adjusts the learned reward according to the newly observed state-action pairs with the help of a meta-regularization term, and the lower level learns the corresponding policy. We propose a novel algorithm to solve this problem and guarantee that the algorithm achieves sub-linear local regret $O(\sqrt{T} + \log T + \sqrt{T} \log T)$. If the reward function is linear, we prove that the proposed algorithm achieves sub-linear regret $O(\log T)$. Experiments are used to validate the proposed algorithm.

## 1 Introduction

Inverse reinforcement learning (IRL) aims to learn a reward function and a corresponding policy that are consistent with the demonstrated trajectories of an expert. In recent years, several IRL methods are proposed to help learn the reward and policy, including maximum margin methods [1, 2], maximum entropy methods [3, 4], maximum likelihood methods [5, 6], and Bayesian methods [7, 8].

The aforementioned IRL works learn from pre-collected demonstration sets and do not improve the learned model during deployment. Online IRL [9, 10, 11] instead can learn from sequentially arrived demonstrated trajectories and continuously improve the learned reward and policy from the newly observed complete trajectories. However, recent applications of IRL motivate the need to learn incrementally from an ongoing trajectory before it terminates. For example, inferring a moving shooter's intention from its ongoing movement in order to evacuate the hiding victims [12] before the shooter finds them. In this case, we need to quickly update the inference about the shooter's intention once a new movement of the shooter is observed, so that we can use the latest inference to plan a rescue strategy as soon as possible. We cannot wait until the shooter trajectory ends, in case the shooter has found the victims. Another example is learning a target customer's investment preference from its daily updated investment trajectory in a stock market [13] in order to recommend appropriate stocks [14, 15] before other competitors get this customer. However, current IRL works cannot learn from an ongoing trajectory because they have to wait to collect at least one complete trajectory to learn from. To bridge the gap, this paper proposes in-trajectory IRL, a new type of IRL that learns a

38th Conference on Neural Information Processing Systems (NeurIPS 2024).

reward function and a corresponding policy at the initial state-action pair of an ongoing trajectory, and keeps updating the learned reward and policy once a new state-action pair of the ongoing trajectory is observed. We summarize our contributions as follows:

**Contribution statement**. This paper proposes the first in-trajectory IRL framework, termed **me**ta-**r**egularized **in-t**rajectory inverse reinforcement learning" (MERIT-IRL), to learn a reward function and a corresponding policy from an ongoing trajectory. Our contributions are twofold.

First, we formulate this in-trajectory learning problem as an online bi-level optimization problem where the upper level continuously updates the learned reward according to the newly observed state-action pairs and the lower level computes the corresponding policy. We develop a novel online learning algorithm (MERIT-IRL) to solve this problem. The major novelty of the proposed algorithm is that we propose a novel reward update mechanism specially designed for the in-trajectory learning setting. This special reward update not only aims to explain the expert trajectory observed so far, but also aims to consider for the future. Moreover, since the data is lacking as there is only one ongoing trajectory, we introduce a meta-regularization term to embed prior knowledge and avoid overfitting.

Second, we theoretically guarantee that MERIT-IRL achieves sub-linear local regret $O(\sqrt{T} + \log T + \sqrt{T}\log T)$. If the reward function is linear, we prove that MERIT-IRL achieves sub-linear regret $O(\log T)$. The major novelty of the theoretical analysis is to address the difficulty that the input data is not identically independent distributed (i.i.d.) but temporally correlated, i.e., the data $(s_t, a_t)$ at each time $t$ is affected by the data $(s_{t-1}, a_{t-1})$ at last time.

## 2 Related works

Due to the space limit, we only discuss the related works on learning from incomplete trajectories here, and we include the discussion on more related works in Appendix A.

Papers [16, 17] use partial trajectories to update the learned reward function by comparing the expert trajectories after the expert states and the trajectories starting from those expert states rolled out by the learned policy. However, given the current expert state, they use expert trajectory suffix (i.e., future trajectory) to compare while we can only access expert trajectory prefix (i.e., previous trajectory) from an ongoing trajectory. Papers [18, 19, 20] use imitation learning to learn from incomplete demonstrations. In specific, paper [18] uses a discounted sum along the future trajectory as the weight for weighted behavior cloning and works effectively even if only portions of trajectories are observed. Papers [19, 20] extend GAIL [21] to solve for the case where the action sequences are not complete. However, these works all require a pre-collected set of demonstrations so that they are not in-trajectory learning since the trajectory in their cases is not ongoing.

## 3 Problem Formulation

In this section, we formulate the problem of in-trajectory IRL. In in-trajectory IRL, there is an expert whose decision-making is based on a Markov decision process (MDP). An MDP is a tuple $(\mathcal{S}, \mathcal{A}, \gamma, r_E, P_0, P)$ which consists of a state set $\mathcal{S}$, an action set $\mathcal{A}$, a discount factor $\gamma \in [0, 1)$, a reward function $r_E : \mathcal{S} \times \mathcal{A} \to \mathbb{R}$, and the initial state distribution $P_0(\cdot)$. The state transition probability (density) function is denoted by $P(\cdot|\cdot, \cdot)$ such that $P(s'|s, a)$ denotes the probability (density) of state transition to $s'$ from $s$ by taking action $a$. The expert is using its policy $\pi_E$ to demonstrate an ongoing trajectory $\zeta^E = S_0^E, A_0^E, S_1^E, A_1^E, \cdots$ and at each time $t$, only the state-action pair $(S_t^E, A_t^E)$ is observed. We want to learn a reward function and a corresponding policy from the ongoing trajectory and update the learned reward function and policy at each time $t$.

Many IRL algorithms [5, 6, 11, 22, 23, 24] employ a bi-level learning structure. In this structure, the upper level learns a reward function while the lower level aims to find an associated policy by solving an RL problem under the current learned reward function. Inspired by their bi-level learning structure, we formulate the in-trajectory IRL problem as an online non-convex bi-level optimization problem. In specific, we aim to learn a reward function $r_\theta$ (parameterized by $\theta$) in the upper level and a policy corresponding to $r_\theta$ in the lower level. The loss function at time $t$ is defined as:

$$L_t(\theta; (S_t^E, A_t^E)) \triangleq -\gamma^t \log \pi_\theta(A_t^E|S_t^E) + \frac{\lambda\gamma^t}{2}||\theta - \bar{\theta}||^2, \quad \pi_\theta = \arg\max_\pi J_\theta(\pi) + H(\pi). \quad (1)$$

Note that $L_t(\theta; (S_t^E, A_t^E))$ is defined using $\pi_\theta$ and $\theta$ is the parameter of the reward function $r_\theta$, here the policy $\pi_\theta$ is also parameterized by $\theta$ because it is computed by solving an RL problem (in the lower level) under the reward function $r_\theta$, and thus is indirectly parameterized by $\theta$. Maximum likelihood IRL (ML-IRL) [5] has a similar bi-level formulation with (1), however, ML-IRL only solves an offline optimization problem and its analysis does not hold for non i.i.d. input data and continuous state-action space. We discuss our distinctions from ML-IRL in Appendix A.1.

The upper-level loss function $L_t$ has two terms. The first term $-\gamma^t \log \pi_\theta(A_t^E | S_t^E)$ is the discounted negative log-likelihood of the state-action pair $(S_t^E, A_t^E)$ at time $t$ and the second term $\frac{\lambda \gamma^t}{2} ||\theta - \bar{\theta}||^2$ is the discounted meta-regularization term [25] where $\lambda$ is a hyper-parameter. The likelihood function is commonly used in IRL [5, 6] to learn a reward function. Basically, the upper-level loss function at time $t$ encourages to find a reward function $r_\theta$ that makes the observed state-action pair $(S_t^E, A_t^E)$ most likely and meanwhile, the reward parameter $\theta$ should not be too far from the prior experience, i.e., the meta-prior $\bar{\theta}$. Note that $\bar{\theta}$ is a pre-trained meta-prior that embeds the information of "relevant experience". We will introduce the training of $\bar{\theta}$ in Subsection 4.3 and Appendix C.

The lower-level problem is used to compute $\pi_\theta$ using the current reward function $r_\theta$. It proposes to find a policy $\pi_\theta$ that maximizes the entropy-regularized cumulative reward $J_\theta(\pi) + H(\pi)$. The cumulative reward of a policy $\pi$ under the reward function $r_\theta$ is $J_\theta(\pi) \triangleq E_{S,A}^\pi[\sum_{t=0}^\infty \gamma^t r_\theta(S_t, A_t)]$ where the initial state is drawn from $P_0$. The causal entropy of a policy $\pi$ is defined as $H(\pi) \triangleq E_{S,A}^\pi[-\sum_{t=0}^\infty \gamma^t \log \pi(A_t|S_t)]$.

Since the expert demonstrates $\{(S_t^E, A_t^E)\}_{t \geq 0}$ sequentially, we have a sequence of loss functions $\{L_t(\theta; (S_t^E, A_t^E))\}_{t \geq 0}$. We use this sequence of loss functions (1) to formulate an online learning problem. A typical online learning problem is to minimize the regret: $\sum_{t=0}^{T-1} L_t(\theta_t; (S_t^E, A_t^E)) - \min_\theta \sum_{t=0}^{T-1} L_t(\theta; (S_t^E, A_t^E))$. However, it is too challenging to minimize the regret in our case because the loss function $L_t$ could be non-convex. Therefore, we aim to minimize the local regret which is widely adopted in online non-convex optimization [26, 27] and online IRL [11]. The local regret quantifies the general stationarity of a sequence of loss functions under the learned parameters. In specific, given a sequence of loss functions $\{f_t(x)\}_{t \geq 0}$, the local regret [11, 26, 27] at time $t$ is defined as $||\frac{1}{t+1}\sum_{i=0}^t \nabla f_i(x_t)||^2$ which quantifies the gradient norms of the average of all the previous loss functions under the current learned parameter $x_t$. The total local regret is defined as the sum of the local regret at each time $t$, i.e., $\sum_{t=0}^{T-1} ||\frac{1}{t+1}\sum_{i=0}^t \nabla f_i(x_t)||^2$. In our case, we replace $\{f_t\}_{t \geq 0}$ with the loss function $\{L_t\}_{t \geq 0}$ defined in (1) and thus formulate the local regret (2)-(3) which has a bi-level formulation. We aim to minimize the following local regret:

$$E_{\{(S_t^E, A_t^E) \sim \mathbb{P}_t^{\pi_E}(\cdot, \cdot)\}_{t \geq 0}} \left[ \sum_{t=0}^{T-1} ||\frac{1}{t+1}\sum_{i=0}^t \nabla L_i(\theta_t; (S_i^E, A_i^E))||^2 \right], \tag{2}$$

$$\text{s.t.} \quad \pi_{\theta_t} = \arg\max_\pi J_{\theta_t}(\pi) + H(\pi), \tag{3}$$

where $(S_t^E, A_t^E) \sim \mathbb{P}_t^{\pi_E}(\cdot, \cdot)$ means that $(S_t^E, A_t^E)$ is drawn from the state-action distribution $\mathbb{P}_t^{\pi_E}(\cdot, \cdot)$, and $\mathbb{P}_t^\pi(\cdot, \cdot)$ is the state-action distribution induced by $\pi$ at time $t$ in the MDP.

**Difficulties of solving problem** (2)-(3). We want to design a fast algorithm to solve problem (2)-(3) since we need to finish the update of $\theta$ and $\pi$ before the next state-action pair is observed and the time between two consecutive state-action pairs can be short. However, designing and analyzing such a fast algorithm is difficult due to the following challenges:

(i) First and foremost, current state-of-the-arts [26, 27] on online non-convex optimization use follow-the-leader-based algorithms which solve $\min_\theta \sum_{i=0}^t L_i(\theta; (S_i^E, A_i^E))$ to near stationarity at each time $t$. This is time-consuming because they require multiple gradient descent updates of $\theta$. One way to alleviate this problem is to use online gradient descent (OGD) which only updates $\theta$ by one gradient descent step at each time $t$. However, since OGD does not solve the problem to near stationarity at any time $t$, it is extremely difficult to quantify the overall stationarity after $T$ iterations. While OGD has been well studied in online convex optimization, it is rarely studied in online non-convex optimization. The recent work [11] uses OGD to quantify the local regret, however, its analysis can only hold when the input data is i.i.d. In contrast, the input data in our problem is not i.i.d. In specific, the input data at time $t$ (i.e., $(S_t^E, A_t^E)$) is actually affected by the input data at last step (i.e.,

$(S_{t-1}^E, A_{t-1}^E)$). This correlation between any two consecutive input data makes it difficult to analyze the growth rate of the local regret.

(ii) Second, it is time-consuming if we fully solve the lower-level problem (3) to get $\pi_\theta$ because this requires multiple policy updates to solve an RL problem. Therefore, we use a "single-loop" method which only requires one-step policy update for a given $r_\theta$. However, since the policy is only updated once, the updated policy can be far from $\pi_\theta$ and thus making the analysis difficult. Single-loop methods are widely adopted to solve hierarchical problems, including bi-level optimization [28, 29], game theory [30, 31], min-max problems [32, 33], etc. Recently, single-loop methods are applied to IRL [5], however, the paper [5] only solves an offline optimization problem and its analysis does not hold for non i.i.d. input data and continuous state-action space. We include a section in Appendix A.1 to discuss our distinctions from [5].

## 4 Algorithm and Theoretical Analysis

This section has three parts. The first part presents a novel online learning algorithm that solves the problem (2)-(3) and tackles the aforementioned two difficulties. The second part proves that Algorithm 1 achieves sub-linear local regret. If the reward function is linear, we prove that Algorithm 1 achieves sub-linear regret. The third part introduces a meta-learning method to get the meta-prior $\bar{\theta}$.

### 4.1 The proposed algorithm

In practice, the expert will demonstrate a specific trajectory $s_0^E, a_0^E, s_1^E, a_1^E, \cdots$. For distinction, we use the capital letters (e.g., $S$) to represent random variables and the lower-case letters (e.g., $s$) to represent specific values. To design a fast algorithm, we propose an online-gradient-descent-based single-loop algorithm. In specific, at each time $t$, the algorithm updates both policy $\pi$ and reward parameter $\theta$ only once. The policy update is to solve the lower-level problem (3) and the reward update is to solve the upper-level problem (2). In the following, we elaborate the procedure of policy update and reward update.

---

**Algorithm 1** Meta-regularized In-trajectory Inverse Reinforcement Learning (MERIT-IRL)

---

**Input**: Initialized policy $\pi_0$, the streaming input data $\{(s_t^E, a_t^E)\}_{t \geq 0}$
**Output**: Learned reward parameter $\theta_T$ and policy $\pi_T$
 1: Compute $\bar{\theta}$ using the meta-regularization in Section 4.3 and Appendix C, and set $\theta_0 = \bar{\theta}$
 2: **for** $t = 0, 1, \cdots, T-1$ **do**
 3:    Compute the soft Q-function $Q_{\theta_t, \pi_t}^{\text{soft}}$ (defined in Appendix B.1) under the current reward function $r_{\theta_t}$ and policy $\pi_t$
 4:    Update $\pi_{t+1}(a|s) \propto \exp(Q_{\theta_t, \pi_t}^{\text{soft}}(s, a))$ for any $(s, a) \in \mathcal{S} \times \mathcal{A}$
 5:    Roll out policy $\pi_{t+1}$ twice: one starting from $s_0^E$ to get $s_0^E, a_0', s_1', a_1', \cdots$, and the other starting from $(s_t^E, a_t^E)$ to get $s_{t+1}'', a_{t+1}'', \cdots$
 6:    Compute $g_t = \sum_{i=0}^\infty \gamma^i \nabla_\theta r_{\theta_t}(s_i', a_i') - \sum_{i=0}^\infty \gamma^i \nabla_\theta r_{\theta_t}(s_i'', a_i'') + \frac{\lambda(1-\gamma^{t+1})}{1-\gamma}(\theta_t - \bar{\theta})$ where $s_0' = s_0^E$ and $(s_i'', a_i'') = (s_i^E, a_i^E)$ for $0 \leq i \leq t$
 7:    Update $\theta_{t+1} = \theta_t - \alpha_t g_t$
 8: **end for**

---

**Policy update** (lines 3-4 of Algorithm 1). At each time $t$, we only partially solve the lower-level problem (3) via one-step soft policy iteration [5, 34]. In specific, the soft policy iteration contains two steps: policy evaluation and policy improvement. Policy evaluation aims to compute the soft Q-function $Q_{\theta_t, \pi_t}^{\text{soft}}$ (see the expression in Appendix B.1) under the current learned reward function $r_{\theta_t}$ and learned policy $\pi_t$. Policy improvement aims to update policy according to $\pi_{t+1}(s, a) \propto \exp(Q_{\theta_t, \pi_t}^{\text{soft}}(s, a))$ for any $(s, a) \in \mathcal{S} \times \mathcal{A}$. In practical implementations, $\pi_{t+1}$ can be obtained by one-step policy update in soft Q-learning [34] or one-step actor update in soft actor-critic [35].

**Reward update** (lines 5-7 of Algorithm 1). At each time $t$, the algorithm observes $(s_t^E, a_t^E)$ and aims to leverage all the previously observed data to update the reward parameter. In specific, as $L_t(\theta; (s_t^E, a_t^E)) = -\gamma^t \log \pi_\theta(a_t^E | s_t^E) + \frac{\lambda \gamma^t}{2} ||\theta - \bar{\theta}||^2$ is the meta-regularized negative log-likelihood

of $(s_t^E, a_t^E)$, the algorithm can formulate $\sum_{i=0}^{t} L_i(\theta; (s_i^E, a_i^E))$ at time $t$ using all the previously collected data (i.e., $\{(s_i^E, a_i^E)\}_{i=0,\cdots,t}$). To update the reward parameter, the algorithm partially minimizes $\sum_{i=0}^{t} L_i(\theta; (s_i^E, a_i^E))$ via one-step gradient descent.

**Lemma 1.** *The gradient of $\sum_{i=0}^{t} L_i(\theta; (s_i^E, a_i^E))$ can be calculated as follows:*

$$
\nabla \sum_{i=0}^{t} L_i(\theta; (s_i^E, a_i^E)) = E_{S,A}^{\pi_\theta} \left[ \sum_{i=0}^{\infty} \gamma^i \nabla_\theta r_\theta(S_i, A_i) \Big| S_0 = s_0^E \right] + \frac{\lambda(1 - \gamma^{t+1})}{1 - \gamma}(\theta - \bar{\theta})
$$

$$
- \sum_{i=0}^{t} \gamma^i \nabla_\theta r_\theta(s_i^E, a_i^E) - E_{S,A}^{\pi_\theta} \left[ \sum_{i=t+1}^{\infty} \gamma^i \nabla_\theta r_\theta(S_i, A_i) \Big| S_t = s_t^E, A_t = a_t^E \right]. \tag{4}
$$

Note that the gradient (4) holds for continuous state-action space. Since the gradient (4) has expectation terms under the policy $\pi_\theta$, we can only approximate it. In specific, we roll out the policy $\pi_{t+1}$ twice: one starting from $s_0^E$ to get a trajectory $\{(s_i', a_i')\}_{i \geq 0}$ where $s_0' = s_0^E$, and the other one starting from $(s_t^E, a_t^E)$ to get a trajectory $\{(s_i'', a_i'')\}_{i \geq 0}$ where $(s_i'', a_i'') = (s_i^E, a_i^E)$ for $0 \leq i \leq t$. Then we use the empirical estimate $g_t = \sum_{i=0}^{\infty} \gamma^i \nabla_\theta r_{\theta_t}(s_i', a_i') - \sum_{i=0}^{\infty} \gamma^i \nabla_\theta r_{\theta_t}(s_i'', a_i'') + \frac{\lambda(1-\gamma^{t+1})}{1-\gamma}(\theta_t - \bar{\theta})$ to approximate $\nabla \sum_{i=0}^{t} L_i(\theta_t; (s_i^E, a_i^E))$. With the gradient approximation $g_t$, we utilize stochastic online gradient descent $\theta_{t+1} = \theta_t - \alpha_t g_t$ to update the reward parameter.

**Discussion on our special design of the reward update**. The right subfigure in Figure 1 visualizes our reward update (modulo the meta-regularization term). The green trajectory (i.e., $\{(s_t^E, a_t^E)\}_{t \geq 0}$) is the expert trajectory, and the red trajectories (i.e., $\{(s_t', a_t')\}_{t \geq 0}$ and $\{(s_i'', a_i'')\}_{i > t}$) are the trajectories generated by the learned policy. Given the expert trajectory prefix (i.e., the incomplete trajectory $\{(s_i^E, a_i^E)\}_{i=0}^{t}$ observed so far), our method completes the expert trajectory by rolling out the learned policy starting from $(s_t^E, a_t^E)$ and filling the trajectory suffix $\{(s_i'', a_i'')\}_{i > t}$. The combined complete trajectory includes the expert trajectory prefix $\{(s_i^E, a_i^E)\}_{i=0}^{t}$ and the learner-filled trajectory suffix $\{(s_i'', a_i'')\}_{i \geq t}$. We update the reward function by comparing this combined trajectory to a complete trajectory $\{(s_t', a_t')\}_{t \geq 0}$ generated by the learned policy starting from the expert's initial state $s_0^E$.

A more straightforward way for the reward update is to directly compare the trajectory prefixes (visualized in the middle of Figure 1) at each time $t$. However, this naive method can be problematic. We explain the issue of this naive method and the advantage of our method in the following context.

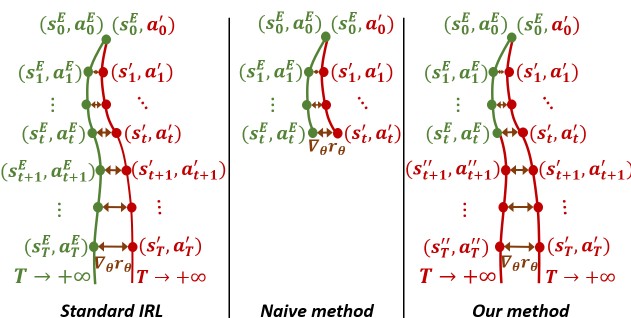

*Standard IRL*    *Naive method*    *Our method*

Figure 1: Standard IRL (left), the naive method for intrajectory learning (middle), and our method (right).

Figure 1 visualizes the reward update for standard IRL (left), the naive method (i.e., directly run standard IRL algorithms on the expert trajectory prefix) (middle), and our method (right). The standard IRL (left) updates the reward function by comparing the complete expert trajectory and the complete trajectory generated by the learned policy. This case is ideal, however, it is infeasible when the trajectory is ongoing and we can only observe an incomplete expert trajectory $\{(s_i^E, a_i^E)\}_{i=0}^{t}$ at each time $t$. The naive method (middle) updates the reward function by simply comparing the expert trajectory prefix $\{(s_i^E, a_i^E)\}_{i=0}^{t}$ and the incomplete trajectory $\{(s_i', a_i')\}_{i=0}^{t}$ with the same length generated by the learned policy starting from the expert's initial state $s_0^E$. This kind of reward update is myopic as it does not consider for the future. Since it runs standard IRL on the trajectory prefix observed so far, it will always regard the current state-action pair as the terminal state-action pair and thus has no ability to consider the state-action pairs in the future. In contrast, our special design gives the algorithm the ability to consider for the future. Instead of only comparing incomplete trajectory prefixes, our method compares complete trajectories just as the standard IRL. In specific, we compare the combined complete trajectory $\{\{(s_i^E, a_i^E)\}_{i=0}^{t}, \{(s_i'', a_i'')\}_{i>t}\}$ and the complete trajectory $\{(s_i', a_i')\}_{i \geq 0}$ solely generated by the learned policy. The comparison between the learner prefix $\{s_i', a_i'\}_{i=0}^{t}$ and expert

prefix $\{s_i^E, a_i^E\}_{i=0}^t$ encourages the learned reward function to explain the expert's demonstrated behaviors so far, and the comparison between the suffixes ($\{s_i', a_i'\}_{i>t}$ and $\{s_i'', a_i''\}_{i>t}$) encourages that we are learning a reward function that is useful for predicting the future. Note that as $t$ increases, the expert trajectory prefix weights more and more in the combined complete trajectory, and eventually we will recover the standard IRL reward update when $t$ goes to infinity. In Theorems 1 and 2, we theoretically guarantee that the proposed reward update can achieve sub-linear (local) regret. This shows the perfect consistency between the intuition and theory.

## 4.2 Theoretical analysis

To quantify the local regret of Algorithm 1, we have two challenges: (i) Since we only update $\pi$ by one step at each time $t$, we have $\pi_{t+1}$ instead of the optimal solution $\pi_{\theta_t}$ of the lower-level problem (3). The policy $\pi_{t+1}$ can be far away from $\pi_{\theta_t}$ and thus the empirical gradient estimate $g_t$ can be a bad approximation of the gradient (4). (ii) Since we only update $\theta$ once at each time $t$ instead of finding a near-stationary point $\theta'$ such that $||\sum_{i=0}^t L_i(\theta'; (s_i^E, a_i^E))|| \leq \epsilon$ as in [26, 27], the gradient norm $||\sum_{i=0}^t \nabla L_i(\theta_t; (s_i^E, a_i^E))||$ is not stabilized under the threshold $\epsilon$ at every time $t$. Therefore the local regret (i.e., $\sum_{t=0}^{T-1} ||\frac{1}{t+1} \sum_{i=0}^t \nabla L_i(\theta_t; (s_i^E, a_i^E))||^2$) is hard to quantify and may not be sub-linear in $T$. What's worse, the input data is not i.i.d. but correlated, i.e., the input data $(s_t^E, a_t^E)$ at time $t$ is affected by the input data $(s_{t-1}^E, a_{t-1}^E)$ at last step. This correlation makes it even more difficult to quantify the local regret.

To solve the first challenge, we adopt the idea of two-timescale stochastic approximation [28] where the lower level updates in a faster timescale and the upper level updates in a slower timescale. The policy update is faster because it converges linearly under a fixed reward function [36] while the reward update is slower given that we choose $\alpha_t \propto (t+1)^{-1/2}$. Intuitively, since the policy update is faster than the reward update, the reward parameter is "relatively fixed" compared to the policy. It is expected that $\pi_{t+1}$ shall stay close to $\pi_{\theta_t}$ and at last converges to $\pi_{\theta_t}$ when $t$ increases.

To solve the second challenge, we divide our analysis into two steps: (i) We quantify the difference of the gradient norms between the current correlated state-action distribution $\mathbb{P}_t^{\pi_E}(\cdot, \cdot)$ and a stationary state-action distribution for any loss function $L_i$, $i \geq 0$. Note that $\mathbb{P}_{t+1}^{\pi_E}(\cdot, \cdot)$ is affected by $\mathbb{P}_t^{\pi_E}(\cdot, \cdot)$. (ii) We quantify the local regret under the stationary distribution. The benefit of doing so is that the input data is i.i.d. under the stationary distribution, and thus we can cast the online gradient descent method as a stochastic gradient descent method and quantify its local regret. Finally, we can quantify the local regret under the current correlated distribution $\mathbb{P}_t^{\pi_E}(\cdot, \cdot)$ by combining (i) and (ii).

We start our analysis with the definitions of stationary state distribution and stationary state-action distribution. For a given policy $\pi$, the corresponding stationary state distribution is $\mu^\pi(s) \triangleq (1 - \gamma) \sum_{t=0}^\infty \gamma^t \mathbb{P}_t^\pi(s)$ and the stationary state-action distribution is $\mu^\pi(s, a) \triangleq (1 - \gamma) \sum_{t=0}^\infty \gamma^t \mathbb{P}_t^\pi(s, a)$.

**Assumption 1.** *The parameterized reward function $r_\theta$ satisfies $|r_{\theta_1}(s, a) - r_{\theta_2}(s, a)| \leq \bar{C}_r ||\theta_1 - \theta_2||$ and $||\nabla_\theta r_{\theta_1}(s, a) - \nabla_\theta r_{\theta_2}(s, a)|| \leq \tilde{C}_r ||\theta_1 - \theta_2||$ for any $(\theta_1, \theta_2)$ and any $(s, a) \in \mathcal{S} \times \mathcal{A}$ where $\bar{C}_r$ and $\tilde{C}_r$ are positive constants.*

**Assumption 2** (Ergodicity). *There exist constants $C_M > 0$ and $\rho \in (0, 1)$ such that for any policy $\pi$ and any $t \geq 0$, the following holds for the Markov chain induced by the policy $\pi$ and the state transition function $P$: $\sup_{S_0 \sim P_0} d_{TV}(\mathbb{P}_t^\pi(\cdot), \mu^\pi(\cdot)) \leq C_M \rho^t$ where $d_{TV}(\mathbb{P}_1(\cdot), \mathbb{P}_2(\cdot)) \triangleq \frac{1}{2} \int_{s \in \mathcal{S}} |\mathbb{P}_1(s) - \mathbb{P}_2(s)| ds$ is the total variation distance between the two state distributions $\mathbb{P}_1$ and $\mathbb{P}_2$, $S_0$ is the initial state, and $\mathbb{P}_t^\pi(\cdot)$ is the state distribution induced by the policy $\pi$ at time $t$.*

Assumptions 1-2 are common in RL [37, 38, 39, 40]. Assumption 2 holds for any time-homogeneous Markov chain with finite state space or any uniformly ergodic Markov chain with general state space.

**Proposition 1.** *Suppose Assumptions 1-2 hold and $\alpha_t \in (0, \frac{1-\gamma}{\lambda})$, we have the following relation for any $i \geq 0$ and any $\theta_t, t \geq 0$:*

$$\left| E_{(S_i^E, A_i^E) \sim \mathbb{P}_i^{\pi_E}(\cdot, \cdot)} \left[ ||\nabla L_i(\theta_t; (S_i^E, A_i^E))||^2 \right] - E_{(S_i^E, A_i^E) \sim \mu^{\pi_E}(\cdot, \cdot)} \left[ ||\nabla L_i(\theta_t; (S_i^E, A_i^E))||^2 \right] \right|,$$

$$\leq 8 C_M \bar{C}_r^2 \left( \frac{2 - \gamma}{1 - \gamma} \right)^2 \rho^i \gamma^{2i},$$

*where $(S_i^E, A_i^E) \sim \mathbb{P}_i^{\pi_E}(\cdot, \cdot)$ means that $(S_i^E, A_i^E)$ is drawn from the correlated distribution $\mathbb{P}_i^{\pi_E}(\cdot, \cdot)$ and $(S_i^E, A_i^E) \sim \mu^{\pi_E}(\cdot, \cdot)$ means that $(S_i^E, A_i^E)$ is drawn from the stationary distribution $\mu^{\pi_E}(\cdot, \cdot)$.*

Proposition 1 quantifies the gap of gradient norms between the current correlated distribution $\mathbb{P}_i^{\pi_E}$ and the stationary distribution $\mu^{\pi_E}$. We next quantify the local regret under the stationary distribution $\mu^{\pi_E}$ with the following lemma:

**Lemma 2.** *Suppose Assumptions 1-2 hold and choose* $\alpha_t = \frac{(1-\gamma)(t+1)^{-1/2}}{\lambda}$, *it holds that:*

$$E_{\{(S_t^E, A_t^E) \sim \mu^{\pi_E}(\cdot, \cdot)\}_{t \geq 0}} \left[ \sum_{t=0}^{T-1} || \frac{1}{t+1} \sum_{i=0}^{t} \nabla L_i(\theta_t; (S_i^E, A_i^E)) ||^2 \right]$$
$$\leq D_1(\log T + 1) + D_2\sqrt{T} + D_3\sqrt{T}(\log T + 1),$$

*where $D_1$, $D_2$, and $D_3$ are positive constants whose expressions can be found in Appendix B.4.*

Lemma 2 quantifies the local regret under the stationary distribution $\mu^{\pi_E}$. With Proposition 1 and Lemma 2, we can quantify the local regret under the current correlated distribution.

**Theorem 1.** *Suppose Assumptions 1-2 hold and choose* $\alpha_t = \frac{(1-\gamma)(t+1)^{-1/2}}{\lambda}$, *we have that:*

$$E_{\{(S_t^E, A_t^E) \sim \mathbb{P}_t^{\pi_E}(\cdot, \cdot)\}_{t \geq 0}} \left[ \sum_{t=0}^{T-1} || \frac{1}{t+1} \sum_{i=0}^{t} \nabla L_i(\theta_t; (S_i^E, A_i^E)) ||^2 \right]$$
$$\leq \left( D_1 + \frac{8C_M \bar{C}_r^2 (2-\gamma)^2}{(1-\rho\gamma^2)(1-\gamma)^2} \right)(\log T + 1) + D_2\sqrt{T} + D_3\sqrt{T}(\log T + 1).$$

Theorem 1 is based on Proposition 1 and Lemma 2. It shows that Algorithm 1 achieves sub-linear local regret. Moreover, if the reward function is linear, Algorithm 1 achieves sub-linear regret:

**Theorem 2.** *Suppose the expert reward function $r_E$ and the parameterized reward $r_\theta$ are linear, and Assumptions 1-2 hold. Choose* $\alpha_t = \frac{1-\gamma}{\lambda(t+1)(1-\gamma^{t+1})}$ *we have that:*

$$E_{\{(S_t^E, A_t^E) \sim \mathbb{P}_t^{\pi_E}(\cdot, \cdot)\}_{t \geq 0}} \left[ \sum_{t=0}^{T-1} L_t(\theta_t; (S_t^E, A_t^E)) \right]$$
$$- \min_\theta E_{\{(S_t^E, A_t^E) \sim \mathbb{P}_t^{\pi_E}(\cdot, \cdot)\}_{t \geq 0}} \left[ \sum_{t=0}^{T-1} L_t(\theta; (S_t^E, A_t^E)) \right] \leq D_4 + D_5(\log T + 1),$$

*where $D_4$ and $D_5$ are positive constants whose expressions are in Appendix B.6.*

### 4.3 Meta-Regularization

Since there is only one training trajectory and this trajectory is not complete during the learning process, we need to add a regularization term to avoid overfitting. Inspired by humans' using relevant experience to help do inference, we introduce the meta-regularization $\frac{\lambda}{2}||\theta - \bar{\theta}||^2$ where $\lambda$ is the hyper-parameter and the meta-prior $\bar{\theta}$ is learned from "relevant experience". In specific, we introduce a set of relevant tasks $\{\mathcal{T}_j\}_{j \sim P_\mathcal{T}}$ where each task $\mathcal{T}_j$ is an IRL problem and $P_\mathcal{T}$ is the implicit task distribution. The tasks $\{\mathcal{T}_j\}_{j \sim P_\mathcal{T}}$ are relevant in the sense that they share the components $(\mathcal{S}, \mathcal{A}, \gamma, P_0, P)$ of the MDP with our in-trajectory learning problem. However, the expert's reward functions of different tasks are different and are drawn from an unknown reward function distribution. For example, in the stock market case mentioned in the introduction, the experts of different tasks invest in the same stock market but may have different preferences. As standard in meta-learning [41, 42, 43], we assume that the expert's reward function $r_E$ of our in-trajectory learning problem is also drawn from the same unknown reward function distribution. Note that the reward functions of the relevant tasks $\{\mathcal{T}_j\}_{j \sim P_\mathcal{T}}$ are different from $r_E$ even if they are drawn from the same unknown reward function distribution.

For each task $\mathcal{T}_j$, there is a batch of trajectories and we divide this batch into two sets, i.e., $\mathcal{D}_j^{\text{tr}}$ and $\mathcal{D}_j^{\text{eval}}$. The training set $\mathcal{D}_j^{\text{tr}}$ only has one trajectory, just as our in-trajectory learning problem, and the evaluation set $\mathcal{D}_j^{\text{eval}}$ has abundant trajectories. Define the loss function on a certain data set $\mathcal{D} \triangleq \{\zeta^v\}_{v=1}^m$ as $L(\theta, \mathcal{D}) \triangleq -\sum_{v=1}^m \sum_{t=0}^\infty \gamma^t \log \pi_\theta(a_t^v | s_t^v)$. The goal of each task $\mathcal{T}_j$ is to learn a task-specific adaptation $\phi_j$ using the training set $\mathcal{D}_j^{\text{tr}}$, such that $\phi_j$ can minimize the test

loss $L(\phi_j, \mathcal{D}_j^{\text{eval}})$ on the evaluation set $\mathcal{D}_j^{\text{eval}}$. The goal of meta-regularization is to find a meta-prior $\bar{\theta}$, from which such task-specific adaptations $\phi_j$ can be adapted to all tasks $\{\mathcal{T}_j\}_{j\sim P_\mathcal{T}}$. In specific, meta-regularization [25] proposes a bi-level optimization problem (5). The lower-level problem uses only one trajectory $\mathcal{D}_j^{\text{tr}}$ to find the task-specific adaptation $\phi_j$ such that the meta-regularized loss function $L(\phi, \mathcal{D}_j^{\text{tr}}) + \frac{\lambda}{2(1-\gamma)}||\phi - \bar{\theta}||^2$ is minimized. The upper-level problem is to find a meta-prior $\bar{\theta}$ such that the corresponding task-specific adaptations $\{\phi_j\}_{j\sim P_\mathcal{T}}$ can minimize the expected loss function $L(\phi_j, \mathcal{D}_j^{\text{eval}})$ over the evaluation sets of all tasks $\{\mathcal{T}_i\}_{j\sim P_\mathcal{T}}$.

$$\min_{\bar{\theta}} E_{j\sim P_\mathcal{T}}\left[L(\phi_j, \mathcal{D}_j^{\text{eval}})\right], \quad \text{s.t. } \phi_j = \arg\min_{\phi} L(\phi, \mathcal{D}_j^{\text{tr}}) + \frac{\lambda}{2(1-\gamma)}||\phi - \bar{\theta}||^2. \tag{5}$$

The lower-level loss function in (5) is the offline version of our in-trajectory loss function (1) (i.e., $L(\phi, \mathcal{D}_j^{\text{tr}}) + \frac{\lambda}{2(1-\gamma)}||\phi - \bar{\theta}||^2 = \sum_{t=0}^{\infty} L_t(\phi; (s_t^{\text{tr}}, a_t^{\text{tr}}))$) where $(s_t^{\text{tr}}, a_t^{\text{tr}}) \in \mathcal{D}_j^{\text{tr}}$. Our in-trajectory learning problem can also be regarded as to find a task-specific adaptation. Note that the in-trajectory learning problem is online while the lower-level problem in (5) is offline because the in-trajectory problem is ongoing where we keep observing new state-action pairs. In contrast, the lower-level problem in (5) is based on "experience" that has already happened. Due to the space limit, we include the algorithm and theoretical guarantees of solving the problem (5) in Appendix C.

## 5 Experiments

We present three experiments to show the effectiveness of MERIT-IRL. We use four baselines for comparisons. (i) **IT-IRL**: this method is MERIT-IRL without meta-regularization. (ii) **Naive MERIT-IRL**: this method has the meta-regularization term but uses the naive way (in the middle of Figure 1) to update reward. (iii) **Naive IT-IRL**: this method uses the naive way to update the learned reward and does not have the meta-regularization term. (iv) **Hindsight**: this method is meta-regularized ML-IRL [5] which can access the complete expert trajectory and uses the standard IRL (visualized in the left of Figure 1) with meta-regularization to update the learned reward. The experiment details are in Appendix D.

### 5.1 MuJoCo experiment

In this subsection, we consider the target velocity problem for three MuJoCo robots: HalfCheetah, Walker, and Hopper. The target velocity problem is widely used in meta-RL [44] and meta-IRL [45]. In specific, the robots aim to maintain a target velocity in each task and the target velocity of different tasks is different. To test the performance of MERIT-IRL, we use 10 test tasks whose target velocity is randomly between 1.5 and 2.0. In the test tasks, there is only one expert trajectory and the state-action pairs of this trajectory are sequentially revealed (to MERIT-IRL, IT-IRL, Naive MERIT-IRL, and Naive IT-IRL) in an online fashion. The baseline Hindsight uses the complete expert trajectory to learn a reward function. The ground truth reward is designed as $-|v - v_{\text{target}}|$ (as in [44]) where $v$ is the current robot velocity and $v_{\text{target}}$ is the target velocity. To learn the meta-prior $\bar{\theta}$, we use 50 relevant tasks whose target velocity is randomly between 0 and 3.

Figures 2a-2c show the in-trajectory learning performance where the $x$-axis is the time step $t$ of the expert trajectory and the $y$-axis is the cumulative reward of the learned policy $\pi_t$ when only the first $t$ steps of the expert trajectory are observed. The $x$-limit is $1,000$ because the trajectory length in MuJoCo is $1,000$. Note that the baseline "Hindsight" is not in-trajectory learning since it learns from a complete expert trajectory. For comparison, we use two horizontal lines (close to each other) to show the performance of Hindsight and the expert in the figures. Figure 2a shows that MERIT-IRL achieves similar performance with the expert when only $40\%$ of the complete expert trajectory ($t = 400$) is observed while IT-IRL can only achieve performance close to the expert after observing more than $90\%$ of the complete expert trajectory ($t = 900$). This shows the effectiveness of the meta-regularization. Naive MERIT-IRL and Naive IT-IRL fail to imitate the expert even if the complete expert trajectory is observed ($t = 1,000$). This shows the effectiveness of our special design of the reward update. The discussions on Figures 2b and 2c are in Appendix D.2.

Table 1 shows the results after observing the complete expert trajectory. MERIT-IRL performs much better than IT-IRL, Naive MERIT-IRL, and Naive IT-IRL. MERIT-IRL achieves similar performance

Table 1: Experiment results. The mean and standard deviation are calculated from 10 test tasks.

| | MERIT-IRL | IT-IRL | Naive MERIT-IRL | Naive IT-IRL | Hindsight | Expert |
|---|---|---|---|---|---|---|
| HalfCheetah | $-214.63 \pm 53.96$ | $-386.78 \pm 152.65$ | $-548.22 \pm 40.51$ | $-765.27 \pm 104.41$ | $-208.74 \pm 37.23$ | $-181.51 \pm 28.35$ |
| Walker | $-654.77 \pm 102.59$ | $-891.79 \pm 156.90$ | $-962.42 \pm 111.60$ | $-1349.25 \pm 158.88$ | $-648.17 \pm 157.92$ | $-634.17 \pm 120.57$ |
| Hopper | $-476.72 \pm 32.09$ | $-669.88 \pm 53.63$ | $-691.03 \pm 93.35$ | $-1112.06 \pm 74.33$ | $-455.70 \pm 74.93$ | $-421.74 \pm 84.30$ |
| Stock market | $386.70 \pm 62.95$ | $256.81 \pm 68.61$ | $192.49 \pm 75.34$ | $72.33 \pm 16.73$ | $390.30 \pm 77.37$ | $403.15 \pm 61.94$ |

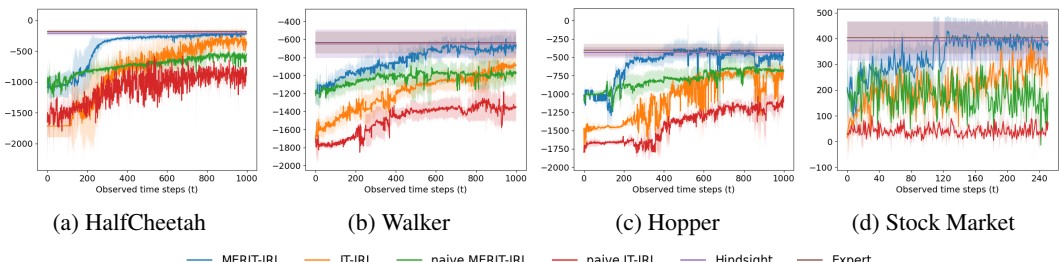

(a) HalfCheetah    (b) Walker    (c) Hopper    (d) Stock Market

— MERIT-IRL  — IT-IRL  — naive MERIT-IRL  — naive IT-IRL  — Hindsight  — Expert

Figure 2: In-trajectory learning performance.

with Hindsight and expert. Note that it is not expected that MERIT-IRL outperforms Hindsight since Hindsight uses the complete expert trajectory to learn.

### 5.2 Stock market experiment

RL to train a stock trading agent has been widely studied in AI for finance [46, 47, 48]. In this experiment, we use IRL to learn the reward function (i.e., investing preference) of the target investor in a stock market scenario. In specific, we use the real-world data of 30 constituent stocks in Dow Jones Industrial Average from 2021-01-01 to 2022-01-01. We use a benchmark called "FinRL" [48] to configure the real-world stock data into an MDP environment. The target investor (i.e., expert) has an initial asset of $\$1,000$ and trades stocks on every stock market opening day. The stock market opens 252 days between 2021-01-01 and 2022-01-01, and thus the trajectory length is 252. The reward function of the target investor is defined as $p_1 - p_2$ where $p_1$ is the investor's profit which is the money earned from trading stocks subtracting the transaction cost, and $p_2$ models the investor's preference of whether willing to take risks. In specific, $p_2$ is positive if the investor buys stocks whose turbulence indices are larger than a certain turbulence threshold, and zero otherwise. The value of $p_2$ depends on the type and amount of the trading stocks. The turbulence thresholds of different investors are different. The turbulence index measures the price fluctuation of a stock. If the turbulence index is high, the corresponding stock has a high fluctuating price and thus is risky to buy [48]. Therefore, an investor unwilling to take risks has a relatively low turbulence threshold. We include experiment details in Appendix D.3. To test performance, we use 10 test tasks whose turbulence thresholds are randomly between 45 and 50. To learn $\bar{\theta}$, we use 50 relevant tasks whose turbulence thresholds are randomly between 30 and 60.

Figure 2d shows that MERIT-IRL achieves similar cumulative reward with the expert at $t = 140$ which is less than $60\%$ of the whole trajectory, while the three in-trajectory baselines fail to imitate the expert before the ongoing trajectory terminates. The last row in Table 1 shows that MERIT-IRL achieves similar performance with Hindsight and the expert. More discussions on the results are in Appendix D.3.

### 5.3 Learning from a shooter's ongoing trajectory

This part presents the experiment of learning from an ongoing shooter trajectory. Following [12], we model the shooter's movement as a navigation problem. We build a simulator in Gazebo (Figure 3a) where the shooter moves from the door (lower left corner) to the red target (upper right corner). The learner observes the ongoing trajectory of the shooter and keeps updating the learned reward and policy. In our case, the complete shooter trajectory has the length of 140. Figures 3b-3g show our in-trajectory learning performance where the heat maps visualize the learned reward. We normalize the learned reward to $[0, 1]$. We can observe that as the ongoing trajectory is expanding, the learned reward function becomes more and more precise to locate the goal area. When $t = 40$, we cannot tell

the goal area from the heat map (Figure 3b). However, as the time $t$ grows, we can almost locate the goal area when $t = 60$ (Figure 3c) and precisely locate the goal area when $t = 80$ (Figure 3d).

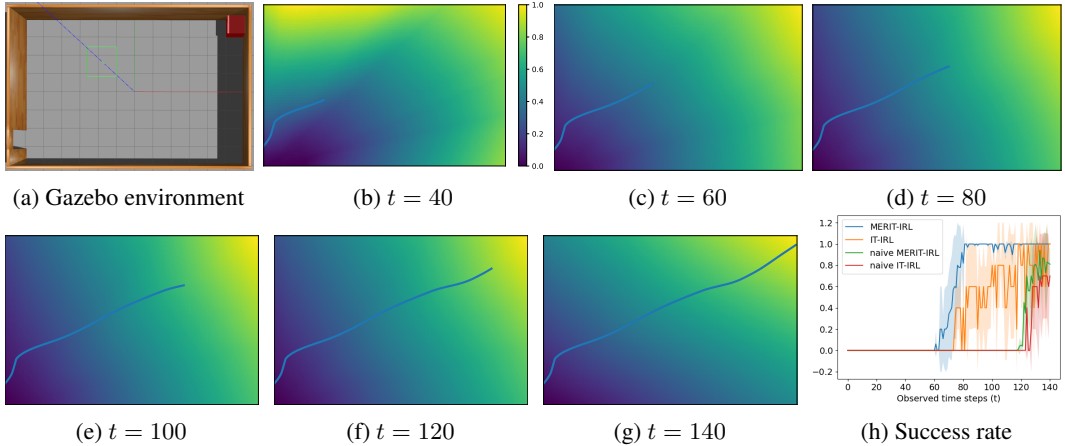

(a) Gazebo environment      (b) $t = 40$      (c) $t = 60$      (d) $t = 80$

(e) $t = 100$      (f) $t = 120$      (g) $t = 140$      (h) Success rate

Figure 3: Learning performance on the active shooting scenario.

Figure 3h shows the policy learning performance. Since there is no ground truth reward in this problem, we use "success rate" to quantify the performance of the learned policy. The success rate is the rate that the learned policy successfully reaches the goal. From 3h, we can see that MERIT-IRL outperforms the other baselines and can achieve $100\%$ success rate when $t = 80$ (i.e., only observing $57\%$ of the complete trajectory). Note that we do not include Hindsight and Expert in Figure 3 since they both achieve $100\%$ success rate.

## 6 Conclusion

This paper proposes MERIT-IRL, the first in-trajectory inverse reinforcement learning theoretical framework that learns a reward function while observing an initial portion of a trajectory and keeps updating the learned reward function when extended portions (i.e., new state-action pairs) of the trajectory are observed. Experiments show that MERIT-IRL can imitate the expert from the ongoing expert trajectory before it terminates.

## 7 Acknowledgements

This work is partially supported by the National Science Foundation through grants ECCS 1846706 and ECCS 2140175. We would like to thank the reviewers for their insightful and constructive suggestions.

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

This appendix consists of four parts: related works, proof, meta-regularization algorithm and convergence guarantee, and Experiment details.

# A  Related works

**Applying IRL to predict from ongoing trajectories**. Papers [49, 50, 51] use standard IRL to predict goals of incomplete (or ongoing) trajectories. In specific, they first learn the reward function corresponding to each potential goal candidate from complete trajectories in the training phase and then use Bayesian methods to pick the most likely goal candidate of incomplete trajectories in the testing phase. However, these works are not in-trajectory learning since they do not learn a reward function from an incomplete ongoing trajectory.

**Online non-convex optimization**. This paper casts the in-trajectory learning problem as an online non-convex bi-level optimization problem where at each online iteration, a new state-action pair is input. Current literature on online non-convex optimization has two major categories. The first one is to use the regret originally defined in online convex optimization [52, 53]. However, it assumes to find a global optimal solution of a non-convex optimization problem at each online iteration. Therefore, the second category studies "local regret" [26, 27] and uses follow-the-leader-based methods to minimize the local regret. However, the follow-the-leader-based methods need to solve a non-convex optimization problem to obtain a near-stationary point at each online iteration, which can be computationally expensive and time-consuming. If the streaming data arrives at a fast speed, the computation at each online iteration may not be finished before the next data arrives. The computational burden of each online iteration can be mitigated by online gradient descent (OGD) methods where we only partially solve the non-convex optimization problem by one-step gradient descent at each online iteration. While OGD is widely studied in online convex optimization, it is rarely studied in online non-convex optimization. [11] uses OGD to quantify the local regret, however, its analysis only holds when the input data is identically independent distributed (i.i.d.). In contrast, the input data in our problem is not i.i.d. In specific, the input data at time $t$ (i.e., $(S_t^E, A_t^E)$) is affected by the input data at last step (i.e., $(S_{t-1}^E, A_{t-1}^E)$). This temporal correlation between any two consecutive input data makes it difficult to analyze the growth rate of the local regret.

**Regularization and meta-learning in IRL**. Moreover, the data of in-trajectory learning is extremely lacking since there is only one demonstrated trajectory and this trajectory is not complete during the learning process. The lack of data can easily lead to overfitting and a common way to alleviate this problem is to use regularizers [21, 54]. Inspired by humans' using relevant experience to help the inference, we introduce a novel regularization method called meta-regularization [25, 55]. Compared to the regularizers commonly used in IRL [21, 54], the meta-regularizer provides human-experience-like prior information which helps recover the reward function from few data. Similar to the meta-initialization method [55, 44] commonly used in IRL [56], meta-regularization provides an initialization that the algorithm starts at. However, more importantly, meta-regularization also provides a regularization term to avoid overfitting.

## A.1  Distinction from Maximum-likelihood inverse reinforcement learning (ML-IRL) [5]

We discuss our distinctions from ML-IRL from the following three aspects: problem setting, algorithm design, and theoretical analysis.

**Distinction in problem setting**. We study in-trajectory IRL and formulate an online optimization problem, while ML-IRL studies standard IRL and formulates an offline optimization problem.

**Distinctions in algorithm design**. ML-IRL and our algorithm both update policy and reward in a single loop. However, we propose a novel reward update mechanism specially designed for the in-trajectory learning case. This special design requires to use the current learned policy to complete the expert trajectory, which gives the algorithm the ability to consider for the future. This special design of reward update is novel compared to ML-IRL.

**Distinctions in theoretical analysis**. The analysis in our paper is substantially different from that in ML-IRL due to three facts: (1) The input data in our paper is not i.i.d., while the input data in ML-IRL is i.i.d. (2) We solve an online optimization problem, while ML-IRL solves an offline optimization problem. (3) Our analysis holds for continuous state-action space, while the analysis of ML-IRL is

limited to finite state-action space. We now discuss the distinctions in theoretical analysis caused by the three facts in detail.

In our case, the input data is not i.i.d. but temporally correlated, i.e., the input data $(s_t, a_t)$ is affected by the input data $(s_{t-1}, a_{t-1})$ at last time step. In contrast, the input data in ML-IRL is i.i.d. sampled from a pre-collected data set. To solve this non i.i.d. issue of the input data, we propose a novel theoretical technique that has three steps (detailed in Subsection 4.2). Step 1: We propose the stationary distribution $\mu^{\pi_E}$ and quantify the gradient norm difference between the real distribution $\mathbb{P}_t^{\pi_E}$ and this stationary distribution $\mu^{\pi_E}$ in Proposition 1. Step 2: We quantify the local regret over the stationary distribution in Lemma 2. The benefit of doing this is that the input data can be regarded as i.i.d. sampled from this stationary distribution. Step 3: We combine step 1 and step 2, and quantify the (local) regret over the real distribution, where the data is not i.i.d., in Theorem 1 and Theorem 2. We can see that step 1 and step 3 are to solve the non i.i.d. issue of the input data, so that the corresponding theorem statements (Proposition 1, Theorem 1, and Theorem 2) are novel compared to ML-IRL because ML-IRL does not have this non i.i.d. issue. The only theorem statement relevant to ML-IRL is Lemma 2 in step 2 where we both analyze the algorithm over a stationary distribution, and the data is i.i.d. sampled from the stationary distribution.

However, Lemma 2 in step 2 still has significant distinctions from ML-IRL because Lemma 2 quantifies the local regret in the context of online optimization, while ML-IRL quantifies convergence in the context of offline optimization. First, the objective function is dynamically changing in the online setting because the learner observes a new state-action pair at each online iteration, while the objective function is fixed in ML-IRL. Second, the local regret contains the term $L_t(\theta_t; (s_t^E, a_t^E))$, however, $\theta_t$ is computed before the learner knows $(s_t^E, a_t^E)$. This makes it more difficult to quantify the local regret because the learner does not know $L_t$ when it computes $\theta_t$. These two difficulties do not appear in the offline optimization in ML-IRL. To solve these two issues, we need to additionally construct a new time-invariant function $\bar{L}$ in Appendix B.4 and quantify the convergence of the new function $\bar{L}$. Then, in order to quantify the local regret of $\{L_t\}_{t \geq 0}$, we need to quantify the difference between the real loss function $\{L_t\}_{t \geq 0}$ and the constructed loss function $\bar{L}$.

Moreover, our theoretical analysis holds for continuous state-action space while the theoretical analysis in ML-IRL is limited to finite state-action space. The extension to continuous state-action space brings new difficulties and requires significant novel analysis. In general, the difficulties stem from two aspects: (1) The constants in ML-IRL, e.g., the smoothness constant of the loss function $L$ and the coefficient of convergence rate, include the term $|\mathcal{S}| \times |\mathcal{A}|$. When the state-action space is continuous, those constants are not finite because $|\mathcal{S}| \times |\mathcal{A}|$ is now infinite. To address this issue, we propose new methods to bound those constants. For example, in order to show that the loss function $L$ is smooth, rather than using $||\nabla L(\theta_1) - \nabla L(\theta_2)||$ to find the smoothness constant as in ML-IRL, we aim to show that $||\nabla^2 L(\theta)||$ is upper bounded by a constant $C_L$ in Lemma A.2 and this constant $C_L$ does not rely on $|\mathcal{S}| \times |A|$. Given that $||\nabla^2 L(\theta)|| \leq C_L$, the loss function $L$ is $C_L$-smooth. (2) Since the action space $\mathcal{A}$ is finite in ML-IRL, their proved properties of the $Q$-function $Q^{\text{soft}}$ (e.g., Lipschitz continuity, contraction property, monotonic improvement, and smoothness) can be easily extended to the value function $V^{\text{soft}}$ by summing over different actions $a \in \mathcal{A}$. When the action space becomes continuous, summing over infinitely many different actions does not preserve those properties. Thus we have to propose new methods to prove those propoerties of the value function $V^{\text{soft}}$. In specific, we prove the Lipschitz continuity, contraction property, monotonic improvement, and smoothness of the value function $V^{\text{soft}}$ in Claims 3-5 in Appendix B.4.

# B  Proof

This section provides the proof of all the proposition, lemmas, and theorems in the paper. To start with, we first introduce the expression of soft $Q$-function and soft Bellman policy.

## B.1  Notions

The soft $Q$-function and soft value function are:

$$Q_{\theta,\pi}^{\text{soft}}(s, a) \triangleq r_\theta(s, a) + \gamma \int_{s' \in \mathcal{S}} P(s'|s, a) V_{\theta,\pi}^{\text{soft}}(s') ds',$$

$$V_{\theta,\pi}^{\text{soft}}(s) \triangleq E_{S,A}^{\pi}\left[\sum_{t=0}^{\infty} \gamma^t \big(r_\theta(S_t, A_t) - \log \pi(A_t|S_t)\big)\Big| S_0 = s_0^E\right].$$

The soft Bellman policy is as follows:

$$\pi_\theta(a|s) = \frac{\exp(Q_\theta^{\text{soft}}(s,a))}{\exp(V_\theta^{\text{soft}}(s))},$$

$$Q_\theta^{\text{soft}}(s,a) = r_\theta(s,a) + \gamma \int_{s'\in\mathcal{S}} P(s'|s,a) V_\theta^{\text{soft}}(s')ds',$$

$$V_\theta^{\text{soft}}(s) = \log\left(\int_{a\in\mathcal{A}} \exp(Q_\theta^{\text{soft}}(s,a))da\right).$$

It has been proved [34] that the soft Bellman policy $\pi_\theta$ is the optimal solution of the lower-level problem (4). We define $J_\theta(s) \triangleq E_{S,A}^{\pi_\theta}[\sum_{t=0}^{\infty} \gamma^t r_\theta(S_t, A_t)|S_0 = s]$ as the expected cumulative reward of policy $\pi_\theta$ starting from state $s$ and $J_\theta(s,a) \triangleq E_{S,A}^{\pi_\theta}[\sum_{t=0}^{\infty} \gamma^t r_\theta(S_t, A_t)|S_0 = s, A_0 = a]$.

**Lemma 3.** *We have the gradient* $\nabla_\theta \log \pi_\theta(a|s) = E_{S,A}^{\pi_\theta}[\sum_{t=0}^{\infty} \gamma^t \nabla_\theta r_\theta(S_t, A_t)|S_0 = s, A_0 = a] - E_{S,A}^{\pi_\theta}[\sum_{t=0}^{\infty} \gamma^t \nabla_\theta r_\theta(S_t, A_t)|S_0 = s].$

*Proof.* Define $Z_\theta(s,a) \triangleq \exp(Q_\theta^{\text{soft}}(s,a))$ and $Z_\theta(s) \triangleq \exp(V_\theta^{\text{soft}}(s))$, therefore $Z_\theta$ is smooth in $\theta$ given that it is a composition of logarithmic, exponential, and linear functions of $r_\theta$ and $r_\theta$ is smooth in $\theta$ (Assumption 1).

$$\begin{aligned}
\nabla_\theta \log Z_\theta(s) &= \frac{\int_{a\in\mathcal{A}} \nabla_\theta Z_\theta(s,a)da}{Z_\theta(s)}, \\
&= \int_{a\in\mathcal{A}} \frac{Z_\theta(s,a)}{Z_\theta(s)} \nabla_\theta \log Z_\theta(s,a)da, \\
&= \int_{a\in\mathcal{A}} \pi_\theta(a|s)\left[\nabla_\theta r_\theta(s,a) + \gamma \int_{s'\in\mathcal{S}} P(s'|s,a)\nabla_\theta \log Z_\theta(s')ds'\right]da, \\
&= \int_{a\in\mathcal{A}} \pi_\theta(a|s)\left[\nabla_\theta r_\theta(s,a) + \gamma \int_{s'\in\mathcal{S}} P(s'|s,a) \int_{a'\in\mathcal{A}} \Big(\nabla_\theta r_\theta(s',a')\right. \\
&\left. + \gamma \int_{s''\in\mathcal{S}} P(s''|s',a')\nabla_\theta \log Z_\theta(s'')ds''\Big)da'ds'\right]da.
\end{aligned}$$

Keep the expansion, we can get $\nabla_\theta \log Z_\theta(s) = E_{S,A}^{\pi_\theta}[\sum_{t=0}^{\infty} \gamma^t \nabla_\theta r_\theta(S_t, A_t)|S_0 = s]$ and similarly we can get $\nabla_\theta \log Z_\theta(s,a) = E_{S,A}^{\pi_\theta}[\sum_{t=0}^{\infty} \gamma^t \nabla_\theta r_\theta(S_t, A_t)|S_0 = s, A_0 = a]$. Thus we have the gradient $\nabla_\theta \log \pi_\theta(a|s) = \nabla_\theta \log Z_\theta(s,a) - \nabla_\theta \log Z_\theta(s)$. $\square$

### B.2  Proof of Lemma 1

Recall that $\sum_{i=0}^{t} L_i(\theta; (s_i^E, a_i^E)) = -\sum_{i=0}^{t} \gamma^i \log \pi_\theta(a_i^E|s_i^E)$. When the dynamics $P$ is deterministic, we have that

$$\begin{aligned}
\nabla \sum_{i=0}^{t} L_i(\theta; (s_i^E, a_i^E)) &= -\sum_{i=0}^{t} \gamma^i \nabla_\theta \log \pi_\theta(a_i^E|s_i^E), \\
&= -\sum_{i=0}^{t} \gamma^i \left[\nabla_\theta Q_\theta^{\text{soft}}(s_i^E, a_i^E) - \nabla_\theta V_\theta^{\text{soft}}(s_i^E)\right], \\
&= -\sum_{i=0}^{t} \gamma^i \left[\nabla_\theta r_\theta(s_i^E, a_i^E) + \gamma \nabla_\theta V_\theta^{\text{soft}}(s_{i+1}^E) - \nabla_\theta V_\theta^{\text{soft}}(s_i^E)\right], \\
&= -\sum_{i=0}^{t} \gamma^i \nabla_\theta r_\theta(s_i^E, a_i^E) - \sum_{i=1}^{t+1} \gamma^i \nabla_\theta V_\theta^{\text{soft}}(s_i^E) + \sum_{i=0}^{t} \gamma^i \nabla_\theta V_\theta^{\text{soft}}(s_i^E),
\end{aligned}$$

$$= -\sum_{i=0}^{t} \gamma^i \nabla_\theta r_\theta(s_i^E, a_i^E) - \gamma^{t+1} \nabla_\theta V_\theta^{\text{soft}}(s_{t+1}^E) + \nabla_\theta V_\theta^{\text{soft}}(s_0^E),$$

$$\overset{(a)}{=} -\sum_{i=0}^{t} \gamma^i \nabla_\theta r_\theta(s_i^E, a_i^E) - E_{S,A}^{\pi_\theta}\left[\sum_{i=t+1}^{\infty} \gamma^i \nabla_\theta r_\theta(S_i, A_i) | S_i = s_{t+1}^E\right]$$

$$+ E_{S,A}^{\pi_\theta}\left[\sum_{i=0}^{\infty} \gamma^i \nabla_\theta r_\theta(S_i, A_i) | S_0 = s_0^E\right],$$

$$= -\sum_{i=0}^{t} \gamma^i \nabla_\theta r_\theta(s_i^E, a_i^E) - E_{S,A}^{\pi_\theta}\left[\sum_{i=t+1}^{\infty} \gamma^i \nabla_\theta r_\theta(S_i, A_i) | S_t = s_t^E, A_t = a_t^E\right]$$

$$+ E_{S,A}^{\pi_\theta}\left[\sum_{i=0}^{\infty} \gamma^i \nabla_\theta r_\theta(S_i, A_i) | S_0 = s_0^E\right],$$

where equality $(a)$ follows from the proof of Lemma 3.

When the dynamics $P$ is stochastic, we can prove that the above gradient is an unbiased estimate of $\nabla E_{\{(S_i^E, A_i^E) \sim \mathbb{P}_i^{\pi^E}(\cdot, \cdot)(\cdot, \cdot)\}_{i \geq 0}}\left[\sum_{i=0}^{t} L_i(\theta; (S_i^E, A_i^E))\right]$:

$$\nabla E_{\{(S_i^E, A_i^E) \sim \mathbb{P}_i^{\pi^E}(\cdot, \cdot)(\cdot, \cdot)\}_{i \geq 0}}\left[\sum_{i=0}^{t} L_i(\theta; (S_i^E, A_i^E))\right],$$

$$= \nabla E_{\{(S_i^E, A_i^E) \sim \mathbb{P}_i^{\pi^E}(\cdot, \cdot)(\cdot, \cdot)\}_{i \geq 0}}\left[-\sum_{i=0}^{t} \gamma^i \nabla_\theta \log \pi_\theta(A_i^E | S_i^E)\right],$$

$$= E_{\{(S_i^E, A_i^E) \sim \mathbb{P}_i^{\pi^E}(\cdot, \cdot)(\cdot, \cdot)\}_{i \geq 0}}\left[-\sum_{i=0}^{t} \gamma^i [\nabla_\theta r_\theta(S_i^E, A_i^E) + \gamma E_{S_{i+1} \sim P(\cdot | S_i^E, A_i^E)}[\nabla_\theta V_\theta^{\text{soft}}(S_{i+1})]\right.$$

$$\left. - \nabla_\theta V_\theta^{\text{soft}}(S_i^E)]\right],$$

$$= E_{\{(S_i^E, A_i^E) \sim \mathbb{P}_i^{\pi^E}(\cdot, \cdot)(\cdot, \cdot)\}_{i \geq 0}}\left[-\sum_{i=0}^{t} \gamma^i [\nabla_\theta r_\theta(S_i^E, A_i^E) + \gamma \nabla_\theta V_\theta^{\text{soft}}(S_{i+1}^E) - \nabla_\theta V_\theta^{\text{soft}}(S_i^E)]\right],$$

$$= E_{\{(S_i^E, A_i^E) \sim \mathbb{P}_i^{\pi^E}(\cdot, \cdot)(\cdot, \cdot)\}_{i \geq 0}}\left[-\sum_{i=0}^{t} \gamma^i \nabla_\theta r_\theta(S_i^E, A_i^E) - \gamma^{t+1} \nabla_\theta V_\theta^{\text{soft}}(S_{t+1}^E)\right.$$

$$\left. + E_{S,A}^{\pi_\theta}\left[\sum_{i=0}^{\infty} \gamma^i \nabla_\theta r_\theta(S_i, A_i) | S_0 = S_0^E\right]\right],$$

$$\overset{(b)}{=} E_{\{(S_i^E, A_i^E) \sim \mathbb{P}_i^{\pi^E}(\cdot, \cdot)(\cdot, \cdot)\}_{i \geq 0}}\left[-\sum_{i=0}^{t} \gamma^i \nabla_\theta r_\theta(S_i^E, A_i^E)\right.$$

$$\left. - E_{S,A}^{\pi_\theta}[\sum_{i=t+1}^{\infty} \gamma^i \nabla_\theta r_\theta(S_i, A_i) | S_t = S_t^E, A_t = A_t^E] + E_{S,A}^{\pi_\theta}[\sum_{i=0}^{\infty} \gamma^i \nabla_\theta r_\theta(S_i, A_i) | S_0 = S_0^E]\right],$$

where equality $(b)$ follows from the fact that

$$E_{\{(S_i^E, A_i^E) \sim \mathbb{P}_i^{\pi^E}(\cdot, \cdot)(\cdot, \cdot)\}_{i \geq 0}}\left[E_{S,A}^{\pi_\theta}[\sum_{i=t+1}^{\infty} \gamma^i \nabla_\theta r_\theta(S_i, A_i) | S_t = S_t^E, A_t = A_t^E]\right],$$

$$= E_{\{(S_i^E, A_i^E) \sim \mathbb{P}_i^{\pi^E}(\cdot, \cdot)(\cdot, \cdot)\}_{i \geq 0}}\left[E_{S,A}^{\pi_\theta}[\sum_{i=t+1}^{\infty} \gamma^i \nabla_\theta r_\theta(S_i, A_i) | S_{t+1} = S_{t+1}^E]\right],$$

because $\mathbb{P}_{t+1}^{\pi^E}(\cdot) = \mathbb{P}_t^{\pi^E}(S_t^E, A_t^E) P(\cdot | S_t^E, A_t^E)$ and $S_{t+1}^E \sim P(\cdot | S_t^E, A_t^E)$.

Since we quantify the local regret in expectation in Theorem 1, this unbiased estimate can be used when the dynamics is stochastic.

## B.3 Proof of Proposition 1

From Assumption 2, we know that $d_{\mathrm{TV}}(\mathbb{P}_t^{\pi_E}(\cdot), \mu^{\pi_E}(\cdot)) = \frac{1}{2}\int_{s\in\mathcal{S}}|\mathbb{P}_t^{\pi_E}(s) - \mu^{\pi_E}(s)|ds \leq C_M\rho^t$ where the initial state is $s_0$. For any state-action pair $(s,a) \in \mathcal{S}\times\mathcal{A}$, we know that $\mathbb{P}_t^{\pi_E}(s,a) = \mathbb{P}_t^{\pi_E}(s)\pi_E(a|s)$ and $\mu^{\pi_E}(s,a) = \mu^{\pi_E}(s)\pi_E(a|s)$. Therefore, we have that:

$$d_{\mathrm{TV}}(\mathbb{P}^{\pi_E}(\cdot,\cdot), \mu^{\pi_E}(\cdot,\cdot)),$$

$$= \frac{1}{2}\int_{s\in\mathcal{S}}\int_{a\in\mathcal{A}}|\mathbb{P}_t^{\pi_E}(s)\pi_E(a|s) - \mu^{\pi_E}(s)\pi_E(a|s)|dsda,$$

$$= \frac{1}{2}\int_{s\in\mathcal{S}}\int_{a\in\mathcal{A}}|\mathbb{P}_t^{\pi_E}(s) - \mu^{\pi_E}(s)|\pi_E(a|s)dsda,$$

$$= \frac{1}{2}\int_{s\in\mathcal{S}}|\mathbb{P}_t^{\pi_E}(s) - \mu^{\pi_E}(s)|ds,$$

$$\leq C_M\rho^t.$$

**Claim 1.** *The trajectory of $\theta_t$ is bounded, i.e., $\|\theta_t - \bar{\theta}\| \leq \frac{2\bar{C}_r}{\lambda}$ for any $t \geq 0$.*

*Proof.*

$$\|\theta_{t+1} - \bar{\theta}\| = \|\theta_t - \alpha_t g_t - \bar{\theta}\|,$$

$$= \left\|\theta_t - \bar{\theta} - \alpha_t\left[\sum_{i=0}^{\infty}\gamma^i\nabla_\theta r_{\theta_t}(s_i', a_i') - \sum_{i=0}^{\infty}\gamma^i\nabla_\theta r_{\theta_t}(s_i'', a_i'') + \frac{\lambda(1-\gamma^{t+1})}{1-\gamma}(\theta_t - \bar{\theta})\right]\right\|,$$

$$= \left\|(1 - \frac{\alpha_t\lambda(1-\gamma^{t+1})}{1-\gamma})(\theta_t - \bar{\theta}) - \alpha_t\left[\sum_{i=0}^{\infty}\gamma^i\nabla_\theta r_{\theta_t}(s_i', a_i') - \sum_{i=0}^{\infty}\gamma^i\nabla_\theta r_{\theta_t}(s_i'', a_i'')\right]\right\|,$$

$$\overset{(a)}{\leq} (1 - \frac{\alpha_t\lambda(1-\gamma^{t+1})}{1-\gamma})\|\theta_t - \bar{\theta}\| + \alpha_t\left\|\sum_{i=0}^{\infty}\gamma^i\nabla_\theta r_{\theta_t}(s_i', a_i') - \sum_{i=0}^{\infty}\gamma^i\nabla_\theta r_{\theta_t}(s_i'', a_i'')\right\|,$$

$$\overset{(b)}{\leq} (1 - \frac{\alpha_t\lambda(1-\gamma^{t+1})}{1-\gamma})\|\theta_t - \bar{\theta}\| + \frac{2\alpha_t\bar{C}_r}{1-\gamma},$$

$$\leq (1 - \frac{\alpha_t\lambda}{1-\gamma})\|\theta_t - \bar{\theta}\| + \frac{2\alpha_t\bar{C}_r}{1-\gamma},$$

where $(a)$ follows triangle inequality and $(b)$ uses the upper bound of $\nabla_\theta r_\theta$ in Assumption 1. Therefore, we have the following relation:

$$\|\theta_{t+1} - \bar{\theta}\| - \frac{2\bar{C}_r}{\lambda} \leq (1 - \frac{\alpha_t\lambda}{1-\gamma})\left(\|\theta_t - \bar{\theta}\| - \frac{2\bar{C}_r}{\lambda}\right),$$

$$\Rightarrow \|\theta_t - \bar{\theta}\| \leq (1 - \frac{\alpha_t\lambda}{1-\gamma})^t\left(\|\theta_0 - \bar{\theta}\| - \frac{2\bar{C}_r}{\lambda}\right) + \frac{2\bar{C}_r}{\lambda},$$

$$\overset{(c)}{=} \frac{2\bar{C}_r}{\lambda}\left[1 - (1 - \frac{\alpha_t\lambda}{1-\gamma})^t\right] \overset{(d)}{\leq} \frac{2\bar{C}_r}{\lambda},$$

where $(c)$ follows the fact that $\theta_0 = \bar{\theta}$ and $(d)$ follows the fact that $\alpha_t \leq \frac{1-\gamma}{\lambda}$. $\qquad\square$

Recall that the loss function $L_i(\theta; (s_i^E, a_i^E)) = -\gamma^i\log\pi_\theta(a_i^E|s_i^E) + \frac{\lambda\gamma^i}{2}\|\theta - \bar{\theta}\|^2$ and thus $\nabla L_i(\theta; (s_i^E, a_i^E)) = -\gamma^i[\nabla_\theta Q_\theta^{\mathrm{soft}}(s_i^E, a_i^E) - \nabla_\theta V_\theta^{\mathrm{soft}}(s_i^E)] + \lambda\gamma^i(\theta - \bar{\theta})$. From Lemma 3, we know that $\nabla_\theta V_\theta^{\mathrm{soft}}(s_i^E) = E_{S,A}^{\pi_\theta}[\sum_{t=0}^{\infty}\gamma^t\nabla_\theta r_\theta(S_t, A_t)|S_0 = s_i^E]$ and $\nabla_\theta Q_\theta^{\mathrm{soft}}(s_i^E, a_i^E) = E_{S,A}^{\pi_\theta}[\sum_{t=0}^{\infty}\gamma^t\nabla_\theta r_\theta(S_t, A_t)|S_0 = s_i^E, A_0 = a_i^E]$. Then, $\|\nabla_\theta V_\theta^{\mathrm{soft}}(s_i^E)\| \leq \frac{\bar{C}_r}{1-\gamma}$ and $\|\nabla_\theta Q_\theta^{\mathrm{soft}}(s_i^E, a_i^E)\| \leq \frac{\bar{C}_r}{1-\gamma}$.

Now we can see that

$$\|\nabla L_i(\theta_t; ; (s_i^E, a_i^E))\| = \gamma^i\|\nabla_\theta Q_{\theta_t}^{\mathrm{soft}}(s_i^E, a_i^E) - \nabla_\theta V_{\theta_t}^{\mathrm{soft}}(s_i^E) + \lambda(\theta_t - \bar{\theta})\|,$$

$$\leq \gamma^i ||\nabla_\theta Q_{\theta_t}^{\text{soft}}(s_i^E, a_i^E) - \nabla_\theta V_{\theta_t}^{\text{soft}}(s_i^E) + \lambda(\theta_t - \bar\theta)|| \leq \gamma^i \left(\frac{2\bar C_r}{1-\gamma} + 2\bar C_r\right),$$

$$\Rightarrow ||\nabla L_i(\theta_t; (s_i^E, a_i^E))||^2 \leq 4\bar C_r^2 \gamma^{2i} \left(\frac{2-\gamma}{1-\gamma}\right)^2.$$

Therefore, we have that

$$\left| E_{(S_i^E, A_i^E) \sim \mathbb{P}_i^{\pi_E}(\cdot, \cdot)}\left[||\nabla L_i(\theta_t; (S_i^E, A_i^E))||^2\right] - E_{(S_i^E, A_i^E) \sim \mu^{\pi_E}(\cdot, \cdot)}\left[||\nabla L_i(\theta_t; (S_i^E, A_i^E))||^2\right]\right|,$$

$$= \left| \int_{s \in \mathcal{S}} \int_{a \in \mathcal{A}} \mathbb{P}_i^{\pi_E}(s, a)||\nabla L_i(\theta_t; (s, a))||^2 dads \right.$$

$$\left. - \int_{s \in \mathcal{S}} \int_{a \in \mathcal{A}} \mu^{\pi_E}(s, a)||\nabla L_i(\theta_t; (s, a))||^2 dads \right|,$$

$$\leq \int_{s \in \mathcal{S}} \int_{a \in \mathcal{A}} |\mathbb{P}_i^{\pi_E}(s, a) - \mu^{\pi_E}(s, a)| \cdot ||\nabla L_i(\theta_t; (s, a))||^2 dads,$$

$$\leq 2C_M \rho^i \cdot 4\bar C_r^2 \gamma^{2i} \left(\frac{2-\gamma}{1-\gamma}\right)^2 = 8C_M \bar C_r^2 \left(\frac{2-\gamma}{1-\gamma}\right)^2 \rho^i \gamma^{2i}.$$

**Lemma 4.** *Suppose Assumptions 1-2 hold, the we have the following for any $(s, a) \in \mathcal{S} \times \mathcal{A}$ and any $\theta_1, \theta_2, t$: $||\nabla L_t(\theta_1, (s, a)) - \nabla L_t(\theta_2, (s, a))|| \leq C_L ||\theta_1 - \theta_2||$ and $|Q_{\theta_1, \pi_{\theta_1}}^{soft}(s, a) - Q_{\theta_2, \pi_{\theta_2}}^{soft}(s, a)| \leq C_Q ||\theta_1 - \theta_2||$, where $C_L = \frac{2\bar C_r}{1-\gamma} + \frac{4\bar C_r^3}{(1-\gamma)^4} + \lambda$ and $C_Q = \frac{\bar C_r}{1-\gamma}$.*

*Proof.* Note that $Q_{\theta, \pi_\theta}^{\text{soft}} = Q_\theta^{\text{soft}}$ and $\nabla_\theta Q_\theta^{\text{soft}}(s, a) = E_{S, A}^{\pi_\theta}[\sum_{t=0}^\infty \gamma^t \nabla_\theta r_\theta(S_t, A_t)|S_0 = s, A_0 = a]$ (proof of Lemma 3). Therefore, we have that

$$||\nabla_\theta Q_\theta^{\text{soft}}(s, a)|| \leq \frac{\bar C_r}{1-\gamma} \triangleq C_Q$$

We know from Lemma 1 that $\nabla L_t(\theta; (s_t^E, a_t^E)) = -\gamma^t [\nabla_\theta Q_\theta^{\text{soft}}(s_t^E, a_t^E) - \nabla_\theta V_\theta^{\text{soft}}(s_t^E)] + \lambda\gamma^t(\theta - \bar\theta)$. To find the smoothness constant of $L_t$, we need to compute the Hessian of $L_t$. First, we have that

$$\nabla_{\theta\theta}^2 Q_\theta^{\text{soft}}(s, a) = \nabla_\theta E_{S, A}^{\pi_\theta}[\sum_{t=0}^\infty \gamma^t \nabla_\theta r_\theta(S_t, A_t)|S_0 = s, A_0 = a],$$

$$= \nabla_{\theta\theta}^2 r_\theta(s, a) + \gamma \int_{s' \in \mathcal{S}} P(s'|s, a)\nabla_\theta E_{S, A}^{\pi_\theta}[\sum_{t=0}^\infty \gamma^t \nabla_\theta r_\theta(S_t, A_t)|S_0 = s']ds',$$

$$= \nabla_{\theta\theta}^2 r_\theta(s, a)$$

$$+ \gamma \int_{s' \in \mathcal{S}} P(s'|s, a)\nabla_\theta \int_{a' \in \mathcal{A}} \pi_\theta(a'|s') E_{S, A}^{\pi_\theta}[\sum_{t=0}^\infty \gamma^t \nabla_\theta r_\theta(S_t, A_t)|S_0 = s', A_0 = a']da'ds',$$

$$= \nabla_{\theta\theta}^2 r_\theta(s, a) + \gamma \int_{s' \in \mathcal{S}} P(s'|s, a) \int_{a' \in \mathcal{A}} \left[\nabla_\theta \pi_\theta(a'|s') \cdot E_{S, A}^{\pi_\theta}[\sum_{t=0}^\infty \gamma^t \nabla_\theta r_\theta(S_t, A_t)|S_0 = s',\right.$$

$$\left. A_0 = a'] + \pi_\theta(a'|s') \cdot \nabla_\theta E_{S, A}^{\pi_\theta}[\sum_{t=0}^\infty \gamma^t \nabla_\theta r_\theta(S_t, A_t)|S_0 = s', A_0 = a']\right]da'ds'.$$

Keep the expansion, we can get

$$\nabla_{\theta\theta}^2 Q_\theta^{\text{soft}}(s, a) = E_{S, A}^{\pi_\theta}[\sum_{t=0}^\infty \gamma^t \nabla_{\theta\theta}^2 r_\theta(S_t, A_t)|S_0 = s_0, A_0 = a_0]$$

$$+ E_{S, A}^{\pi_\theta}\left[\sum_{i=0}^\infty \gamma^i \nabla_\theta \pi_\theta(A_i|S_i) \cdot E_{S', A'}^{\pi_\theta}[\sum_{t=0}^\infty \gamma^t \nabla_\theta r_\theta(S_t', A_t')|S_0' = S_i, A_0' = A_i]\Big|S_0 = s_0, A_0 = a_0\right].$$

Now we take a look at the second term in the above equality:

$$\nabla_\theta \pi_\theta(a|s) \cdot E_{S,A}^{\pi_\theta}[\sum_{t=0}^\infty \gamma^t \nabla_\theta r_\theta(S_t, A_t)|S_0 = s, A_0 = a],$$

$$= \pi_\theta(a|s) \nabla_\theta \log \pi_\theta(a|s) \cdot E_{S,A}^{\pi_\theta}[\sum_{t=0}^\infty \gamma^t \nabla_\theta r_\theta(S_t, A_t)|S_0 = s, A_0 = a],$$

$$= \pi_\theta(a|s)\left[\nabla_\theta Q_\theta^{\text{soft}}(s, a) - \nabla_\theta V_\theta^{\text{soft}}(s)\right] \cdot E_{S,A}^{\pi_\theta}[\sum_{t=0}^\infty \gamma^t \nabla_\theta r_\theta(S_t, A_t)|S_0 = s, A_0 = a],$$

$$\Rightarrow \left\|\nabla_\theta \pi_\theta(a|s) \cdot E_{S,A}^{\pi_\theta}[\sum_{t=0}^\infty \gamma^t \nabla_\theta r_\theta(S_t, A_t)|S_0 = s, A_0 = a]\right\|,$$

$$\leq (\frac{\bar{C}_r}{1-\gamma} + \frac{\bar{C}_r}{1-\gamma}) \cdot \frac{\bar{C}_r}{1-\gamma} = \frac{2\bar{C}_r^3}{(1-\gamma)^3}.$$

Therefore, we have that

$$||\nabla_{\theta\theta}^2 Q_\theta^{\text{soft}}(s, a)|| \leq \left\|E_{S,A}^{\pi_\theta}[\sum_{t=0}^\infty \gamma^t \nabla_{\theta\theta}^2 r_\theta(S_t, A_t)|S_0 = s_0, A_0 = a_0]\right\|$$

$$+ \left\|E_{S',A'}^{\pi_\theta}\left[\sum_{i=0}^\infty \gamma^i \nabla_\theta \pi_\theta(A_i'|S_i') \cdot E_{S,A}^{\pi_\theta}[\sum_{t=0}^\infty \gamma^t \nabla_\theta r_\theta(S_t, A_t)|S_0 = S_i', A_0 = A_i']\right]\right\|,$$

$$\leq \frac{\tilde{C}_r}{1-\gamma} + \sum_{t=0}^\infty \gamma^t \frac{2\bar{C}_r^3}{(1-\gamma)^3} = \frac{\tilde{C}_r}{1-\gamma} + \frac{2\bar{C}_r^3}{(1-\gamma)^4}.$$

Similarly, we can get $||\nabla_{\theta\theta}^2 V_\theta^{\text{soft}}(s)|| \leq \frac{\tilde{C}_r}{1-\gamma} + \frac{2\bar{C}_r^3}{(1-\gamma)^4}$. Therefore, we have that

$$||\nabla^2 L_t(\theta; (s_t^E, a_t^E))|| \leq \gamma^t \left(||\nabla_{\theta\theta}^2 Q_\theta^{\text{soft}}(s_t^E, a_t^E)|| + ||\nabla_{\theta\theta}^2 V_\theta^{\text{soft}}(s_t^E)|| + \lambda\right),$$

$$\leq \gamma^t \left(\frac{2\tilde{C}_r}{1-\gamma} + \frac{4\bar{C}_r^3}{(1-\gamma)^4} + \lambda\right) \leq \frac{2\tilde{C}_r}{1-\gamma} + \frac{4\bar{C}_r^3}{(1-\gamma)^4} + \lambda \triangleq C_L. \tag{6}$$

$\square$

### B.4 Proof of Lemma 2

This proof is based on the proof in ML-IRL [5]. The differences are: (i) their proof only holds for finite state-action space while we extend to continuous state-action space; (ii) their analysis is for offline settings while we extend to online settings to quantify the local regret. We first introduce the following claims which serve as building blocks in this subsection.

**Claim 2.** *For any given policy $\pi$ and state-action pair $(s, a)$, it holds that $|Q_{\theta_1,\pi}^{soft}(s, a) - Q_{\theta_2,\pi}^{soft}(s, a)| \leq C_Q||\theta_1 - \theta_2||$ and $|V_{\theta_1,\pi}^{soft}(s) - V_{\theta_2,\pi}^{soft}(s)| \leq C_Q||\theta_1 - \theta_2||$.*

*Proof.*

$$Q_{\theta_1,\pi}^{\text{soft}}(s, a) - Q_{\theta_2,\pi}^{\text{soft}}(s, a),$$

$$= E_{S,A}^\pi\left[\sum_{t=0}^\infty \gamma^t \left[r_{\theta_1}(S_t, A_t) - \log \pi(A_t|S_t)\right]\Big|S_0 = s, A_0 = a\right]$$

$$- E_{S,A}^\pi\left[\sum_{t=0}^\infty \gamma^t \left[r_{\theta_2}(S_t, A_t) - \log \pi(A_t|S_t)\right]\Big|S_0 = s, A_0 = a\right],$$

$$= E_{S,A}^\pi\left[\sum_{t=0}^\infty \gamma^t \left[r_{\theta_1}(S_t, A_t) - r_{\theta_2}(S_t, A_t)\right]\Big|S_0 = s, A_0 = a\right],$$

$$\Rightarrow |Q_{\theta_1,\pi}(s,a) - Q_{\theta_2,\pi}(s,a)| \le \sum_{t=0}^{\infty} \gamma^t \bar{C}_r ||\theta_1 - \theta_2|| = \frac{\bar{C}_r}{1-\gamma}||\theta_1 - \theta_2|| = C_Q||\theta_1 - \theta_2||.$$

Similarly, we can get that

$$|V_{\theta_1,\pi}^{\text{soft}}(s) - V_{\theta_2,\pi}^{\text{soft}}(s)| \le E_{S,A}^{\pi}\left[\sum_{t=0}^{\infty} \gamma^t[r_{\theta_1}(S_t, A_t) - r_{\theta_2}(S_t, A_t)]\Big| S_0 = s\right],$$

$$\le \sum_{t=0}^{\infty} \gamma^t \bar{C}_r ||\theta_1 - \theta_2|| = \frac{\bar{C}_r}{1-\gamma}||\theta_1 - \theta_2|| = C_Q||\theta_1 - \theta_2||.$$

$\square$

**Claim 3.** *The soft Bellman operator $\mathcal{T}_{\theta}^{soft}$:*

$$(\mathcal{T}_{\theta}^{soft}Q)(s,a) \triangleq r_{\theta}(s,a) + \gamma \int_{s'\in\mathcal{S}} P(s'|s,a)\log\left[\int_{a'\in\mathcal{A}} \exp(Q(s',a'))da'\right]ds',$$

$$(\mathcal{T}_{\theta}^{soft}V)(s) \triangleq \log\left[\int_{a\in\mathcal{A}} \exp\left(r_{\theta}(s,a) + \gamma \int_{s'\in\mathcal{S}} P(s'|s,a)V(s')ds'\right)da\right],$$

*is a contraction map with constant $\gamma$.*

*Proof.* It has been proved that $\mathcal{T}_{\theta}^{\text{soft}}Q$ is a contraction map with constant $\gamma$ (Appendix A.2 in [34]). Here we show that $\mathcal{T}_{\theta}^{\text{soft}}V$ is a contraction map with constant $\gamma$. Define a norm of $V$ as $||V_1 - V_2|| = \sup_{s\in\mathcal{S}} |V_1(s) - V_2(s)|$ and suppose $||V_1 - V_2|| = \epsilon$. Then we have that

$$\mathcal{T}_{\theta}^{\text{soft}}V_1(s) = \log\left[\int_{a\in\mathcal{A}} \exp\left(r_{\theta}(s,a) + \gamma \int_{s'\in\mathcal{S}} P(s'|s,a)V_1(s')ds'\right)da\right],$$

$$\le \log\left[\int_{a\in\mathcal{A}} \exp\left(r_{\theta}(s,a) + \gamma \int_{s'\in\mathcal{S}} P(s'|s,a)[V_2(s') + \epsilon]ds'\right)da\right],$$

$$= \log\left[\int_{a\in\mathcal{A}} \exp\left(r_{\theta}(s,a) + \gamma \int_{s'\in\mathcal{S}} P(s'|s,a)V_2(s')ds' + \gamma\epsilon\right)da\right],$$

$$= \log\left[\int_{a\in\mathcal{A}} \exp(\gamma\epsilon)\exp\left(r_{\theta}(s,a) + \gamma \int_{s'\in\mathcal{S}} P(s'|s,a)V_2(s')ds'\right)da\right],$$

$$= \mathcal{T}_{\theta}^{\text{soft}}V_2(s) + \gamma\epsilon.$$

Similarly, we can get $\mathcal{T}_{\theta}^{\text{soft}}V_1(s) \ge \mathcal{T}_{\theta}^{\text{soft}}V_2(s) - \gamma\epsilon$. Therefore, $||\mathcal{T}_{\theta}^{\text{soft}}V_1 - \mathcal{T}_{\theta}^{\text{soft}}V_2|| \le \gamma\epsilon = \gamma||V_1 - V_2||$. $\square$

**Claim 4.** *It holds that $Q_{\theta_t,\pi_{t+1}}^{soft}(s,a) \ge \mathcal{T}_{\theta_t}^{soft}(Q_{\theta_t,\pi_t}^{soft})(s,a)$ and $V_{\theta_t,\pi_{t+1}}^{soft}(s) \ge \mathcal{T}_{\theta_t}^{soft}(V_{\theta_t,\pi_t}^{soft})(s)$ for any $(s,a)$.*

*Proof.*

$$Q_{\theta_t,\pi_{t+1}}^{\text{soft}}(s,a) \overset{(i)}{=} r_{\theta_t}(s,a) + \gamma \int_{s'\in\mathcal{S}} P(s'|s,a)E_{a'\sim\pi_{t+1}}[Q_{\theta_t,\pi_{t+1}}^{\text{soft}}(s',a') - \log\pi_{t+1}(a'|s')]ds',$$

$$\overset{(ii)}{\ge} r_{\theta_t}(s,a) + \gamma \int_{s'\in\mathcal{S}} P(s'|s,a)E_{A'\sim\pi_{t+1}(\cdot|s')}[Q_{\theta_t,\pi_t}^{\text{soft}}(s',A') - \log\pi_{t+1}(A'|s')]ds',$$

$$= r_{\theta_t}(s,a) + \gamma \int_{s'\in\mathcal{S}} P(s'|s,a)\log\left[\int_{a'\in\mathcal{A}} \exp(Q_{\theta_t,\pi_t}^{\text{soft}}(s',a'))da'\right]ds',$$

$$= \mathcal{T}_{\theta}^{\text{soft}}(Q_{\theta_t,\pi_t}^{\text{soft}})(s,a),$$

where $(i)$ follow equations (2)-(3) in [35] and $(ii)$ follows policy improvement theorem (Theorem 4 in [34]). Similarly, we can get that

$$V_{\theta_t,\pi_{t+1}}^{\text{soft}}(s) = E_{A\sim\pi_{t+1}(\cdot|s)}[Q_{\theta_t,\pi_{t+1}}^{\text{soft}}(s,A) - \log\pi_{t+1}(A|s)],$$

$$\geq E_{A\sim\pi_{t+1}(\cdot|s)}[Q^{\text{soft}}_{\theta_t,\pi_t}(s,A) - \log\pi_{t+1}(A|s)],$$

$$= \log\left[\int_{a\in\mathcal{A}}\exp(Q^{\text{soft}}_{\theta_t,\pi_t}(s,a))da\right],$$

$$= \log\left[\int_{a\in\mathcal{A}}\exp\left(r_{\theta_t}(s,a) + \gamma\int_{s'\in\mathcal{S}}P(s'|s,a)V^{\text{soft}}_{\theta_t,\pi_t}(s')ds'\right)da\right],$$

$$= \mathcal{T}^{\text{soft}}_{\theta_t}(V^{\text{soft}}_{\theta_t,\pi_t})(s).$$

$\square$

**Claim 5.** *The following holds for any $(s,a)\in\mathcal{S}\times\mathcal{A}$ and $\theta_1$, $\theta_2$, t:*

$$|V^{soft}_{\theta_1}(s,a) - V^{soft}_{\theta_2}(s,a)| \leq C_Q||\theta_1 - \theta_2||.$$

*Proof.* Note that $\nabla_\theta V^{\text{soft}}_\theta(s) = E^{\pi_\theta}_{S,A}[\sum_{t=0}^\infty\gamma^t\nabla_\theta r_\theta(S_t,A_t)|S_0=s]$ (proof of Lemma 3), therefore

$$||\nabla_\theta V^{\text{soft}}_\theta(s)|| \leq \frac{\bar{C}_r}{1-\gamma} = C_Q.$$

$\square$

We first show the convergence of $\pi_t$ in Algorithm 1. For any $(s,a)\in\mathcal{S}\times\mathcal{A}$, we know that

$$|\log\pi_{t+1}(a|s) - \log\pi_{\theta_t}(a|s)| \leq |Q^{\text{soft}}_{\theta_t,\pi_t}(s,a) - Q^{\text{soft}}_{\theta_t}(s,a)| + |V^{\text{soft}}_{\theta_t,\pi_t}(s) - V^{\text{soft}}_{\theta_t}(s)|.$$

Now we take a look at the term $|Q^{\text{soft}}_{\theta_t,\pi_t}(s,a) - Q^{\text{soft}}_{\theta_t}(s,a)|$:

$$|Q^{\text{soft}}_{\theta_t,\pi_t}(s,a) - Q^{\text{soft}}_{\theta_t}(s,a)|,$$

$$\leq |Q^{\text{soft}}_{\theta_t,\pi_t}(s,a) - Q^{\text{soft}}_{\theta_{t-1},\pi_t}(s,a)| + |Q^{\text{soft}}_{\theta_{t-1},\pi_t}(s,a) - Q^{\text{soft}}_{\theta_{t-1}}(s,a)| + |Q^{\text{soft}}_{\theta_{t-1}}(s,a) - Q^{\text{soft}}_{\theta_t}(s,a)|,$$

$$\overset{(a)}{\leq} C_Q||\theta_t - \theta_{t-1}|| + |Q^{\text{soft}}_{\theta_{t-1},\pi_t}(s,a) - Q^{\text{soft}}_{\theta_{t-1}}(s,a)| + C_Q||\theta_t - \theta_{t-1}||,$$

$$= 2C_Q||\theta_t - \theta_{t-1}|| + |Q^{\text{soft}}_{\theta_{t-1},\pi_t}(s,a) - Q^{\text{soft}}_{\theta_{t-1}}(s,a)|,$$

$$\overset{(b)}{=} 2C_Q||\theta_t - \theta_{t-1}|| + Q^{\text{soft}}_{\theta_{t-1}}(s,a) - Q^{\text{soft}}_{\theta_{t-1},\pi_t}(s,a),$$

$$\overset{(c)}{\leq} 2C_Q||\theta_t - \theta_{t-1}|| + Q^{\text{soft}}_{\theta_{t-1}}(s,a) - \mathcal{T}^{\text{soft}}_{\theta_{t-1}}(Q^{\text{soft}}_{\theta_{t-1},\pi_{t-1}})(s,a),$$

$$\overset{(d)}{=} 2C_Q||\theta_t - \theta_{t-1}|| + \mathcal{T}^{\text{soft}}_{\theta_{t-1}}(Q^{\text{soft}}_{\theta_{t-1}})(s,a) - \mathcal{T}^{\text{soft}}_{\theta_{t-1}}(Q^{\text{soft}}_{\theta_{t-1},\pi_{t-1}})(s,a),$$

$$\overset{(e)}{\leq} 2C_Q||\theta_t - \theta_{t-1}|| + \gamma|Q^{\text{soft}}_{\theta_{t-1}}(s,a) - Q^{\text{soft}}_{\theta_{t-1},\pi_{t-1}}(s,a)|, \tag{7}$$

where $(a)$ follows Lemma 2 and Claim 2, $(b)$ follows the fact that $\pi_\theta$ is the optimal solution, $(c)$ follows Claim 4, $(d)$ follows the fact that $Q^{\text{soft}}_{\theta_{t-1}}$ is a fixed point of $\mathcal{T}^{\text{soft}}_{\theta_{t-1}}$ (Theorem 2 in [34]), and $(e)$ follows Claim 3.

Similarly we can bound the term $|V^{\text{soft}}_{\theta_t,\pi_t}(s) - V^{\text{soft}}_{\theta_t}(s)|$:

$$|V^{\text{soft}}_{\theta_t,\pi_t}(s) - V^{\text{soft}}_{\theta_t}(s)|,$$

$$\leq |V^{\text{soft}}_{\theta_t,\pi_t}(s) - V^{\text{soft}}_{\theta_{t-1},\pi_t}(s)| + |V^{\text{soft}}_{\theta_{t-1},\pi_t}(s) - V^{\text{soft}}_{\theta_{t-1}}(s)| + |V^{\text{soft}}_{\theta_{t-1}}(s) - V^{\text{soft}}_{\theta_{t-1},\pi_{t-1}}(s)|,$$

$$\overset{(f)}{\leq} C_Q||\theta_t - \theta_{t-1}|| + |V^{\text{soft}}_{\theta_{t-1},\pi_t}(s) - V^{\text{soft}}_{\theta_{t-1}}(s)| + C_Q||\theta_t - \theta_{t-1}||,$$

$$\overset{(g)}{\leq} 2C_Q||\theta_t - \theta_{t-1}|| + V^{\text{soft}}_{\theta_{t-1}}(s) - \mathcal{T}^{\text{soft}}_{\theta_{t-1}}(V^{\text{soft}}_{\theta_{t-1},\pi_t})(s),$$

$$\leq 2C_Q||\theta_t - \theta_{t-1}|| + \gamma|V^{\text{soft}}_{\theta_{t-1}}(s) - V^{\text{soft}}_{\theta_{t-1},\pi_t}(s)|,$$

where $(f)$ follows Claim 2 and claim 5 and $(e)$ follows Claim 4.

Now we take a look at the term $||\theta_t - \theta_{t-1}||$:

$$||\theta_t - \theta_{t-1}|| = \alpha_{t-1}||g_{t-1}||,$$

$$\leq \alpha_{t-1}\left[\left\|\sum_{t=0}^{\infty}\gamma^t\nabla_\theta r_{\theta_{t-1}}(s_t',a_t')\right\| + \left\|\sum_{t=0}^{\infty}\gamma^t\nabla_\theta r_{\theta_{t-1}}(s_t'',a_t'')\right\| + \frac{\lambda(1-\gamma^{t-1})}{1-\gamma}\left\|\theta_{t-1}-\bar\theta\right\|\right],$$

$$\overset{(h)}{\leq}\alpha_{t-1}\left(\frac{\bar C_r}{1-\gamma}+\frac{\bar C_r}{1-\gamma}+\frac{\lambda(1-\gamma^{t-1})}{1-\gamma}\cdot\frac{2\bar C_r}{\lambda}\right),$$

$$\leq\frac{4\alpha_{t-1}\bar C_r}{1-\gamma}, \tag{8}$$

where $(h)$ follows claim 1. Recall that

$$|\log\pi_{t+1}(a|s)-\log\pi_{\theta_t}(a|s)|\leq|Q^{\text{soft}}_{\theta_t,\pi_t}(s,a)-Q^{\text{soft}}_{\theta_t}(s,a)|+|V^{\text{soft}}_{\theta_t,\pi_t}(s)-V^{\text{soft}}_{\theta_t}(s)|.$$

Summing from $i=0$ to $t$, we get

$$\sum_{i=1}^{t}\left[|Q^{\text{soft}}_{\theta_i,\pi_i}(s,a)-Q^{\text{soft}}_{\theta_i}(s,a)|+|V^{\text{soft}}_{\theta_i,\pi_i}(s)-V^{\text{soft}}_{\theta_i}(s)|\right],$$

$$\leq\sum_{i=1}^{t}\left[4C_Q\|\theta_i-\theta_{i-1}\|+\gamma\left(|Q^{\text{soft}}_{\theta_{i-1}}(s,a)-Q^{\text{soft}}_{\theta_{i-1},\pi_{i-1}}(s,a)|+|V^{\text{soft}}_{\theta_{i-1},\pi_{i-1}}(s)-V^{\text{soft}}_{\theta_{i-1}}(s)|\right)\right],$$

$$\Rightarrow(1-\gamma)\sum_{i=0}^{t-1}\left[|Q^{\text{soft}}_{\theta_i,\pi_i}(s,a)-Q^{\text{soft}}_{\theta_i}(s,a)|+|V^{\text{soft}}_{\theta_i,\pi_i}(s)-V^{\text{soft}}_{\theta_i}(s)|\right],$$

$$\overset{(i)}{\leq}\frac{16C_Q\bar C_r}{1-\gamma}\sum_{i=1}^{t}\alpha_{i-1}+\left(|Q^{\text{soft}}_{\theta_0}(s,a)-Q^{\text{soft}}_{\theta_0,\pi_0}(s,a)|+|V^{\text{soft}}_{\theta_0,\pi_0}(s)-V^{\text{soft}}_{\theta_0}(s)|\right.$$

$$\left.-|Q^{\text{soft}}_{\theta_t}(s,a)-Q^{\text{soft}}_{\theta_t,\pi_t}(s,a)|-|V^{\text{soft}}_{\theta_t,\pi_t}(s)-V^{\text{soft}}_{\theta_t}(s)|\right),$$

$$\Rightarrow\frac{1}{t}\sum_{i=0}^{t-1}\left[|Q^{\text{soft}}_{\theta_i,\pi_i}(s,a)-Q^{\text{soft}}_{\theta_i}(s,a)|+|V^{\text{soft}}_{\theta_i,\pi_i}(s)-V^{\text{soft}}_{\theta_i}(s)|\right]$$

$$\leq\frac{16C_Q\bar C_r}{t(1-\gamma)^2}\sum_{i=1}^{t}\alpha_{i-1}+\frac{1}{t(1-\gamma)}\left(|Q^{\text{soft}}_{\theta_0}(s,a)-Q^{\text{soft}}_{\theta_0,\pi_0}(s,a)|+|V^{\text{soft}}_{\theta_0,\pi_0}(s)-V^{\text{soft}}_{\theta_0}(s)|\right.$$

$$\left.-|Q^{\text{soft}}_{\theta_t}(s,a)-Q^{\text{soft}}_{\theta_t,\pi_t}(s,a)|-|V^{\text{soft}}_{\theta_t,\pi_t}(s)-V^{\text{soft}}_{\theta_t}(s)|\right),$$

$$=\frac{\bar D_1}{\sqrt t}+\frac{\bar D_2}{t},$$

where $(i)$ follows (8), $\bar D_1=\frac{16C_Q\bar C_r}{(1-\gamma)^2}$, and $\bar D_2=\frac{1}{1-\gamma}\left(|Q^{\text{soft}}_{\theta_0}(s,a)-Q^{\text{soft}}_{\theta_0,\pi_0}(s,a)|+|V^{\text{soft}}_{\theta_0,\pi_0}(s)-$

$V^{\text{soft}}_{\theta_0}(s)|-|Q^{\text{soft}}_{\theta_t}(s,a)-Q^{\text{soft}}_{\theta_t,\pi_t}(s,a)|-|V^{\text{soft}}_{\theta_t,\pi_t}(s)-V^{\text{soft}}_{\theta_t}(s)|\right)$. Therefore, we can see that

$$\frac{1}{t}\sum_{i=0}^{t-1}|\log\pi_{i+1}(a|s)-\log\pi_{\theta_i}(a|s)|\leq\frac{\bar D_1}{\sqrt t}+\frac{\bar D_2}{t}. \tag{9}$$

Define the loss function $\bar L(\theta)\triangleq E_{(S,A)\sim\mu^{\pi_E}(\cdot,\cdot)}\left[-\log\pi_\theta(A|S)+\frac{\lambda}{2}\|\theta-\bar\theta\|^2\right]$, then we can see that $E_{(S_i^E,A_i^E)\sim\mu^{\pi_E}(\cdot,\cdot)}[L_i(\theta;(S_i^E,A_i^E))]=\gamma^i\bar L(\theta)$. Moreover, we have that

$$\|\nabla L_i(\theta_t;(S_i^E,A_i^E))\|\overset{(j)}{\leq}\gamma^i\lambda\|\theta_t-\bar\theta\|$$

$$+\gamma^i\|E^{\pi_{\theta_t}}_{S,A}[\sum_{k=0}^{\infty}\gamma^k\nabla_\theta r_{\theta_t}(S_k,A_k)|S_0=S_i^E]-E^{\pi_{\theta_t}}_{S,A}[\sum_{k=0}^{\infty}\gamma^k\nabla_\theta r_{\theta_t}(S_k,A_k)|S_0=S_i^E,A_0=A_i^E]\|,$$

$$\overset{(k)}{\leq}2\gamma^i\bar C_r+\frac{2\gamma^i\bar C_r}{1-\gamma}, \tag{10}$$

where $(j)$ follows Lemma 3 and $(k)$ follows Claim 1.

Therefore, we have that

$$E_{(S_i^E, A_i^E) \sim \mu^{\pi_E}(\cdot, \cdot)}[||\frac{\nabla L_i(\theta_t; (S_i^E, A_i^E))}{\gamma^i} - \nabla \bar{L}(\theta_t)||^2] \overset{(l')}{\leq} \left[ 2\bar{C}_r + \frac{2\bar{C}_r}{1-\gamma} \right]^2,$$

$$\Rightarrow E_{(S_i^E, A_i^E) \sim \mu^{\pi_E}(\cdot, \cdot)}[||\frac{1}{t} \sum_{i=0}^{t-1} L_i(\theta_t; (S_i^E, A_i^E)) - \gamma^i \bar{L}(\theta_t)||^2],$$

$$= \left( \frac{1-\gamma^t}{1-\gamma} \right)^2 E_{(S_i^E, A_i^E) \sim \mu^{\pi_E}(\cdot, \cdot)}[||\frac{1}{t} \sum_{i=0}^{t-1} \frac{L_i(\theta_t; (S_i^E, A_i^E))}{\gamma^i} - \bar{L}(\theta_t)||^2],$$

$$\leq \frac{1}{t} \cdot \frac{4\bar{C}_r^2 (2-\gamma)^2}{(1-\gamma)^4}, \tag{11}$$

where $(l')$ follows the fact that a bounded variable $X \in [-a, a]$ has bounded variance at most $a^2$.
Now we take a look at the term $g_t - \frac{1-\gamma^{t+1}}{1-\gamma} \nabla \bar{L}(\theta_t)$:

$$E_{(S,A) \sim \mu^{\pi_E}(\cdot, \cdot)} \left[ g_t - \frac{1-\gamma^{t+1}}{1-\gamma} \nabla \bar{L}(\theta_t) \right],$$

$$= E_{\{(S_i^E, A_i^E) \sim \mu^{\pi_E}(\cdot, \cdot)\}_{i \geq 0}} \left[ g_t - \sum_{i=0}^{t} \nabla L_i(\theta_t; (S_i^E, A_i^E)) \right]$$

$$+ E_{\{(S_i^E, A_i^E) \sim \mu^{\pi_E}(\cdot, \cdot)\}_{i \geq 0}} \left[ \sum_{i=0}^{t} \nabla L_i(\theta_t; (S_i^E, A_i^E)) - \frac{1-\gamma^{t+1}}{1-\gamma} \nabla \bar{L}(\theta_t) \right],$$

$$= E_{\{(S_i^E, A_i^E) \sim \mu^{\pi_E}(\cdot, \cdot)\}_{i \geq 0}} \left[ g_t - \sum_{i=0}^{t} \nabla L_i(\theta_t; (S_i^E, A_i^E)) \right],$$

$$= E_{\{(S_i^E, A_i^E) \sim \mu^{\pi_E}(\cdot, \cdot)\}_{i \geq 0}} \left[ \sum_{i=0}^{\infty} \gamma^i \nabla_\theta r_{\theta_t}(s_i', a_i') - E_{S,A}^{\pi_{\theta_t}}[\sum_{i=0}^{\infty} \gamma^i \nabla_\theta r_{\theta_t}(S_i, A_i)|S_0 = S_0^E] \right]$$

$$+ E_{\{(S_i^E, A_i^E) \sim \mu^{\pi_E}(\cdot, \cdot)\}_{i \geq 0}} \left[ \sum_{i=t+1}^{\infty} \gamma^i \nabla_\theta r_{\theta_t}(s_i'', a_i'') \right.$$

$$\left. - E_{S,A}^{\pi_{\theta_t}}[\sum_{i=t+1}^{\infty} \gamma^i \nabla_\theta r_{\theta_t}(S_i, A_i)|S_t = S_t^E, A_t = A_t^E] \right],$$

$$= E_{\{(S_i^E, A_i^E) \sim \mu^{\pi_E}(\cdot, \cdot)\}_{i \geq 0}} \left[ E_{S,A}^{\pi_{t+1}}[\sum_{i=0}^{\infty} \gamma^i \nabla_\theta r_{\theta_t}(S_i, A_i)|S_0 = S_0^E] \right.$$

$$- E_{S,A}^{\pi_{\theta_t}}[\sum_{i=0}^{\infty} \gamma^i \nabla_\theta r_{\theta_t}(S_i, A_i)|S_0 = S_0^E] + E_{S,A}^{\pi_{t+1}}[\sum_{i=t+1}^{\infty} \gamma^i \nabla_\theta r_{\theta_t}(S_i, A_i)|S_t = S_t^E, A_t = A_t^E]$$

$$\left. - E_{S,A}^{\pi_{\theta_t}}[\sum_{i=t+1}^{\infty} \gamma^i \nabla_\theta r_{\theta_t}(S_i, A_i)|S_t = S_t^E, A_t = A_t^E] \right].$$

From equation (64) in [5], we know that

$$\left\| E_{S,A}^{\pi_{t+1}}[\sum_{i=0}^{\infty} \gamma^i \nabla_\theta r_{\theta_t}(S_i, A_i)|S_0 = s_0] - E_{S,A}^{\pi_{\theta_t}}[\sum_{i=0}^{\infty} \gamma^i \nabla_\theta r_{\theta_t}(S_i, A_i)|S_0 = s_0] \right\|,$$

$$\leq \frac{2\bar{C}_r}{1-\gamma} \int_{s \in \mathcal{S}} \int_{a \in \mathcal{A}} |Q_{\theta_t}^{\text{soft}}(s,a) - Q_{\theta_t, \pi_t}^{\text{soft}}(s,a)| da ds,$$

$$\leq \frac{2\bar{C}_r C_d}{1-\gamma} \sup_{(s,a) \in \mathcal{S} \times \mathcal{A}} \{|Q_{\theta_t}^{\text{soft}}(s,a) - Q_{\theta_t, \pi_t}^{\text{soft}}(s,a)|\},$$

where $C_d$ is the product of the area of $\mathcal{S}$ and the area of $\mathcal{A}$.

Therefore, we can get that

$$E_{(S,A)\sim\mu^{\pi_E}}\left[\left\|g_t - \frac{1-\gamma^{t+1}}{1-\gamma}\nabla\bar{L}(\theta_t)\right\|\right] \le \frac{4\bar{C}_r C_d}{1-\gamma}\sup_{(s,a)\in\mathcal{S}\times\mathcal{A}}\{|Q_{\theta_t}^{\text{soft}}(s,a) - Q_{\theta_t,\pi_t}^{\text{soft}}(s,a)|\}. \quad (12)$$

From (7) and (8), we know that

$$|Q_{\theta_i,\pi_i}^{\text{soft}}(s,a) - Q_{\theta_i}^{\text{soft}}(s,a)| \le \gamma|Q_{\theta_{i-1},\pi_{i-1}}^{\text{soft}}(s,a) - Q_{\theta_{i-1}}^{\text{soft}}(s,a)| + \frac{8\alpha_{i-1}C_Q\bar{C}_r}{1-\gamma},$$

$$\Rightarrow \alpha_i|Q_{\theta_i,\pi_i}^{\text{soft}}(s,a) - Q_{\theta_i}^{\text{soft}}(s,a)| \le \alpha_{i-1}\gamma|Q_{\theta_{i-1},\pi_{i-1}}^{\text{soft}}(s,a) - Q_{\theta_{i-1}}^{\text{soft}}(s,a)| + \frac{8\alpha_{i-1}^2 C_Q\bar{C}_r}{1-\gamma},$$

$$\Rightarrow \sum_{i=1}^{t}\alpha_i|Q_{\theta_i,\pi_i}^{\text{soft}}(s,a) - Q_{\theta_i}^{\text{soft}}(s,a)| \le \sum_{i=0}^{t-1}\alpha_i\gamma|Q_{\theta_i,\pi_i}^{\text{soft}}(s,a) - Q_{\theta_i}^{\text{soft}}(s,a)| + \sum_{i=0}^{t-1}\frac{8\alpha_i^2 C_Q\bar{C}_r}{1-\gamma},$$

$$\Rightarrow (1-\gamma)\sum_{i=0}^{t-1}\alpha_i|Q_{\theta_i,\pi_i}^{\text{soft}}(s,a) - Q_{\theta_i}^{\text{soft}}(s,a)|,$$

$$\le \alpha_0|Q_{\theta_0,\pi_0}^{\text{soft}}(s,a) - Q_{\theta_0}^{\text{soft}}(s,a)| - \alpha_t|Q_{\theta_t,\pi_t}^{\text{soft}}(s,a) - Q_{\theta_t}^{\text{soft}}(s,a)| + \sum_{i=0}^{t-1}\frac{8\alpha_i^2 C_Q\bar{C}_r}{1-\gamma}, \quad (13)$$

and similarly we can see that

$$\sum_{i=1}^{t}|Q_{\theta_i,\pi_i}^{\text{soft}}(s,a) - Q_{\theta_i}^{\text{soft}}(s,a)| \le \sum_{i=0}^{t-1}\gamma|Q_{\theta_i,\pi_i}^{\text{soft}}(s,a) - Q_{\theta_i}^{\text{soft}}(s,a)| + \frac{8\alpha_i C_Q\bar{C}_r}{1-\gamma},$$

$$\Rightarrow (1-\gamma)\sum_{i=1}^{t-1}|Q_{\theta_i,\pi_i}^{\text{soft}}(s,a) - Q_{\theta_i}^{\text{soft}}(s,a)|,$$

$$\le |Q_{\theta_0,\pi_0}^{\text{soft}}(s,a) - Q_{\theta_0}^{\text{soft}}(s,a)| - |Q_{\theta_t,\pi_t}^{\text{soft}}(s,a) - Q_{\theta_t}^{\text{soft}}(s,a)| + \sum_{i=0}^{t-1}\frac{8\alpha_i C_Q\bar{C}_r}{1-\gamma}, \quad (14)$$

Telescoping from $i = 0$ to $t - 1$, we get

$$\sum_{i=0}^{t-1}\alpha_i E_{(S,A)\sim\mu^{\pi_E}}\left[\left\|g_i - \frac{1-\gamma^{i+1}}{1-\gamma}\nabla\bar{L}(\theta_i)\right\|\right],$$

$$\overset{(l)}{\le} \frac{4\bar{C}_r C_d}{1-\gamma}\sum_{i=0}^{t-1}\alpha_i|Q_{\theta_i,\pi_i}^{\text{soft}}(S,A) - Q_{\theta_i}^{\text{soft}}(S,A)|,$$

$$\overset{(m)}{\le} \frac{4\bar{C}_r C_d}{(1-\gamma)^2}\left[\alpha_0|Q_{\theta_0,\pi_0}^{\text{soft}}(S,A) - Q_{\theta_0}^{\text{soft}}(S,A)| - \alpha_t|Q_{\theta_t,\pi_t}^{\text{soft}}(S,A) - Q_{\theta_t}^{\text{soft}}(S,A)| + \sum_{i=0}^{t-1}\frac{8\alpha_i^2 C_Q\bar{C}_r}{1-\gamma}\right], \quad (15)$$

where $(l)$ follows (12) and $(m)$ follows (13). Similarly, we can see that

$$\sum_{i=0}^{t-1}E_{(S,A)\sim\mu^{\pi_E}}\left[\left\|g_i - \frac{1-\gamma^{i+1}}{1-\gamma}\nabla\bar{L}(\theta_i)\right\|\right],$$

$$\le \frac{4\bar{C}_r C_d}{1-\gamma}\sum_{i=0}^{t-1}|Q_{\theta_i,\pi_i}^{\text{soft}}(S,A) - Q_{\theta_i}^{\text{soft}}(S,A)|,$$

$$\le \frac{4\bar{C}_r C_d}{(1-\gamma)^2}\left[|Q_{\theta_0,\pi_0}^{\text{soft}}(S,A) - Q_{\theta_0}^{\text{soft}}(S,A)| - |Q_{\theta_t,\pi_t}^{\text{soft}}(S,A) - Q_{\theta_t}^{\text{soft}}(S,A)| + \sum_{i=0}^{t-1}\frac{8\alpha_i C_Q\bar{C}_r}{1-\gamma}\right], \quad (16)$$

Now, we start to quantify the local regret:

$$\bar{L}(\theta_{i+1}) \geq \bar{L}(\theta_i) + [\nabla\bar{L}(\theta_i)]^\top(\theta_{i+1} - \theta_i) - \frac{C_L}{2}||\theta_{i+1} - \theta_i||^2,$$

$$\overset{(n)}{\geq} \bar{L}(\theta_i) + \alpha_t[\nabla\bar{L}(\theta_i)]^\top g_i - \frac{8\alpha_i^2\bar{C}_r^2 C_L}{(1-\gamma)^2},$$

$$= \bar{L}(\theta_i) + \alpha_i\frac{1-\gamma^{i+1}}{1-\gamma}||\nabla\bar{L}(\theta_i)||^2 + \alpha_i[\nabla\bar{L}(\theta_i)]^\top(g_i - \frac{1-\gamma^{i+1}}{1-\gamma}\nabla\bar{L}(\theta_i)) - \frac{8\alpha_i^2\bar{C}_r^2 C_L}{(1-\gamma)^2},$$

$$\geq \bar{L}(\theta_i) + \alpha_i\frac{1-\gamma^{i+1}}{1-\gamma}||\nabla\bar{L}(\theta_i)||^2 - \alpha_i||\nabla\bar{L}(\theta_i)|| \cdot ||g_i - \frac{1-\gamma^{i+1}}{1-\gamma}\nabla\bar{L}(\theta_i)|| - \frac{8\alpha_i^2\bar{C}_r^2 C_L}{(1-\gamma)^2},$$

$$\overset{(o)}{\geq} \bar{L}(\theta_i) + \alpha_i\frac{1-\gamma^{i+1}}{1-\gamma}||\nabla\bar{L}(\theta_i)||^2 - \alpha_i\frac{2\bar{C}_r(2-\gamma)}{1-\gamma} \cdot ||g_i - \frac{1-\gamma^{i+1}}{1-\gamma}\nabla\bar{L}(\theta_i)|| - \frac{8\alpha_i^2\bar{C}_r^2 C_L}{(1-\gamma)^2},$$

$$\Rightarrow E_{\{(S_i^E, A_i^E)\sim\mu^{\pi_E}(\cdot,\cdot)\}_{i\geq 0}}\left[\sum_{i=0}^{t-1}\alpha_i\frac{1-\gamma^{i+1}}{1-\gamma}||\nabla\bar{L}(\theta_i)||^2\right],$$

$$\leq \bar{L}(\theta_t) - \bar{L}(\theta_0) + \frac{2\bar{C}_r(2-\gamma)}{1-\gamma}\sum_{i=0}^{t-1}\alpha_i E_{\{(S_i^E, A_i^E)\sim\mu^{\pi_E}(\cdot,\cdot)\}_{i\geq 0}}\left[||g_i - \frac{1-\gamma^{i+1}}{1-\gamma}\nabla\bar{L}(\theta_i)||\right]$$

$$+ \sum_{i=0}^{t-1}\frac{8\alpha_i^2\bar{C}_r^2 C_L}{(1-\gamma)^2},$$

$$\overset{(p)}{\leq} \bar{L}(\theta_t) - \bar{L}(\theta_0) + \frac{8\bar{C}_r^2 C_d(2-\gamma)}{(1-\gamma)^3}E_{\{(S_i^E, A_i^E)\sim\mu^{\pi_E}(\cdot,\cdot)\}_{i\geq 0}}\left[\alpha_0|Q_{\theta_0,\pi_0}^{\text{soft}}(S_0^E, A_0^E) - Q_{\theta_0}^{\text{soft}}(S_0^E, A_0^E)|\right.$$

$$\left. - \alpha_t|Q_{\theta_t,\pi_t}^{\text{soft}}(S_t^E, A_t^E) - Q_{\theta_t}^{\text{soft}}(S_t^E, A_t^E)|\right] + \left(\frac{32C_Q\bar{C}_r^2 C_d}{(1-\gamma)^3} + \frac{8\bar{C}_r^2 C_L}{(1-\gamma)^2}\right)\sum_{i=0}^{t-1}\alpha_i^2,$$

where $(n)$ follows (8), $(o)$ follows (10), $(p)$ follows (15). Therefore, we have that

$$E_{\{(S_i^E, A_i^E)\sim\mu^{\pi_E}(\cdot,\cdot)\}_{i\geq 0}}\left[\sum_{t=0}^{T-1}\alpha_{T-1}||\nabla\bar{L}(\theta_t)||^2\right], \tag{17}$$

$$\leq E_{\{(S_i^E, A_i^E)\sim\mu^{\pi_E}(\cdot,\cdot)\}_{i\geq 0}}\left[\sum_{t=0}^{T-1}\alpha_t\frac{1-\gamma^t}{1-\gamma}||\nabla\bar{L}(\theta_t)||^2\right],$$

$$\leq \bar{L}(\theta_T) - \bar{L}(\theta_0) + \frac{8\bar{C}_r^2 C_d(2-\gamma)}{(1-\gamma)^3}E_{\{(S_i^E, A_i^E)\sim\mu^{\pi_E}(\cdot,\cdot)\}_{i\geq 0}}\left[\alpha_0|Q_{\theta_0,\pi_0}^{\text{soft}}(S_0^E, A_0^E) - Q_{\theta_0}^{\text{soft}}(S_0^E, A_0^E)|\right.$$

$$\left. - \alpha_T|Q_{\theta_T,\pi_T}^{\text{soft}}(S_T^E, A_T^E) - Q_{\theta_T}^{\text{soft}}(S_T^E, A_T^E)|\right] + \left(\frac{32C_Q\bar{C}_r^2 C_d}{(1-\gamma)^3} + \frac{8\bar{C}_r^2 C_L}{(1-\gamma)^2}\right)\sum_{i=0}^{T-1}\alpha_i^2,$$

$$\Rightarrow E_{\{(S_i^E, A_i^E)\sim\mu^{\pi_E}(\cdot,\cdot)\}_{i\geq 0}}\left[\sum_{t=0}^{T-1}||\nabla\bar{L}(\theta_t)||^2\right] \leq D_2\sqrt{T} + D_3\sqrt{T}(\log T + 1), \tag{18}$$

where $D_2 = \bar{L}(\theta_T) - \bar{L}(\theta_0) + \frac{8\bar{C}_r^2 C_d(2-\gamma)}{(1-\gamma)^3}E_{\{(S_i^E, A_i^E)\sim\mu^{\pi_E}(\cdot,\cdot)\}_{i\geq 0}}\left[\alpha_0|Q_{\theta_0,\pi_0}^{\text{soft}}(S_0^E, A_0^E) - Q_{\theta_0}^{\text{soft}}(S_0^E, A_0^E)| - \alpha_T|Q_{\theta_T,\pi_T}^{\text{soft}}(S_T^E, A_T^E) - Q_{\theta_T}^{\text{soft}}(S_T^E, A_T^E)|\right]$ and $D_3 = \frac{2(1-\gamma)}{\lambda}\left(\frac{32C_Q\bar{C}_r^2 C_d}{(1-\gamma)^3} + \frac{8\bar{C}_r^2 C_L}{(1-\gamma)^2}\right)$.

Then we can see that

$$E_{\{(S_i^E, A_i^E)\sim\mu^{\pi_E}(\cdot,\cdot)\}_{i\geq 0}}\left[\sum_{t=0}^{T-1}||\frac{1}{t+1}\sum_{i=0}^{t}\nabla L_i(\theta_t; (S_i^E, A_i^E))||^2\right],$$

$$\leq 2E_{\{(S_i^E, A_i^E)\sim\mu^{\pi_E}(\cdot,\cdot)\}_{i\geq 0}}\left[\sum_{t=0}^{T-1}||\frac{1}{t+1}\sum_{i=0}^{t}\nabla L_i(\theta_t; (S_i^E, A_i^E)) - \gamma^i\nabla\bar{L}(\theta_t)||^2\right]$$

$$+ 2E_{\{(S_i^E, A_i^E) \sim \mu^{\pi E}(\cdot, \cdot)\}_{i \geq 0}} \left[ \sum_{t=0}^{T-1} \left\| \frac{1 - \gamma^t}{(t+1)(1-\gamma)} \nabla \bar{L}(\theta_t) \right\|^2 \right],$$

$$\leq 2E_{\{(S_i^E, A_i^E) \sim \mu^{\pi E}(\cdot, \cdot)\}_{i \geq 0}} \left[ \sum_{t=0}^{T-1} \left\| \frac{1}{t+1} \sum_{i=0}^{t} \nabla L_i(\theta_t; (S_i^E, A_i^E)) - \gamma^i \nabla \bar{L}(\theta_t) \right\|^2 \right]$$

$$+ 2E_{\{(S_i^E, A_i^E) \sim \mu^{\pi E}(\cdot, \cdot)\}_{i \geq 0}} \left[ \sum_{t=0}^{T-1} \| \nabla \bar{L}(\theta_t) \|^2 \right],$$

$$\overset{(q)}{\leq} \sum_{t=0}^{T-1} \frac{8 \bar{C}_r^2 (2-\gamma)^2}{(t+1)(1-\gamma)^4} + 2E_{\{(S_i^E, A_i^E) \sim \mu^{\pi E}(\cdot, \cdot)\}_{i \geq 0}} \left[ \sum_{t=0}^{T-1} \| \nabla \bar{L}(\theta_t) \|^2 \right],$$

$$\leq D_1 (\log T + 1) + D_2 \sqrt{T} + D_3 \sqrt{T} (\log T + 1),$$

where $D_1 = \frac{8 \bar{C}_r^2 (2-\gamma)^2}{(1-\gamma)^4}$. The $(q)$ follows (11). Note that to achieve this local regret rate, we actually need to take an extra expectation over the dynamics $P$ because we need to roll out $\pi_t$ to formulate $g_t$. Here, we omit the expectation over the dynamics.

## B.5 Proof of Theorem 1

We know that

$$E_{\{(S_i^E, A_i^E) \sim \mu^{\pi E}(\cdot, \cdot)\}_{i \geq 0}} \left[ \sum_{i=0}^{t} \| \nabla L_i(\theta_t; (S_i^E, A_i^E)) \|^2 \right],$$

$$\leq E_{\{(S_i^E, A_i^E) \sim \mu^{\pi E}(\cdot, \cdot)\}_{i \geq 0}} \left[ \sum_{i=0}^{t} \| \gamma^i \nabla \bar{L}(\theta_t) \|^2 \right]$$

$$+ E_{\{(S_i^E, A_i^E) \sim \mu^{\pi E}(\cdot, \cdot)\}_{i \geq 0}} \left[ \sum_{i=0}^{t} \| \nabla L_i(\theta_t; (S_i^E, A_i^E)) - \gamma^i \nabla \bar{L}(\theta_t) \|^2 \right],$$

$$\leq E_{\{(S_i^E, A_i^E) \sim \mu^{\pi E}(\cdot, \cdot)\}_{i \geq 0}} \left[ \sum_{i=0}^{t} \| \nabla \bar{L}(\theta_t) \|^2 \right]$$

$$+ E_{\{(S_i^E, A_i^E) \sim \mu^{\pi E}(\cdot, \cdot)\}_{i \geq 0}} \left[ \sum_{i=0}^{t} \| \nabla L_i(\theta_t; (S_i^E, A_i^E)) - \gamma^i \nabla \bar{L}(\theta_t) \|^2 \right],$$

$$\overset{(a)}{\leq} O(\sqrt{t+1} + \sqrt{t+1} \log(t+1)) + \sum_{i=0}^{t} \gamma^{2i} \cdot 4 \bar{C}_r^2 \left( \frac{2-\gamma}{1-\gamma} \right)^2,$$

$$\Rightarrow E_{\{(S_i^E, A_i^E) \sim \mu^{\pi E}(\cdot, \cdot)\}_{i \geq 0}} \left[ \frac{1}{t+1} \sum_{i=0}^{t} \| \nabla L_i(\theta_t; (S_i^E, A_i^E)) \|^2 \right] \leq O \left( \frac{1}{\sqrt{t+1}} + \frac{\log(t+1)}{\sqrt{t+1}} + \frac{1}{t+1} \right),$$
(19)

where $(a)$ follows (18) and (11).

Therefore,

$$\sum_{t=0}^{T-1} E_{\{(S_i^E, A_i^E) \sim \mathbb{P}_i^{\pi E}(\cdot, \cdot)\}_{i \geq 0}} \left[ \left\| \frac{1}{t+1} \sum_{i=0}^{t} \nabla L_i(\theta_t; (S_i^E, A_i^E)) \right\|^2 \right],$$

$$\leq \sum_{t=0}^{T-1} E_{\{(S_i^E, A_i^E) \sim \mathbb{P}_i^{\pi E}(\cdot, \cdot)\}_{i \geq 0}} \left[ \frac{1}{t+1} \sum_{i=0}^{t} \| \nabla L_i(\theta_t; (S_i^E, A_i^E)) \|^2 \right],$$

$$\leq \sum_{t=0}^{T-1} E_{\{(S_i^E, A_i^E) \sim \mu^{\pi E}(\cdot, \cdot)\}_{i \geq 0}} \left[ \frac{1}{t+1} \sum_{i=0}^{t} \| \nabla L_i(\theta_t; (S_i^E, A_i^E)) \|^2 \right]$$

$$+ \sum_{t=0}^{T-1} \frac{1}{t+1} \sum_{i=0}^{t} \Big[ E_{\{(S_i^E, A_i^E) \sim \mathbb{P}_i^{\pi_E}(\cdot, \cdot)\}_{i \geq 0}} [||\nabla L_i(\theta_t; (S_i^E, A_i^E))||^2}$$

$$- E_{\{(S_i^E, A_i^E) \sim \mu^{\pi_E}(\cdot, \cdot)\}_{i \geq 0}} [||\nabla L_i(\theta_t; (S_i^E, A_i^E))||^2] \Big],$$

$$\overset{(s)}{\leq} D_1 \log T + D_2 \sqrt{T} + D_3 \log T \sqrt{T} + \sum_{t=0}^{T-1} \frac{1}{t+1} \sum_{i=0}^{t} 8 C_M \bar{C}_r^2 \left( \frac{2-\gamma}{1-\gamma} \right)^2 \rho^i \gamma^{2i},$$

$$\leq D_1 \log T + D_2 \sqrt{T} + D_3 \log T \sqrt{T} + \sum_{t=0}^{T-1} \frac{1}{t+1} \cdot 8 C_M \bar{C}_r^2 \left( \frac{2-\gamma}{1-\gamma} \right)^2 \frac{1}{1 - \rho \gamma^2},$$

$$\leq \left( D_1 + \frac{8 C_M \bar{C}_r^2 (2-\gamma)^2}{(1 - \rho \gamma^2)(1-\gamma)^2} \right) \log T + D_2 \sqrt{T} + D_3 \log T \sqrt{T},$$

where $(s)$ follows (19) and Proposition 1. Note that to achieve this local regret rate, we actually need to take an extra expectation over the dynamics $P$ because we need to roll out $\pi_t$ to formulate $g_t$. Here, we omit the expectation over the dynamics.

## B.6 Proof of Theorem 2

Suppose the expert reward function $r_E$ and the parameterized reward function $r_\theta$ are both linear, i.e., $r_E = \theta_E^\top \phi$ and $r_\theta = \theta^\top \phi$, where $\phi : \mathcal{S} \times \mathcal{A} \to \mathbb{R}_+^n$ is an $n$-dimensional feature vector such that $||\phi(s,a)|| \leq \bar{C}_r$ for any $(s,a) \in \mathcal{S} \times \mathcal{A}$. The proof follows the similar idea with that of Theorem 1 where follow two-step process: step (i) quantifies the regret under the stationary distribution $\mu^{\pi_E}(\cdot, \cdot)$ and step (ii) quantifies the difference between the correlated distribution $\mathbb{P}_t^{\pi_E}$ and the stationary distribution $\mu^{\pi_E}$. We start our proof with the following claim:

**Claim 6.** *If the parameterized reward function $r_\theta$ is linear, the function $\bar{L}(\theta)$ is $\lambda$-strongly convex for any $\theta$.*

*Proof.* Recall that $\bar{L}(\theta) = E_{(\bar{S}, \bar{A}) \sim \mu^{\pi_E}(\cdot, \cdot)}[-\log \pi_\theta(\bar{A}|\bar{S})] + \frac{\lambda}{2}||\theta - \bar{\theta}||^2$. Define $\bar{L}(\theta; (S, A)) \triangleq -\log \pi_\theta(\bar{A}|\bar{S}) + \frac{\lambda}{2}||\theta - \bar{\theta}||^2$, from Lemma 3, we can see that

$$\nabla \bar{L}(\theta; (\bar{S}, \bar{A})) = E_{S,A}^{\pi_\theta} \left[ \sum_{i=0}^{\infty} \gamma^i \phi(S_i, A_i) \Big| S_0 = \bar{S} \right] - E_{S,A}^{\pi_\theta} \left[ \sum_{i=0}^{\infty} \gamma^i \phi(S_i, A_i) \Big| S_0 = \bar{S}, A_0 = \bar{A} \right] + \lambda(\theta - \bar{\theta}).$$

Therefore, we have that

$$\nabla_{\theta\theta}^2 \bar{L}(\theta; (\bar{S}, \bar{A})),$$

$$= \nabla_\theta E_{S,A}^{\pi_\theta} \left[ \sum_{i=0}^{\infty} \gamma^i \phi(S_i, A_i) \Big| S_0 = \bar{S} \right] - \nabla_\theta E_{S,A}^{\pi_\theta} \left[ \sum_{i=0}^{\infty} \gamma^i \phi(S_i, A_i) \Big| S_0 = \bar{S}, A_0 = \bar{A} \right] + \lambda.$$

Now, we take a look at

$$\nabla_\theta E_{S,A}^{\pi_\theta} \left[ \sum_{i=0}^{\infty} \gamma^i \phi(S_i, A_i) \Big| S_0 = \bar{S}, A_0 = \bar{A} \right] = \nabla_\theta \Big\{ \phi(\bar{S}, \bar{A})$$

$$+ \int_{s_1 \in \mathcal{S}} P(s_1|\bar{S}, \bar{A}) \int_{a_{t+1} \in \mathcal{A}} \pi_\theta(a_1|s_1) E_{S,A}^{\pi_\theta} \left[ \sum_{i=1}^{\infty} \gamma^i \phi(S_i, A_i) \Big| S_1 = s_1, A_1 = a_1 \right] da_1 ds_1 \Big\},$$

$$= \int_{s_1 \in \mathcal{S}} P(s_1|\bar{S}, \bar{A}) \int_{a_1 \in \mathcal{A}} \Big\{ \nabla_\theta \pi_\theta(a_1|s_1) \cdot E_{S,A}^{\pi_\theta} \left[ \sum_{i=1}^{\infty} \gamma^i \phi(S_i, A_i) \Big| S_1 = s_1, A_1 = a_1 \right]$$

$$+ \pi_\theta(a_1|s_1) \cdot \nabla_\theta E_{S,A}^{\pi_\theta} \left[ \sum_{i=1}^{\infty} \gamma^i \phi(S_i, A_i) \Big| S_1 = s_1, A_1 = a_1 \right] \Big\} da_1 ds_1.$$

Keep the expansion, we can see that

$$\nabla_\theta E_{S,A}^{\pi_\theta} \left[ \sum_{i=0}^{\infty} \gamma^i \phi(S_i, A_i) \Big| S_0 = \bar{S}, A_0 = \bar{A} \right],$$

$$= E_{S,A}^{\pi_\theta}\left\{\sum_{i=1}^{\infty}\nabla_\theta\pi_\theta(A_i'|S_i')\cdot E_{S,A}^{\pi_\theta}\left[\sum_{i=1}^{\infty}\gamma^i\phi(S_i,A_i)\Big|S_1=S_1',A_1=A_1'\right]\Big|S_0'=\bar{S},A_0'=\bar{A}\right\},$$

$$\nabla_\theta E_{S,A}^{\pi_\theta}\left[\sum_{i=0}^{\infty}\gamma^i\phi(S_i,A_i)\Big|S_0=\bar{S}\right],$$

$$= E_{S,A}^{\pi_\theta}\left\{\sum_{i=0}^{\infty}\nabla_\theta\pi_\theta(A_i'|S_i')\cdot E_{S,A}^{\pi_\theta}\left[\sum_{i=0}^{\infty}\gamma^i\phi(S_i,A_i)\Big|S_0=S_0',A_0=A_0'\right]\Big|S_0'=\bar{S}\right\},$$

Thus we have that

$$E_{(\bar{S},\bar{A})\sim\mu^{\pi_E}(\cdot,\cdot)}\left\{\nabla_\theta E_{S,A}^{\pi_\theta}\left[\sum_{i=0}^{\infty}\gamma^i\phi(S_i,A_i)\Big|S_0=\bar{S}\right]-\nabla_\theta E_{S,A}^{\pi_\theta}\left[\sum_{i=0}^{\infty}\gamma^i\phi(S_i,A_i)\Big|S_0=\bar{S},A_0=\bar{A}\right]\right\},$$

$$= E_{(\bar{S},\bar{A})\sim\mu^{\pi_E}(\cdot,\cdot)}E_{S',A'}^{\pi_\theta}\left\{\nabla_\theta\pi_\theta(A_0'|S_0')\cdot E_{S,A}^{\pi_\theta}\left[\sum_{i=0}^{\infty}\gamma^i\phi(S_i,A_i)\Big|S_0=S_0',A_0=A_0'\right]\Big|S_0'=\bar{S}_0\right\}$$

$$+ E_{(\bar{S},\bar{A})\sim\mu^{\pi_E}(\cdot,\cdot)}\left\{E_{S',A'}^{\pi_\theta}\left\{\sum_{i=1}^{\infty}\nabla_\theta\pi_\theta(A_i'|S_i')\cdot E_{S,A}^{\pi_\theta}\left[\sum_{i=1}^{\infty}\gamma^i\phi(S_i,A_i)\Big|S_1=S_1',A_1=A_1'\right]\Big|S_0'=\bar{S}\right\}\right.$$

$$\left.- E_{S,A}^{\pi_\theta}\left\{\sum_{i=1}^{\infty}\nabla_\theta\pi_\theta(A_i'|S_i')\cdot E_{S,A}^{\pi_\theta}\left[\sum_{i=1}^{\infty}\gamma^i\phi(S_i,A_i)\Big|S_1=S_1',A_1=A_1'\right]\Big|S_0=\bar{S},A_0=\bar{A}\right\}\right\},$$

$$= E_{(\bar{S},\bar{A})\sim\mu^{\pi_E}(\cdot,\cdot)}E_{S',A'}^{\pi_\theta}\left\{\nabla_\theta\pi_\theta(A_0'|S_0')\cdot E_{S,A}^{\pi_\theta}\left[\sum_{i=0}^{\infty}\gamma^i\phi(S_i,A_i)\Big|S_0=S_0',A_0=A_0'\right]\Big|S_0'=\bar{S}_0\right\}$$

$$\overset{(a)}{=} E_{(\bar{S},\bar{A})\sim\mu^{\pi_E}(\cdot,\cdot)}E_{S',A'}^{\pi_\theta}\left\{\pi_\theta(A_0'|S_0^E)[X-EX]X|S_0=\bar{S}\right\},$$

$$= \text{Cov}(X)\succ 0, \tag{20}$$

where $(a)$ follows Lemma 3, $X\triangleq E_{S,A}^{\pi_\theta}[\sum_{i=0}^{\infty}\gamma^i\phi(S_i,A_i)|S_0=\bar{S},A_0=A]$ is a random vector, $EX\triangleq E_{S,A}^{\pi_\theta}[\sum_{i=0}^{\infty}\gamma^i\phi(S_i,A_i)|S_0=\bar{S}]$ is the expectation of $X$ over the action distribution, and $\text{Cov}(X)$ is the covariance matrix of the random vector $X$ with itself, which is always positive definite because the policy $\pi_\theta$ is always stochastic.

Therefore, we can see that

$$||\nabla_{\theta\theta}^2\bar{L}(\theta)||=||\text{Cov}(X)+\lambda||\overset{(b)}{\geq}\lambda,$$

where $(b)$ follows (20). $\qquad\square$

**Step (i)**. We first quantify the regret under the stationary distribution $\mu^{\pi_E}(\cdot,\cdot)$. Suppose $\theta^*\in\arg\min\bar{L}(\theta)=\arg\min\frac{1-\gamma^T}{1-\gamma}\bar{L}(\theta)=\arg\min E_{(S_t^E,A_t^E)\sim\mu^{\pi_E}}[\sum_{t=0}^{T-1}L_t(\theta;(S_t^E,A_t^E))]$.

$$||\theta_{t+1}-\theta^*||^2=||\theta_t-\alpha_t g_t-\theta^*||^2=||\theta_t-\theta^*||^2+\alpha_t^2||g_t||^2-2\alpha_t\langle g_t,\theta_t-\theta^*\rangle,$$

$$\Rightarrow E[||\theta_{t+1}-\theta^*||^2],$$

$$\overset{(c)}{\leq} E[||\theta_t-\theta^*||^2]+\frac{16\alpha_t^2\bar{C}_r^2}{(1-\gamma)^2}-2\alpha_t\langle\frac{1-\gamma^{t+1}}{1-\gamma}\nabla\bar{L}(\theta_t),\theta_t-\theta^*\rangle-2\alpha_t\langle g_t-\frac{1-\gamma^{t+1}}{1-\gamma}\nabla\bar{L}(\theta_t),\theta_t-\theta^*\rangle, \tag{21}$$

where $(c)$ follows (8)

Since $\bar{L}(\theta)$ is $\lambda$-strongly convex, we have that

$$\bar{L}(\theta^*)\geq\bar{L}(\theta_t)+\langle\nabla\bar{L}(\theta_t),\theta^*-\theta_t\rangle+\frac{\lambda}{2}||\theta_t-\theta^*||^2,$$

$$\Rightarrow \bar{L}(\theta_t)-\bar{L}(\theta^*)\leq\langle\nabla\bar{L}(\theta_t),\theta_t-\theta^*\rangle-\frac{\lambda}{2}||\theta_t-\theta^*||^2,$$

$$\overset{(d)}{\leq} \frac{1-\gamma}{2\alpha_t(1-\gamma^{t+1})}\left[||\theta_t - \theta^*||^2 - ||\theta_{t+1} - \theta^*||^2\right] - \frac{1-\gamma}{1-\gamma^{t+1}}\langle g_t - \frac{1-\gamma^{t+1}}{1-\gamma}\nabla\bar{L}(\theta_t), \theta_t - \theta^*\rangle$$

$$+ \frac{8\alpha_t\bar{C}_r^2}{(1-\gamma)(1-\gamma^{t+1})} - \frac{\lambda}{2}||\theta_t - \theta^*||^2,$$

$$\Rightarrow E_{(S_t^E, A_t^E)\sim\mu^{\pi_E}(\cdot,\cdot)}[\bar{L}(\theta_t) - \bar{L}(\theta^*)],$$

$$\leq \frac{1-\gamma-\lambda\alpha_t(1-\gamma^{t+1})}{2\alpha_t(1-\gamma^{t+1})}E_{(S_t^E, A_t^E)\sim\mu^{\pi_E}(\cdot,\cdot)}[||\theta_t - \theta^*||^2] - \frac{1-\gamma}{2\alpha_t(1-\gamma^{t+1})}E_{(S_t^E, A_t^E)\sim\mu^{\pi_E}(\cdot,\cdot)}[||\theta_{t+1} - \theta^*||^2]$$

$$+ \frac{8\alpha_t\bar{C}_r^2}{(1-\gamma)(1-\gamma^{t+1})} + \frac{1-\gamma}{1-\gamma^{t+1}}E_{(S_t^E, A_t^E)\sim\mu^{\pi_E}(\cdot,\cdot)}\left[||g_t - \frac{1-\gamma^{t+1}}{1-\gamma}\nabla\bar{L}(\theta_t)|| \cdot ||\theta_t - \theta^*||\right],$$

$$\overset{(e)}{\leq} \frac{1-\gamma-\lambda\alpha_t(1-\gamma^{t+1})}{2\alpha_t(1-\gamma^{t+1})}E_{(S_t^E, A_t^E)\sim\mu^{\pi_E}(\cdot,\cdot)}[||\theta_t - \theta^*||^2] - \frac{1-\gamma}{2\alpha_t(1-\gamma^{t+1})}E_{(S_t^E, A_t^E)\sim\mu^{\pi_E}(\cdot,\cdot)}[||\theta_{t+1} - \theta^*||^2]$$

$$+ \frac{8\alpha_t\bar{C}_r^2}{(1-\gamma)(1-\gamma^{t+1})} + \frac{\hat{C}(1-\gamma)}{1-\gamma^{t+1}}E_{(S_t^E, A_t^E)\sim\mu^{\pi_E}(\cdot,\cdot)}\left[||g_t - \frac{1-\gamma^{t+1}}{1-\gamma}\nabla\bar{L}(\theta_t)||\right], \tag{22}$$

where $(d)$ follows (21), and $(e)$ follows Claim 1 which shows that the trajectory of $\theta_t$ is bounded and thus there is a positive constant $\hat{C}$ such that $||\theta_t - \theta^*|| \leq \hat{C}$.

Telescoping (22) from $t = 0$ to $t = T - 1$, we can see that

$$E_{(S_t^E, A_t^E)\sim\mu^{\pi_E}(\cdot,\cdot)}\left[\sum_{t=0}^{T-1}L_t(\theta_t; (S_t^E, A_t^E)) - \sum_{t=0}^{T-1}L_t(\theta^*; (S_t^E, A_t^E))\right],$$

$$= E_{(S_t^E, A_t^E)\sim\mu^{\pi_E}(\cdot,\cdot)}\left[\sum_{t=0}^{T-1}\gamma^t\bar{L}(\theta_t) - \sum_{t=0}^{T-1}\gamma^t\bar{L}(\theta^*)\right],$$

$$\leq E_{(S_t^E, A_t^E)\sim\mu^{\pi_E}(\cdot,\cdot)}\left[\sum_{t=0}^{T-1}\bar{L}(\theta_t) - \sum_{t=0}^{T-1}\bar{L}(\theta^*)\right],$$

$$\overset{(f)}{\leq} \frac{1-\gamma-\lambda\alpha_0}{2\alpha_0}E_{(S_t^E, A_t^E)\sim\mu^{\pi_E}(\cdot,\cdot)}[||\theta_t - \theta^*||^2] - \frac{1-\gamma}{2\alpha_{T-1}(1-\gamma^T)}E_{(S_t^E, A_t^E)\sim\mu^{\pi_E}(\cdot,\cdot)}[||\theta_T - \theta^*||^2]$$

$$+ \frac{4\bar{C}_rC_d\hat{C}}{(1-\gamma)^2}\left[|Q_{\theta_0,\pi_0}^{\text{soft}}(S,A) - Q_{\theta_0}^{\text{soft}}(S,A)| - |Q_{\theta_T,\pi_T}^{\text{soft}}(S,A) - Q_{\theta_T}^{\text{soft}}(S,A)| + \sum_{i=0}^{t-1}\frac{8\alpha_iC_Q\bar{C}_r}{1-\gamma}\right],$$

$$\leq \bar{D}_4 + \frac{32C_QC_d\hat{C}\bar{C}_r^2}{\lambda(1-\gamma)^3}(\log T + 1), \tag{23}$$

where $(f)$ follows (16), and $\bar{D}_4 = \frac{1-\gamma-\lambda\alpha_0}{2\alpha_0}E_{(S_t^E, A_t^E)\sim\mu^{\pi_E}(\cdot,\cdot)}[||\theta_t - \theta^*||^2] - \frac{1-\gamma}{2\alpha_{T-1}(1-\gamma^T)}E_{(S_t^E, A_t^E)\sim\mu^{\pi_E}(\cdot,\cdot)}[||\theta_T - \theta^*||^2] + \frac{4\bar{C}_rC_d\hat{C}}{(1-\gamma)^2}[|Q_{\theta_0,\pi_0}^{\text{soft}}(S,A) - Q_{\theta_0}^{\text{soft}}(S,A)| - |Q_{\theta_T,\pi_T}^{\text{soft}}(S,A) - Q_{\theta_T}^{\text{soft}}(S,A)|]$.

**Step (ii).** We now quantify the difference between the stationary distribution $\mu^{\pi_E}(\cdot,\cdot)$ and the correlated distribution $\mathbb{P}_t^{\pi_E}(\cdot,\cdot)$. Note that $||\theta_t||$ is bounded (proved in Claim 1), the soft Bellman policy $\pi_\theta$ is continuous in $\theta$ and $(s, a)$, and we assume that the state-action space is bounded, there is a positive constant $C_\pi$ such that $||\log\pi_{\theta_t}(a|s)|| \leq C_\pi$ for any $(s, a) \in \mathcal{S} \times \mathcal{A}$. We start with the following relation:

$$\left|\left|E_{(S_t^E, A_t^E)\sim\mu^{\pi_E}(\cdot,\cdot)}\left[L_t(\theta_t; (S_t^E, A_t^E))\right] - E_{(S_t^E, A_t^E)\sim\mathbb{P}_t^{\pi_E}(\cdot,\cdot)}\left[L_t(\theta_t; (S_t^E, A_t^E))\right]\right|\right|,$$

$$= \gamma^t\left|\left|\int_{s\in\mathcal{S}}\int_{a\in\mathcal{A}}|\mathbb{P}_t^{\pi_E}(s,a) - \mu^{\pi_E}(s,a)| \cdot ||\log\pi_{\theta_t}(a|s)||dads\right|\right|,$$

$$\leq C_\pi C_M\gamma^t\rho^t,$$

$$\Rightarrow \left|\left|E_{(S_t^E, A_t^E)\sim\mu^{\pi_E}(\cdot,\cdot)}\left[\sum_{t=0}^{T-1}L_t(\theta_t; (S_t^E, A_t^E))\right] - E_{(S_t^E, A_t^E)\sim\mathbb{P}_t^{\pi_E}(\cdot,\cdot)}\left[\sum_{t=0}^{T-1}L_t(\theta_t; (S_t^E, A_t^E))\right]\right|\right|,$$

$$\leq \frac{C_\pi C_M}{1-\gamma\rho}. \tag{24}$$

Therefore, we have that

$$E_{(S_t^E, A_t^E) \sim \mathbb{P}_t^{\pi_E}}\left[\sum_{t=0}^{T-1} L_t(\theta_t; (S_t^E, A_t^E))\right] - \min_\theta E_{(S_t^E, A_t^E) \sim \mathbb{P}_t^{\pi_E}}\left[\sum_{t=0}^{T-1} L_t(\theta; (S_t^E, A_t^E))\right],$$

$$\leq E_{(S_t^E, A_t^E) \sim \mathbb{P}_t^{\pi_E}}\left[\sum_{t=0}^{T-1} L_t(\theta_t; (S_t^E, A_t^E))\right] - E_{(S_t^E, A_t^E) \sim \mu^{\pi_E}}\left[\sum_{t=0}^{T-1} L_t(\theta_t; (S_t^E, A_t^E))\right]$$

$$+ E_{(S_t^E, A_t^E) \sim \mu^{\pi_E}}\left[\sum_{t=0}^{T-1} L_t(\theta_t; (S_t^E, A_t^E)) - \sum_{t=0}^{T-1} L_t(\theta^*; (S_t^E, A_t^E))\right]$$

$$+ E_{(S_t^E, A_t^E) \sim \mu^{\pi_E}}\left[\sum_{t=0}^{T-1} L_t(\theta^*; (S_t^E, A_t^E))\right] - \min_\theta E_{(S_t^E, A_t^E) \sim \mathbb{P}_t^{\pi_E}}\left[\sum_{t=0}^{T-1} L_t(\theta; (S_t^E, A_t^E))\right],$$

$$\leq E_{(S_t^E, A_t^E) \sim \mathbb{P}_t^{\pi_E}}\left[\sum_{t=0}^{T-1} L_t(\theta_t; (S_t^E, A_t^E))\right] - E_{(S_t^E, A_t^E) \sim \mu^{\pi_E}}\left[\sum_{t=0}^{T-1} L_t(\theta_t; (S_t^E, A_t^E))\right]$$

$$+ E_{(S_t^E, A_t^E) \sim \mu^{\pi_E}}\left[\sum_{t=0}^{T-1} L_t(\theta_t; (S_t^E, A_t^E)) - \sum_{t=0}^{T-1} L_t(\theta^*; (S_t^E, A_t^E))\right]$$

$$+ E_{(S_t^E, A_t^E) \sim \mu^{\pi_E}}\left[\sum_{t=0}^{T-1} L_t(\theta^*; (S_t^E, A_t^E))\right] - E_{(S_t^E, A_t^E) \sim \mathbb{P}_t^{\pi_E}}\left[\sum_{t=0}^{T-1} L_t(\theta^*; (S_t^E, A_t^E))\right],$$

$$\overset{(g)}{\leq} \frac{C_\pi C_M}{1-\gamma\rho} + \bar{D}_4 + \frac{32 C_Q C_d \hat{C} \bar{C}_r^2}{\lambda(1-\gamma)^3}(\log T + 1) + \frac{C_\pi C_M}{1-\gamma\rho},$$

$$= D_4 + D_5(\log T + 1),$$

where $(f)$ follows (23) and (24), $D_4 = \frac{2C_\pi C_M}{1-\gamma\rho} + \bar{D}_4$, and $D_5 = \frac{32 C_Q C_d \hat{C} \bar{C}_r^2}{\lambda(1-\gamma)^3}$.

## C  Meta-Regularization Algorithm and Convergence Guarantee

To solve problem (5), we use a double-loop method, i.e., we first solve the lower-level problem for $K$ iterations to get an approximate $\phi_{j,K}$ of $\phi_j$ and then use the approximate $\phi_{j,K}$ to solve the upper-level problem. We do not use a single-loop method to solve (5) because we need to get the task-specific adaptation $\phi_j$ for each task $\mathcal{T}_j$. The single-loop method only partially solves the lower-level problem by one-step gradient descent and thus the obtained parameter can be far away from $\phi_j$.

**Lemma 5.** *The gradient of the lower-level problem in (5) is* $E_{S,A}^{\pi_\phi}\left[\sum_{t=0}^\infty \gamma^t \nabla_\phi r_\phi(S_t, A_t) \big| S_0 = s_0^{\text{tr}}\right] - \sum_{t=0}^\infty \gamma^t \nabla_\phi r_\phi(s_t^{\text{tr}}, a_t^{\text{tr}}) + \frac{\lambda}{1-\gamma}(\phi - \bar{\theta})$ *where* $(s_t^{\text{tr}}, a_t^{\text{tr}}) \in \mathcal{D}_j^{\text{tr}}$.

We roll out the policy $\pi_\phi$ from $s_t^{\text{tr}}$ to get a trajectory $s_0^\phi, a_0^\phi, \cdots$ where $s_0^\phi = s_t^{\text{tr}}$ and approximate the lower-level gradient by $g_\phi = \sum_{t=0}^\infty \gamma^t \nabla_\phi r_\phi(s_t^\phi, a_t^\phi) - \sum_{t=0}^\infty \gamma^t \nabla_\phi r_\phi(s_t^{\text{tr}}, a_t^{\text{tr}}) + \frac{\lambda}{1-\gamma}(\phi - \bar{\theta})$. We update $\phi_{j,k+1} = \phi_{j,k} - \beta_k g_{\phi_{j,k}}$ for $K$ times to get $\phi_{j,K}$ where $\beta_k$ is the step size and $\phi_{j,k}$ is the parameter at time $k$, and use $\phi_{j,K}$ to calculate the approximate of the hyper-gradient $\frac{d}{d\theta}L(\phi_j, \mathcal{D}_j^{\text{eval}})$.

**Lemma 6.** *The hyper-gradient (i.e., gradient of the upper-level problem in (5)) is* $\frac{d}{d\theta}L(\phi_j, \mathcal{D}_j^{\text{eval}}) = \left[I + \frac{1-\gamma}{\lambda}\nabla_{\phi\phi}^2 L(\phi_j, \mathcal{D}_j^{\text{tr}})\right]^{-1}\nabla_\phi L(\phi_j, \mathcal{D}_j^{\text{eval}})$, *where*

$$\nabla_\phi L(\phi_j, \mathcal{D}_j^{\text{eval}}) = |\mathcal{D}_j^{\text{eval}}| \cdot E_{S,A}^{\pi_{\phi_j}}\left[\sum_{t=0}^\infty \gamma^t \nabla_\phi r_{\phi_j}(S_t, A_t) \Big| S_0 \sim P_0\right] - \sum_{v=1}^{|\mathcal{D}_j^{\text{eval}}|}\sum_{t=0}^\infty \gamma^t \nabla_\phi r_{\phi_j}(s_t^v, a_t^v),$$

$$\nabla_{\phi\phi}^2 L(\phi_j, \mathcal{D}_j^{\text{tr}}) = E_{S,A}^{\pi_{\phi_j}}\left[\sum_{t=0}^\infty \gamma^t \nabla_{\phi\phi}^2 r_{\phi_j}(S_t, A_t) \Big| S_0 = s_0^{\text{tr}}\right] - \sum_{t=0}^\infty \gamma^t \nabla_{\phi\phi}^2 r_{\phi_j}(s_t^{\text{tr}}, a_t^{\text{tr}}) + e.$$

The $|\mathcal{D}_j^{\text{eval}}|$ is the number of trajectories in $\mathcal{D}_j^{\text{eval}}$, $(s_t^v, a_t^v) \in \mathcal{D}_j^{\text{eval}}$, and $e$ is an extra term whose expression can be found in Appendix C.2.

To approximate the hyper-gradient, we first roll out $\pi_{\phi_{j,K}}$ to get two trajectories $s_0^{\phi_{j,K}}, a_0^{\phi_{j,K}}, \cdots$ and $\bar{s}_0^{\phi_{j,K}}, \bar{a}_0^{\phi_{j,K}}, \cdots$ where $s_0^{\phi_{j,K}}$ is drawn from $P_0$ and $\bar{s}_0^{\phi_{j,K}} = s_0^{\text{tr}}$. We estimate $\nabla_\phi L(\phi_j, \mathcal{D}_j^{\text{eval}})$ via
$$\bar{\nabla}_\phi L(\phi_{j,K}, \mathcal{D}_j^{\text{eval}}) = |\mathcal{D}_j^{\text{eval}}| \cdot \sum_{t=0}^\infty \gamma^t \nabla_\phi r_{\phi_{j,K}}(s_t^{\phi_{j,K}}, a_t^{\phi_{j,K}}) - \sum_{v=1}^{|\mathcal{D}_j^{\text{eval}}|} \sum_{t=0}^\infty \gamma^t \nabla_\phi r_{\phi_{j,K}}(s_t^v, a_t^v)$$
and estimate the term $\nabla_{\phi\phi}^2 L(\phi_j, \mathcal{D}_j^{\text{tr}})$ via $\bar{\nabla}_{\phi\phi}^2 L(\phi_{j,K}, \mathcal{D}_j^{\text{tr}}) = \sum_{t=0}^\infty \gamma^t \nabla_{\phi\phi}^2 r_{\phi_{j,K}}(\bar{s}_t^{\phi_{j,K}}, \bar{a}_t^{\phi_{j,K}}) - \sum_{t=0}^\infty \gamma^t \nabla_{\phi\phi}^2 r_{\phi_{j,K}}(s_t^{\text{tr}}, a_t^{\text{tr}})$. Therefore, we can approximate the hyper-gradient term $\frac{d}{d\theta} L(\phi_j, \mathcal{D}_j^{\text{eval}})$ via $h_j = [I + \frac{1-\gamma}{\lambda} \bar{\nabla}_{\phi\phi}^2 L(\phi_{j,K}, \mathcal{D}_j^{\text{tr}})]^{-1} \bar{\nabla}_\phi L(\phi_{j,K}, \mathcal{D}_j^{\text{eval}})$. We omit the extra term $e$ in the approximate $h_j$ and its impact on the convergence can be bounded (proved in Appendix C.4).

To solve the upper-level problem in (5), at each iteration $n$, we sample a batch of $B$ tasks, and compute the task-specific adaptation $\phi_{j,K}$ and the hyper-gradient $h_j$ for each task. The update law to solve the upper-level problem is: $\bar{\theta}_{n+1} = \bar{\theta}_n - \frac{\tau_n}{B} \sum_{j=1}^B h_j$ where $\tau_n$ is the step size. Note that the time index $k$ is for the lower-level problem and $n$ is for the upper-level problem.

---

**Algorithm 2** Meta-regularization

---

**Input**: Initialized meta-prior $\bar{\theta}_0$ and task-specific adaptation $\phi_{j,0}$
**Output**: Learned prior $\bar{\theta}_N$
1: **for** $n = 0, 1, \cdots, N-1$ **do**
2:     Sample a batch of $B$ tasks $\{\mathcal{T}_j\}_{j=1}^B \sim P_\mathcal{T}$
3:     **for** each task $\mathcal{T}_j$ **do**
4:         **for** $k = 0, 1, \cdots, K-1$ **do**
5:             Compute the soft Bellman policy $\pi_{\phi_{j,k}}$ via soft Q-learning or soft actor-critic
6:             Roll out the policy $\pi_{\phi_{j,k}}$ to get a trajectory $s_0^{\phi_{j,k}}, a_0^{\phi_{j,k}}, \cdots$
7:             Compute the gradient $g_{\phi_{j,k}} = \sum_{t=0}^\infty \gamma^t \nabla_\phi r_{\phi_{j,k}}(s_t^{\phi_{j,k}}, a_t^{\phi_{j,k}}) - \sum_{t=0}^\infty \gamma^t \nabla_\phi r_{\phi_{j,k}}(s_t^{\text{tr}}, a_t^{\text{tr}})$
              $+ \frac{\lambda}{1-\gamma}(\phi_{j,k} - \bar{\theta}_n)$
8:             Update $\phi_{j,k+1} = \phi_{j,k} - \beta_k g_{\phi_{j,k}}$
9:         **end for**
10:         Compute the soft Bellman policy $\pi_{\phi_{j,K}}$ via soft Q-learning or soft actor-critic
11:         Roll out the policy $\pi_{\phi_{j,K}}$ to get two trajectories $s_0^{\phi_{j,K}}, a_0^{\phi_{j,K}}, \cdots$ and $\bar{s}_0^{\phi_{j,K}}, \bar{a}_0^{\phi_{j,K}}, \cdots$
12:         Compute the hyper-gradient $h_j = [I + \frac{1-\gamma}{\lambda} \bar{\nabla}_{\phi\phi}^2 L(\phi_{j,K}, \mathcal{D}_j^{\text{tr}})]^{-1} \bar{\nabla}_\phi L(\phi_{j,K}, \mathcal{D}_j^{\text{eval}})$
13:     **end for**
14:     Update $\bar{\theta}_{n+1} = \bar{\theta}_n - \frac{\tau_n}{B} \sum_{j=1}^B h_j$
15: **end for**

---

**Lemma 7** (Convergence of the lower-level problem). *Suppose Assumptions 1-2 hold and $\lambda \geq \frac{C_L}{2} + \eta$ where $\eta \in (0, \frac{C_L}{2})$. Let $\beta_k = \frac{1-\gamma}{\eta(k+1)}$, then we have*

$$E[||\phi_{j,K} - \phi_j||^2] \leq O(\frac{1}{K}).$$

**Assumption 3.** *The parameterized reward function $r_\theta$ has bounded third-order gradient, i.e., $||\nabla_{\theta\theta\theta}^3 r_\theta(s,a)|| \leq \hat{C}_r$ for any $(s,a)$ where $\hat{C}_r$ is a positive constant.*

**Theorem 3** (Convergence of the upper-level problem). *Suppose Assumption 3 and the condition in Lemma 7 hold. Let $\tau_n = (n+1)^{-1/2}$ and define $F(\bar{\theta})$ as $E_{j \sim P_\mathcal{T}}[L(\phi_j, \mathcal{D}_j^{\text{eval}})]$ under the meta-prior $\bar{\theta}$. Then we have the following convergence:*

$$\frac{1}{N} \sum_{n=0}^{N-1} E[||\nabla F(\bar{\theta}_n)||^2] \leq O(\frac{1}{\sqrt{N}} + \frac{\log N}{\sqrt{N}} + \frac{1}{K}) + C_1,$$

*and the expression of $C_1$ can be found in (27).*

## C.1 Proof of Lemma 5

The proof is similar to that of Lemma 1. We first prove the case of deterministic dynamics and the corresponding result can be an unbiased estimate in cases of stochastic dynamics (proved in Subsection B.2).

$$\nabla_\phi L(\phi, \mathcal{D}_j^{\mathrm{tr}}) = -\sum_{t=0}^{\infty} \gamma^t \nabla_\phi \log \pi_\phi(a_t^{\mathrm{tr}}|s_t^{\mathrm{tr}}),$$

$$= -\sum_{t=0}^{\infty} \gamma^t \left[ \nabla_\phi Q_\phi^{\mathrm{soft}}(s_t^{\mathrm{tr}}, a_t^{\mathrm{tr}}) - \nabla_\phi V_\phi^{\mathrm{soft}}(s_t^{\mathrm{tr}}) \right],$$

$$= -\sum_{t=0}^{\infty} \gamma^t \left[ \nabla_\phi r_\phi(s_t^{\mathrm{tr}}, a_t^{\mathrm{tr}}) + \gamma \nabla_\phi V_\phi^{\mathrm{soft}}(s_{t+1}^{\mathrm{tr}}) - \nabla_\phi V_\phi^{\mathrm{soft}}(s_t^{\mathrm{tr}}) \right],$$

$$= \nabla_\phi V_\phi^{\mathrm{soft}}(s_0^{\mathrm{tr}}) - \sum_{t=0}^{\infty} \gamma^t \nabla_\phi r_\phi(s_t^{\mathrm{tr}}, a_t^{\mathrm{tr}}),$$

$$\stackrel{(a)}{=} E_{S,A}^{\pi_\phi} \left[ \sum_{t=0}^{\infty} \gamma^t \nabla_\phi r_\phi(S_t, A_t) \Big| S_0 = s_0^{\mathrm{tr}} \right] - \sum_{t=0}^{\infty} \gamma^t \nabla_\phi r_\phi(s_t^{\mathrm{tr}}, a_t^{\mathrm{tr}}),$$

where $(a)$ follows the proof of Lemma 3.

## C.2 Proof of Lemma 6

Since the lower-level problem of (5) is unconstrained and $\phi_i$ is the optimal solution, we know that

$$\nabla_\phi L(\phi_j, \mathcal{D}_j^{\mathrm{tr}}) + \frac{\lambda}{1-\gamma}(\phi_j - \bar{\theta}) = 0 \tag{25}$$

We further take derivative of both sides in (25) with respect to $\bar{\theta}$ and we get that

$$\left[ \nabla_{\phi\phi}^2 L(\phi_j, \mathcal{D}_j^{\mathrm{tr}}) + \frac{\lambda}{1-\gamma} \right] \nabla_{\bar{\theta}} \phi_j - \frac{\lambda}{1-\gamma} = 0 \Rightarrow \nabla_{\bar{\theta}} \phi_j = \left[ I + \frac{1-\gamma}{\lambda} \nabla_{\phi\phi}^2 L(\phi_j, \mathcal{D}_j^{\mathrm{tr}}) \right]^{-1}.$$

Therefore, we have that

$$\frac{d}{d\bar{\theta}} L(\phi_j, \mathcal{D}_j^{\mathrm{eval}}) = (\nabla_{\bar{\theta}} \phi_j)^\top \nabla_\phi L(\phi_j, \mathcal{D}_j^{\mathrm{eval}}) = \left[ I + \frac{1-\gamma}{\lambda} \nabla_{\phi\phi}^2 L(\phi_j, \mathcal{D}_j^{\mathrm{tr}}) \right]^{-1} \nabla_\phi L(\phi_j, \mathcal{D}_j^{\mathrm{eval}}).$$

Similar to Lemma 5, we can know that

$$\nabla_\phi L(\phi_j, \mathcal{D}_j^{\mathrm{eval}}) = \sum_{v=1}^{|\mathcal{D}_j^{\mathrm{eval}}|} E_{S,A}^{\pi_{\phi_j}} \left[ \sum_{t=0}^{\infty} \gamma^t \nabla_\phi r_{\phi_j}(S_t, A_t) \Big| S_0 = s_0^v \right] - \sum_{v=1}^{|\mathcal{D}_j^{\mathrm{eval}}|} \sum_{t=0}^{\infty} \gamma^t \nabla_\phi r_{\phi_j}(s_t^v, a_t^v), \tag{26}$$

where $(s_t^v, a_t^v) \in \zeta^v$ and $\zeta^v \in \mathcal{D}_j^{\mathrm{eval}}$.

Since there is usually abundant data in $\mathcal{D}_j^{\mathrm{eval}}$ and $S_0 \sim P_0$, we can reformulate (26) as the following:

$$\nabla_\phi L(\phi_j, \mathcal{D}_j^{\mathrm{eval}}) \approx |\mathcal{D}_j^{\mathrm{eval}}| \cdot E_{S,A}^{\pi_{\phi_j}} \left[ \sum_{t=0}^{\infty} \gamma^t \nabla_\phi r_{\phi_j}(S_t, A_t) \Big| S_0 \sim P_0 \right] - \sum_{v=1}^{|\mathcal{D}_j^{\mathrm{eval}}|} \sum_{t=0}^{\infty} \gamma^t \nabla_\phi r_{\phi_j}(s_t^v, a_t^v).$$

**Claim 7.** *The second-order information* $\nabla_{\phi\phi}^2 Q_\phi^{soft}(s, a) = \Delta(s, a) + E_{S,A}^{\pi_\phi}[\sum_{t=0}^{\infty} \gamma^t Cov(S_t)|S_0 = s, A_0 = a]$ *and* $\nabla_{\phi\phi}^2 V_\phi^{soft}(s) = \Delta(s) + E_{S,A}^{\pi_\phi}[\sum_{t=0}^{\infty} \gamma^t Cov(S_t)|S_0 = s]$ *where* $\Delta(s, a) = E_{S,A}^{\pi_\phi}[\sum_{t=0}^{\infty} \gamma^t \nabla_{\phi\phi}^2 r_\phi(S_t, A_t)|S_0 = s, A_0 = a]$, $\Delta(s) = E_{S,A}^{\pi_\phi}[\sum_{t=0}^{\infty} \gamma^t \nabla_{\phi\phi}^2 r_\phi(S_t, A_t)|S_0 = s]$, *and* $Cov(s) \triangleq \int_{a \in \mathcal{A}} \pi_\phi(a|s)[\Delta(s, a) - \Delta(s)]\Delta(s)da$ *is the covariance matrix of* $\Delta(s, \cdot)$ *at state s.*

The proof of Claim 7 follows the proof of Lemma 4

Now we take a look at the term $\nabla^2_{\phi\phi} L(\phi_j, \mathcal{D}^{\text{tr}}_j)$ and consider the case of deterministic dynamics:

$$\nabla^2_{\phi\phi} L(\phi_j, \mathcal{D}^{\text{tr}}_j) = -\sum_{t=0}^{\infty} \gamma^t \nabla^2_{\phi\phi} \log \pi_{\phi_j}(a^{\text{tr}}_t | s^{\text{tr}}_t),$$

$$= -\sum_{t=0}^{\infty} \gamma^t \left[ \nabla^2_{\phi\phi} Q^{\text{soft}}_{\phi_j}(s^{\text{tr}}_t, a^{\text{tr}}_t) - \nabla^2_{\phi\phi} V^{\text{soft}}_{\phi_j}(s^{\text{tr}}_t) \right],$$

$$= -\sum_{t=0}^{\infty} \gamma^t \left[ \nabla^2_{\phi\phi} r_{\phi_j}(s^{\text{tr}}_t, a^{\text{tr}}_t) + \gamma \nabla^2_{\phi\phi} V^{\text{soft}}_{\phi_j}(s^{\text{tr}}_{t+1}) - \nabla^2_{\phi\phi} V^{\text{soft}}_{\phi_j}(s^{\text{tr}}_t) \right],$$

$$= \nabla^2_{\phi\phi} V^{\text{soft}}_{\phi_j}(s^{\text{tr}}_0) - \sum_{t=0}^{\infty} \gamma^t \nabla^2_{\phi\phi} r_{\phi_j}(s^{\text{tr}}_t, a^{\text{tr}}_t),$$

$$\stackrel{(a)}{=} E^{\pi_{\phi_j}}_{S,A} [\sum_{t=0}^{\infty} \gamma^t \nabla^2_{\phi\phi} r_{\phi_j}(S_t, A_t) | S_0 = s^{\text{tr}}_0] - \sum_{t=0}^{\infty} \gamma^t \nabla^2_{\phi\phi} r_{\phi_j}(s^{\text{tr}}_t, a^{\text{tr}}_t)$$

$$+ E^{\pi_{\phi_j}}_{S,A} [\sum_{t=0}^{\infty} \gamma^t \text{Cov}(S_t) | S_0 = s^{\text{tr}}_0],$$

where $(a)$ follows Claim 7. As proved in Subsection B.2, we can still use this expression in the case of stochastic dynamics. We define $e \triangleq E^{\pi_{\phi_j}}_{S,A} [\sum_{t=0}^{\infty} \gamma^t \text{Cov}(S_t) | S_0 = s^{\text{tr}}_0]$. However, $e$ is intractable to compute because it requires to compute $\text{COV}(S_t)$ at every visited state. To approximate the covariance matrix $\text{COV}(S_t)$, we need to empirically roll out the policy $\pi_{\phi_j}$ from $S_t$ for enough times to get enough samples. We need to do these policy roll-outs at every state $S_t$. For example, suppose we roll out the policy $\pi_{\phi_j}$ ten times to get ten samples at each state $S_t$. Empirically, we need to do these roll-outs $10 \times \bar{T}$ where $\bar{T}$ is a very large integer that we regard as infinity because the trajectory horizon is infinite. This is intractable because we need to roll out the policy for too many times. Moreover, we cannot guarantee that we can approximate $\text{COV}(S_t)$ well given that we only use ten samples to approximate it.

## C.3 Proof of Lemma 7

We know that $||\nabla^2_{\phi\phi} L(\phi, \mathcal{D}^{\text{tr}}_j) + \frac{\lambda}{1-\gamma}|| \leq ||\nabla^2_{\phi\phi} L(\phi, \mathcal{D}^{\text{tr}}_j)|| + \frac{\lambda}{1-\gamma} \leq \sum_{t=0}^{\infty} \left( ||\nabla^2 L_t(\phi)|| + \lambda \gamma^t \right) \stackrel{(a)}{\leq} \frac{1}{1-\gamma} C_L$ where $(a)$ follows (6). Therefore, the lower-level objective function in (8) is $\frac{C_L}{1-\gamma}$-smooth. Moreover, since $\lambda \geq \frac{C_L}{2} + \eta$, then $||\nabla^2_{\phi\phi} L(\phi, \mathcal{D}^{\text{tr}}_j)|| \leq \frac{C_L - 2\eta}{2(1-\gamma)}$. Therefore, the lower-level objective function in (8) is $\frac{2\eta}{1-\gamma}$-strongly convex. Following the standard result for strongly-convex and smooth stochastic optimization, we can reach the result in Lemma 7.

## C.4 Proof of Theorem 3

In this proof, we first bound the hyper-gradient approximation error (i.e., $||\frac{d}{d\bar{\theta}} L(\phi_j, \mathcal{D}^{\text{eval}}_j) - h_j||$) and then prove the convergence. Define $\bar{h}_j \triangleq [I + \frac{1-\gamma}{\lambda} \bar{\nabla}^2_{\phi\phi} L(\phi_j, \mathcal{D}^{\text{tr}}_j)]^{-1} \nabla_\phi L(\phi_j, \mathcal{D}^{\text{eval}}_j)$ where $\bar{\nabla}^2_{\phi\phi} L(\phi_j, \mathcal{D}^{\text{tr}}_j) \triangleq E^{\pi_{\phi_j}}_{S,A} [\sum_{t=0}^{\infty} \gamma^t \nabla^2_{\phi\phi} r_{\phi_j}(S_t, A_t) | S_0 = s^{\text{tr}}_0] - \sum_{t=0}^{\infty} \gamma^t \nabla^2_{\phi\phi} r_{\phi_j}(s^{\text{tr}}_t, a^{\text{tr}}_t)$. Therefore, we have that

$$||\bar{h}_j - \frac{d}{d\bar{\theta}} L(\phi_j, \mathcal{D}^{\text{eval}}_j)||,$$

$$\leq \left\| [I + \frac{1-\gamma}{\lambda} \nabla^2_{\phi\phi} L(\phi_j, \mathcal{D}^{\text{tr}}_j)]^{-1} - [I + \frac{1-\gamma}{\lambda} \bar{\nabla}^2_{\phi\phi} L(\phi_j, \mathcal{D}^{\text{tr}}_j)]^{-1} \right\| \cdot ||\nabla_\phi L(\phi_j, \mathcal{D}^{\text{eval}}_j)||,$$

$$\leq \left[ \left\| [I + \frac{1-\gamma}{\lambda} \nabla^2_{\phi\phi} L(\phi_j, \mathcal{D}^{\text{tr}}_j)]^{-1} \right\| + \left\| [I + \frac{1-\gamma}{\lambda} \bar{\nabla}^2_{\phi\phi} L(\phi_j, \mathcal{D}^{\text{tr}}_j)]^{-1} \right\| \right] \cdot ||\nabla_\phi L(\phi_j, \mathcal{D}^{\text{eval}}_j)||,$$

$$\stackrel{(a)}{\leq} \left( \frac{2\lambda}{2\lambda + 2\eta - C_L} + \frac{\lambda}{\lambda - 2\tilde{C}_r} \right) \cdot \frac{2|\mathcal{D}^{\text{eval}}_j| \bar{C}_r}{1-\gamma} \triangleq C_1 > 0, \tag{27}$$

where $(a)$ follows the fact that $||I + \frac{1-\gamma}{\lambda}\nabla^2_{\phi\phi}L(\phi_j, \mathcal{D}^{\text{tr}}_j)|| \geq 1 - ||\frac{1-\gamma}{\lambda}\nabla^2_{\phi\phi}L(\phi_j, \mathcal{D}^{\text{tr}}_j)|| \geq 1 - \frac{C_L - 2\eta}{2\lambda}$ given that $||\nabla^2_{\phi\phi}L(\phi, \mathcal{D}^{\text{tr}}_j)|| \leq \frac{C_L - 2\eta}{2(1-\gamma)}$ (proved in the proof of Lemma 7). Therefore $||[I + \frac{1-\gamma}{\lambda}\nabla^2_{\phi\phi}L(\phi_j, \mathcal{D}^{\text{tr}}_j)]^{-1}|| \leq \frac{2\lambda}{2\lambda + 2\eta - C_L}$. Similarly, we can bound $||[I + \frac{1-\gamma}{\lambda}\bar{\nabla}^2_{\phi\phi}L(\phi_j, \mathcal{D}^{\text{tr}}_j)]^{-1}|| \leq \frac{\lambda}{\lambda - 2\tilde{C}_r}$ given that $||\bar{\nabla}^2_{\phi\phi}L(\phi_j, \mathcal{D}^{\text{tr}}_j)|| \leq \frac{2\tilde{C}_r}{1-\gamma}$. Note that $\lambda > \frac{C_L}{2}$ and $C_L > \tilde{C}_r$ (proved in (6)), therefore $C_1$ is a positive constant.

Now, we bound the term $||h_j - \bar{h}_j||$. We define $\Delta_{\phi_j} = I + \frac{1-\gamma}{\lambda}\bar{\nabla}^2_{\phi\phi}L(\phi_j, \mathcal{D}^{\text{tr}}_j)$ and $\Delta_{\phi_j} = I + \frac{1-\gamma}{\lambda}\bar{\nabla}^2_{\phi\phi}L(\phi_{j,K}, \mathcal{D}^{\text{tr}}_j)$. Thus we have $||\Delta^{-1}_{\phi_j}|| \leq \frac{\lambda}{\lambda - 2\tilde{C}_r}$ (follows (27)) and similarly $||\Delta^{-1}_{\phi_{j,K}}|| \leq \frac{\lambda}{\lambda - 2\tilde{C}_r}$. Therefore,

$$E[||h_j - \bar{h}_j||] = E\left[\left|\left|\Delta^{-1}_{\phi_{j,K}}\nabla_\phi L(\phi_{j,K}, \mathcal{D}^{\text{eval}}_j) - \Delta^{-1}_{\phi_j}\nabla_\phi L(\phi_j, \mathcal{D}^{\text{eval}}_j)\right|\right|\right],$$

$$\leq E\left[\left|\left|\Delta^{-1}_{\phi_{j,K}}\nabla_\phi L(\phi_{j,K}, \mathcal{D}^{\text{eval}}_j) - \Delta^{-1}_{\phi_{j,K}}\nabla_\phi L(\phi_j, \mathcal{D}^{\text{eval}}_j)\right|\right|\right.$$

$$\left. + \left|\left|\Delta^{-1}_{\phi_{j,K}}\nabla_\phi L(\phi_j, \mathcal{D}^{\text{eval}}_j) - \Delta^{-1}_{\phi_j}\nabla_\phi L(\phi_j, \mathcal{D}^{\text{eval}}_j)\right|\right|\right],$$

$$\leq E\left[||\Delta^{-1}_{\phi_{j,K}}|| \cdot ||\nabla_\phi L(\phi_{j,K}, \mathcal{D}^{\text{eval}}_j) - \nabla_\phi L(\phi_j, \mathcal{D}^{\text{eval}}_j)|| + ||\Delta^{-1}_{\phi_{j,K}} - \Delta^{-1}_{\phi_j}|| \cdot ||\nabla_\phi L(\phi_j, \mathcal{D}^{\text{eval}}_j)||\right],$$

$$\overset{(b)}{\leq} \frac{\lambda}{\lambda - 2\tilde{C}_r} \cdot |\mathcal{D}^{\text{eval}}_j| \cdot \frac{C_L - 2\eta}{2(1-\gamma)}||\phi_{j,K} - \phi_j|| + E[||\Delta^{-1}_{\phi_{j,K}} - \Delta^{-1}_{\phi_j}||] \cdot |\mathcal{D}^{\text{eval}}_j| \cdot \frac{2\bar{C}_r}{1-\gamma},$$

$$\leq O(\frac{1}{K}) + E[||\Delta^{-1}_{\phi_{j,K}}|| \cdot ||\Delta^{-1}_{\phi_j}|| \cdot ||\Delta^{-1}_{\phi_{j,K}} - \Delta^{-1}_{\phi_j}||] \cdot |\mathcal{D}^{\text{eval}}_j| \cdot \frac{2\bar{C}_r}{1-\gamma},$$

$$\leq O(\frac{1}{K}) + \bar{C}_L E[||\Delta^{-1}_{\phi_{j,K}}|| \cdot ||\Delta^{-1}_{\phi_j}||] \cdot |\mathcal{D}^{\text{eval}}_j| \cdot \frac{2\bar{C}_r}{1-\gamma}||\phi_{j,K} - \phi_j|| \leq O(\frac{1}{K}), \tag{28}$$

where $||\nabla^3_{\phi\phi\phi}L(\phi_j, \mathcal{D}^{\text{tr}}_j)|| \leq \tilde{C}_L$. The expression of $\tilde{C}_L$ can be derived following the proof of Lemma 4 by notifying Assumption 3. The $(b)$ holds because (i) $||\nabla^2_{\phi\phi}L(\phi, \mathcal{D}^{\text{eval}}_j)|| = \frac{|\mathcal{D}^{\text{eval}}_j|}{|\mathcal{D}^{\text{tr}}_j|}||\nabla^2_{\phi\phi}L(\phi, \mathcal{D}^{\text{tr}}_j)|| \leq |\mathcal{D}^{\text{eval}}_j| \cdot \frac{C_L - 2\eta}{2(1-\gamma)}$ and (ii) $||\nabla_\phi L(\phi_j, \mathcal{D}^{\text{eval}}_j)|| \leq |\mathcal{D}^{\text{eval}}_j| \cdot \frac{2\bar{C}_r}{1-\gamma}$ (see the expression of $\nabla_\phi L(\phi_j, \mathcal{D}^{\text{eval}}_j)$ in Lemma 5).

$$E[||h_j - \frac{d}{d\bar{\theta}}L(\phi_j, \mathcal{D}^{\text{eval}}_j)||] \leq E[||h_j - \bar{h}_j|| + ||\bar{h}_j - \frac{d}{d\bar{\theta}}L(\phi_j, \mathcal{D}^{\text{eval}}_j)||] \overset{(c)}{\leq} C_1 + O(\frac{1}{K}), \tag{29}$$

where $(c)$ follows (27)-(28). Define $F(\bar{\theta})$ as $E_{j \sim P_\mathcal{T}}[L(\phi_j, \mathcal{D}^{\text{eval}}_j)]$ under the meta-prior $\bar{\theta}$. Note that $||h_j|| \leq ||\Delta^{-1}_{\phi_{j,K}}|| \cdot ||\nabla_\phi L(\phi_{j,K}, \mathcal{D}^{\text{eval}}_j)|| \leq \frac{\lambda}{\lambda - 2\tilde{C}_r} \cdot |\mathcal{D}^{\text{eval}}_j| \cdot \frac{2\bar{C}_r}{1-\gamma}$. Therefore,

$$F(\bar{\theta}_{n+1}) \geq F(\bar{\theta}_n) + (\nabla F(\bar{\theta}_n))^\top (\bar{\theta}_{n+1} - \bar{\theta}_n) - \frac{|\mathcal{D}^{\text{eval}}|(C_L - 2\eta)}{4(1-\gamma)}||\bar{\theta}_{n+1} - \bar{\theta}_n||^2,$$

$$\geq F(\bar{\theta}_n) + \frac{\tau_n}{B}\sum_{j=1}^{B}(\nabla F(\bar{\theta}_n))^\top h_j - \frac{|\mathcal{D}^{\text{eval}}|(C_L - 2\eta)\tau_n^2}{4(1-\gamma)}||\frac{1}{B}\sum_{j=1}^{B}h_j||^2,$$

$$\geq F(\bar{\theta}_n) + \frac{\tau_n}{B}\sum_{j=1}^{B}(\nabla F(\bar{\theta}_n))^\top h_j - \frac{|\mathcal{D}^{\text{eval}}|(C_L - 2\eta)\tau_n^2}{4(1-\gamma)} \cdot \frac{\lambda}{\lambda - 2\tilde{C}_r} \cdot |\mathcal{D}^{\text{eval}}_j| \cdot \frac{2\bar{C}_r}{1-\gamma},$$

$$\geq F(\bar{\theta}_n) + \frac{\tau_n}{B}\sum_{j=1}^{B}(\nabla F(\bar{\theta}_n))^\top h_j - \frac{2\lambda\bar{C}_r|\mathcal{D}^{\text{eval}}|^2(C_L - 2\eta)\tau_n^2}{4(1-\gamma)^2(\lambda - 2\tilde{C}_r)},$$

$$\Rightarrow E[F(\bar{\theta}_{n+1})] \geq E[F(\bar{\theta}_n)] + \tau_n E[||\nabla F(\bar{\theta}_n)||^2] + \frac{\tau_n}{B}\sum_{j=1}^{B}E[h_j - \frac{d}{d\bar{\theta}}L(\phi_j, \mathcal{D}^{\text{eval}}_j)]$$

$$+ \frac{\tau_n}{B}\sum_{j=1}^{B}E[\frac{d}{d\bar{\theta}}L(\phi_j, \mathcal{D}^{\text{eval}}_j) - \nabla F(\bar{\theta}_n)] - \frac{2\lambda\bar{C}_r|\mathcal{D}^{\text{eval}}|^2(C_L - 2\eta)\tau_n^2}{4(1-\gamma)^2(\lambda - 2\tilde{C}_r)},$$

$$\overset{(d)}{\geq} E[F(\bar{\theta}_n)] + \tau_n E[||\nabla F(\bar{\theta}_n)||^2] + \tau_n(C_1 + O(\frac{1}{K})) - \frac{2\lambda\bar{C}_r|\mathcal{D}^{\text{eval}}|^2(C_L - 2\eta)\tau_n^2}{4(1-\gamma)^2(\lambda - 2\tilde{C}_r)},$$

$$\Rightarrow \frac{1}{N}\sum_{n=0}^{N-1}\tau_N E[||\nabla F(\bar{\theta}_n)||^2] \leq \frac{1}{N}\sum_{n=0}^{N-1}\tau_n E[||\nabla F(\bar{\theta}_n)||^2],$$

$$\leq \frac{1}{N}[F(\bar{\theta}_N) - F(\bar{\theta}_0)] + \frac{1}{N}\sum_{n=0}^{N-1}\tau_n(C_1 + O(\frac{1}{K})) + \frac{1}{N}\sum_{n=0}^{N-1}\frac{2\lambda\bar{C}_r|\mathcal{D}^{\text{eval}}|^2(C_L - 2\eta)\tau_n^2}{4(1-\gamma)^2(\lambda - 2\tilde{C}_r)},$$

$$\Rightarrow \frac{1}{N}\sum_{n=0}^{N-1}E[||\nabla F(\bar{\theta}_n)||^2] \leq O(\frac{1}{\sqrt{N}} + \frac{\log N}{\sqrt{N}} + \frac{1}{K}) + C_1,$$

where $|\mathcal{D}^{\text{eval}}| \triangleq \sup_{i\sim P_\mathcal{T}}\{|\mathcal{D}_j^{\text{eval}}|\}$ and $(d)$ follows (29).

## D   Experiment details

The code was running on a laptop whose processor is AMD Ryzen 7 4700U with Radeon Graphics, 2.00GHz, and the installed RAM is 20.0GB. The operating system is Ubuntu 18.04. We use a neural network to parameterize the learned reward function. The neural network has two hidden layers where each hidden layer has 64 neurons. The activation functions are respectively ReLU and Tanh.

### D.1   Baselines

Here we provide the update rule for each baseline. Given a learned reward function $r_\theta$, the policy updates of the four baselines are the same with that of MERIT-IRL, i.e., one-step policy iteration. The difference is the reward update.

**IT-IRL**: IT-IRL is MERIT-IRL without the meta-regularization term. Therefore, the reward update of IT-IRL is $\theta_{t+1} = \theta_t - \alpha_t g_t'$ where $g_t' = \sum_{i=0}^{\infty}\gamma^i\nabla_\theta r_{\theta_t}(s_i', a_i') - \sum_{i=0}^{\infty}\gamma^i\nabla_\theta r_{\theta_t}(s_i'', a_i'')$. Recall from Subsection 4.1 that $\{(s_i', a_i')\}_{i\geq 0}$ is generated by the learned policy $\pi_{\theta_t}$ starting from the initial state $s_0' = s_0^E$, and $\{(s_i'', a_i'')\}_{i\geq 0}$ is generated by the learned policy $\pi_{\theta_t}$ starting from $(s_t'', a_t'')$ where $(s_i'', a_i'') = (s_i^E, a_i^E)$ for $0 \leq i \leq t$.

**Naive MERIT-IRL**: This method has the meta-regularization term, however, it uses the naive way (depicted in the middle of Figure 1) to update the reward function. In specific, it only compares the partial expert trajectory $\{s_i^E, a_i^E\}_{i=0}^t$ and partial learner trajectory $\{s_i', a_i'\}_{i=0}^t$. Therefore, the reward update of Naive MERIT-IRL is $\theta_{t+1} = \theta_t - \alpha_t g_t''$ where $g_t'' = \sum_{i=0}^t\gamma^i\nabla_\theta r_{\theta_t}(s_i', a_i') - \sum_{i=0}^t\gamma^i\nabla_\theta r_{\theta_t}(s_i^E, a_i^E) + \frac{\lambda(1-\gamma^{t+1})}{1-\gamma}(\theta - \bar{\theta})$.

**Naive IT-IRL**: This method does not have the meta-regularization term and uses the naive way to update the reward function. Therefore, the reward update of Naive IT-IRL is $\theta_{t+1} = \theta_t - \alpha_t g_t'''$ where $g_t''' = \sum_{i=0}^t\gamma^i\nabla_\theta r_{\theta_t}(s_i', a_i') - \sum_{i=0}^t\gamma^i\nabla_\theta r_{\theta_t}(s_i^E, a_i^E)$.

**Hindsight**: This method is a standard IRL method with the meta-regularization term where the complete expert trajectory $\{s_i^E, a_i^E\}_{i\geq 0}$ and the complete learner trajectory $\{s_i', a_i'\}_{i\geq 0}$ are compared to update the reward function. Therefore, the reward update of Hindsight is $\theta_{t+1} = \theta_t - \alpha_t g_t''''$ where $g_t'''' = \sum_{i=0}^{\infty}\gamma^i\nabla_\theta r_{\theta_t}(s_i', a_i') - \sum_{i=0}^{\infty}\gamma^i\nabla_\theta r_{\theta_t}(s_i^E, a_i^E) + \frac{\lambda(1-\gamma^{t+1})}{1-\gamma}(\theta - \bar{\theta})$.

### D.2   MuJoCo

#### D.2.1   Walker

Figure 2b shows that MERIT can achieve similar performance with the expert after $t = 600$ while the other three in-trajectory learning baselines fail to imitate the expert before the ongoing trajectory terminates. Note that the naive methods (i.e., Naive MERIT-IRL and Naive IT-IRL) have much smaller improvement from $t = 0$ compared to MERIT-IRL and IT-IRL. The reason is that the naive reward update method is flawed. Intuitively, the reward update mechanism of these two baselines are myopic as explained in Subsection 4.1. Theoretically, the gradients $g_t''$ of Naive MERIT-IRL and $g_t'''$

of Naive IT-IRL are biased estimate of (4) even if $\pi_t$ approaches $\pi_{\theta_t}$ since (4) includes the trajectory suffix ($i > t$) terms while $g_t''$ and $g_t'''$ only include the trajectory prefix ($i \leq t$) terms.

MERIT-IRL performs much better than IT-IRL. The reason is that the meta-regularization term restricts the learned reward parameter within a certain neighborhood of the meta-prior $\bar{\theta}$ (proved in Appendix B.3). Given that $\bar{\theta}$ is trained over a family of relevant tasks, it is expected that the actual reward function parameter of our task shall be "close" to $\bar{\theta}$ [25, 44, 57], i.e., inside this neighborhood. Therefore, MERIT-IRL can efficiently learn the expert's reward function. On the contrary, IT-IRL does not have the meta-prior $\bar{\theta}$ as a guidance and thus has to search over the whole parameter space, which is extremely difficult to learn the expert's reward function when the data is lacking. Note that MERIT-IRL and Naive MERIT-IRL have better initial performance than IT-IRL and Naive IT-IRL since MERIT-IRL and Naive MERIT-IRL starts at the meta-prior $\bar{\theta}$ while IT-IRL and Naive IT-IRL initializes randomly.

### D.2.2 Hopper

Figure 2c shows that MERIT can achieve similar performance with the expert after $t = 500$ while the other three in-trajectory learning baselines fail to imitate the expert before the ongoing trajectory terminates.

### D.3 Stock Market

We use the real-world data of 30 constitute stocks in Dow Jones Industrial Average from 2021-01-01 to 2022-01-01. The 30 stocks are respectively: 'AXP', 'AMGN', 'AAPL', 'BA', 'CAT', 'CSCO', 'CVX', 'GS', 'HD', 'HON', 'IBM', 'INTC', 'JNJ', 'KO', 'JPM', 'MCD', 'MMM', 'MRK', 'MSFT', 'NKE', 'PG', 'TRV', 'UNH', 'CRM', 'VZ', 'V', 'WBA', 'WMT', 'DIS', 'DOW'.

The state of the stock market MDP is the perception of the stock market, including the open/close price of each stock, the current asset, and some technical indices [48]. The action has the same dimension as the number of stocks where each dimension represents the amount of buying/selling the corresponding stock. The detailed formulation of the MDP can be found in FinRL [48, 58].

The turbulence index is a technical index of stock market and is included as a dimension of the state [48, 58]. The function $p_2$ is defined as the amount of buying the stocks whose turbulence index is larger than the turbulence threshold. Therefore, the more the target investor buys the stocks whose turbulence index is larger than the turbulence threshold, the larger $p_2$ will be and thus the smaller reward the target investor will receive.

**Discussion on the experiment results**. In Figure 2d, MERIT-IRL can achieve the similar cumulative reward with the expert when only the first $60\%$ of the trajectory is observed while IT-IRL can achieve performance close to the expert after $t = 220$. This shows that the meta-regularization can help imitate the expert faster. In contrast, Naive MERIT-IRL and Naive IT-IRL barely improves because the naive reward update method is flawed. Intuitively, the reward update mechanism of these two baselines are myopic as explained in Subsection 4.1. Theoretically, the gradients $g_t''$ of Naive MERIT-IRL and $g_t'''$ of Naive IT-IRL are biased estimate of (4) even if $\pi_t$ approaches $\pi_{\theta_t}$ since (4) includes the trajectory suffix ($i > t$) terms while $g_t''$ and $g_t'''$ only include the trajectory prefix ($i \leq t$) terms.

The last row in Table 1 shows the final results of the algorithms. We can see that MERIT-IRL achieves much better performance than the other in-trajectory learning baselines (i.e., IT-IRL, Naive MERIT-IRL, and Naive IT-IRL). MERIT-IRL achieves comparable performance with Hindsight and the expert. Note that it is not expected that MERIT-IRL outperforms Hindsight since Hindsight has the complete expert trajectory to learn.

## E  Potential negative societal impact

Since MERIT-IRL can infer the reward function of the expert, potential negative societal impact may occur when the learner is malicious. Take the stock market experiment as an example, private information like preferences or habits of the investors may be leaked by using MERIT-IRL. To avoid this situation, the investors needs to take additional strategies such as protecting its investment data from unsecure resources.

# F   Limitations

From the objective (2), we can see that the goal of MERIT-IRL is to align with the expert demonstration, i.e., finding a reward function such that its corresponding policy makes the expert trajectory most likely. An ideal case is that we can also directly quantify the reward learning performance and study the reward identifiability issue. Thus, a future work is to study the reward identifiability issue in the context of in-trajectory IRL.

