# OpenReview forum: "In-Trajectory Inverse Reinforcement Learning: Learn Incrementally Before an Ongoing Trajectory Terminates"
_NeurIPS.cc/2024/Conference — NeurIPS 2024 poster_

### Official Review · Reviewer_L1zi · 2024-06-18

**Soundness:** 2
**Presentation:** 2
**Contribution:** 2
**Rating:** 4
**Confidence:** 4

**Summary:**

In this paper, a novel framework for performing Inverse Reinforcement Learning (IRL) from an ongoing trajectory, i.e., to learn a reward function that induces an optimal policy that best explains the expert's demonstrations sequentially, without waiting to having observed the expert's trajectory entirely. Authors propose an online learning algorithm for solving the problem, and this algorithm has the added feature of meta-regularization. Both a theoretical and empirical analysis of the algorithm are provided to validate the results.

**Strengths:**

- The framework of learning from ongoing trajectories is interesting and finds some applications.

**Weaknesses:**

- There is no clear discussion and intuition of what kind of reward function is extracted by the algorithm and what it can be used for. For instance, various works about identifiability or the feasible set concern what we can do with the extracted rewards, because in most cases we cannot just extract a single reward explaining the demonstrations. However, this work does not analyse this fact.
- The presentation of the contents and contributions is poor.

**Questions:**

- What reward are you extracting among all the rewards that explain the observed demonstrations? What can it be used for?
- Since in RL what matters is the cumulative sum of rewards, how can you be sure that you extract a meaningful reward even though you have not observed the expert's policy for the entire horizon?

**Limitations:**

The authors have adequately addressed the limitations and potential negative societal impact of their work.

---

> ### Author Rebuttal · Authors · 2024-08-06
>
> Thanks for your constructive reviews. We believe that our discussion will lead to a better paper. Before addressing your comments, we would like to clarify the goal of IRL and what role the reward unidentifiability issue plays. IRL aims to learn a reward function that can explain the expert demonstrations. Explaining expert demonstrations means that the optimal policy of the learned reward generates the same trajectories as the expert demonstrations [C1,C2]. Current IRL methods can already learn a single reward function to explain expert demonstrations [C1,C7,C8] but cannot solve the reward unidentifiability issue in general cases. Reward unidentifiability means that even if we can learn a single reward to explain the demonstrations, we cannot guarantee that the learned reward is the expert reward because there are many different rewards that can explain the expert demonstrations. There are some papers studying reward identifiability, however, they all impose additional assumptions. For example, [C3] shows that the reward can be recovered up to a constant if we have demonstrations under different discount factors or different environments. [C4] quantifies the reward distance between the learned reward and the ground truth reward under the assumption that the MDP is linear. [C5,C6] study feasible reward set to identify a set of rewards that can explain the demonstrations, however, they do not solve the identifiability issue because the feasible reward set still includes multiple reward functions that can explain the expert demonstrations. In summary, if we want to learn a single reward to explain expert demonstrations, it is not necessary to solve reward unidentifiability issue nor study reward feasible sets. In fact, the reward unidentifiability issue in general IRL cases is still unsolved to IRL community.
>
> **Weakness 1**: There is no clear ... this fact.
> **Answer**: Thanks for mentioning (1) reward identifiability and feasible reward set, (2) what kind of reward is extracted, and (3) what the learned reward can be used for. We address these three comments one by one.
>
> For the reward identifiability and feasible reward set, please refer to the first paragraph in this response that (i) it is sufficient for IRL to learn a single reward function to explain the demonstrations, (ii) reward identifiability and learning a feasible reward set are sufficient but not necessary for reward learning. Note that reward unidentifiability in general IRL cases is still unsolved to IRL community. In this paper, we empirically compare the learned reward and the expert reward in Appendix D.2, and the results show that the learned reward is close to the expert reward. As mentioned in Appendix F, we would like to explore the reward identifiability in future works.
>
> We now discuss what kind of reward we  extract. Lines 89-94 mention that the kind of reward we aim to learn is the reward with maximum likelihood. Maximum likelihood IRL [C8,C9] is a standard IRL approach to learn a reward function to explain the expert demonstrations. The intuition behind this is that we aim to learn a reward function whose corresponding optimal policy makes the expert demonstrations most likely (line 94), i.e., the corresponding optimal policy can generate trajectories that are same with the expert demonstrations. Therefore, the learned reward with maximum likelihood explains the expert demonstrations [C8,C9].
>
> The learned reward can be used to explain the demonstrations. Moreover, compared to imitation learning that directly learns an imitating policy, the benefit of learning a reward is that the learned reward can be used for counterfactual analysis such as the estimation of optimal policies under different environment dynamics [C8].
>
> **Weakness 2**: The presentation ... is poor.
> **Answer**: Thanks for mentioning the presentation of the contents and contributions. We have a contribution statement in lines 38-53 that clearly summarizes our contributions: (1) we study a novel problem setting of in-trajectory learning, formulate it as a bi-level optimization problem, and propose a novel reward update mechanism to solve the problem; (2) we provide solid theoretical analysis to guarantee that the algorithm achieves sub-linear (local) regret where the input data is not i.i.d.
>
> To improve the presentation of other contents, we will (i) add more explanations about problem (1) to highlight the kind of reward we learn, (ii) add the discussion on related works about reward identifiability and feasible reward set, (iii) add more discussions on the technical assumptions and the theoretical statements.
>
> **Question 1**: What reward ... used for?
> **Answer**: Please refer to the answer to weakness 1.
>
> **Question 2**: Since in RL ... the entire horizon?
> **Weakness**: Please refer to the global response.
>
> [C1] Pieter Abbeel, and Andrew Y. Ng. "Apprenticeship learning via inverse reinforcement learning." ICML, 2004.
>
> [C2] Saurabh Arora, and Prashant Doshi, "A survey of inverse reinforcement learning: Challenges, methods and progress." Artificial Intelligence, 2021.
>
> [C3] Haoyang Cao, et al. "Identifiability in inverse reinforcement learning", NeurIPS, 2021.
>
> [C4] Zihao Li, et al. "Reinforcement learning with human feedback: Learning dynamic choices via pessimism", arXiv preprint arXiv:2305.18438, 2023.
>
> [C5] Alberto Maria Metelli, et al. "Towards theoretical understanding of inverse reinforcement learning", ICML, 2023.
>
> [C6] Filippo Lazzati, et al. "Offline Inverse RL: New Solution Concepts and Provably Efficient Algorithms", arXiv preprint, arXiv:2402.15392, 2024.
>
> [C7] Brian D. Ziebart, et al. "Maximum entropy inverse reinforcement learning." AAAI, 2008.
>
> [C8] Siliang Zeng, et al. "Maximum-likelihood inverse reinforcement learning with finite-time guarantees." NeurIPS, 2022.
>
> [C9] Shicheng Liu and Minghui Zhu. "Learning multi-agent behaviors from distributed and streaming demonstrations." NeurIPS, 2022.

---

> > ### Comment · Reviewer_L1zi · 2024-08-09
> >
> > I thank the authors for the time taken to answer to my questions. However, I am still convinced of the score that I assigned to this paper, because I do not understand why it should be useful to learn a reward function incrementally, given that this increases the usual identifiability problems of IRL. In addition, the problem setting is written in a not clear manner, that does not permit to fully understand the paper.

---

> > > ### Author Response · Authors · 2024-08-10
> > >
> > > Dear reviewer,
> > >
> > > Thanks for mentioning (1) why it should be useful to learn a reward function incrementally given that this increases the reward unidentifiability issue, and (2) the writing of the problem setting. We address these two comments below.
> > >
> > > **For the comment (1)**, we answer from two aspects: (i) the connection between learning a useful reward and reward identifiability, and (ii) why the incrementally learned reward is useful.
> > >
> > > (i) The fundamental usefulness of the learned reward is to explain the expert demonstrations. As mentioned in the first paragraph in our previous response, solving the reward identifiability issue is sufficient but not necessary to learn a reward function to explain the demonstrations. We agree that learning a reward incrementally can make it harder to solve reward unidentifiability issue. However, our algorithm can still learn a reward that explains the expert demonstrations despite the learned reward function may be different from the ground truth. In fact, shown in Table 1 in our paper, the optimal policy corresponding to the learned reward can generate trajectories whose cumulative ground truth reward is close to the expert cumulative ground truth reward, i.e., the learned policy generates trajectories close to the expert trajectories.
> > >
> > > (ii) Moreover, compared to imitation learning that directly learns an imitating policy, our learned reward can be used for conterfactual analysis [C8], e.g., instantly estimating the policies under different environment dynamics. Take the stock market as an example, the reward function captures the investor's risk tolerance [C11]. In our experiment, we learn the investor's risk tolerance incrementally in the stock market of Dow Jones Industrial Average (DJIA). The incrementally learned reward can instantly update the risk tolerance of the target investor. The instantly updated reward has two benefits. First, we can use the latest reward to recommend stocks as soon as possible, which gives us advantages to win this customer before our competitors.  Second, we can use the latest reward to estimate the investment behaviors preferred by the target investor in some other stock markets, e.g., CCXT (cryptocurrency) and  QuantConnect  (US securities) [C12], so that we can not only recommend stocks in the stock market where we collect data, i.e., DJIA, but also recommend stocks in other stock markets, e.g., CCXT and QuantConnect. If we use imitation learning, we can definitely learn an instant imitating policy and use this imitating policy to recommend stocks of DJIA as soon as possible. However, the imitating policy learned by imitation learning cannot be used to recommend stocks in CCXT and QuantConnect because they have different dynamics. In contrast, we can still apply the learned reward to these two new markets to learn the corresponding optimal policies and use the corresponding optimal policies to recommend stocks, because the investor's risk tolerance does not change even if the stock markets are different. We include an additional experiment result below where we use the data collected in DJIA to learn the reward function of the target investor and then apply the learned reward to CCXT and QuantConnect. We compare the cumulative reward between the optimal policy under the learned reward and the expert in these two different markets.
> > > |                                   | CCXT                          | QuantConnect                   |
> > > |:---:|:---:|:--:|
> > > | Learned policy under the learned reward | 562.18 ± 34.02                 | 491.72 ± 18.25                 |
> > > | Expert                            | 571.83 ± 27.16                 | 502.82 ± 14.26                 |
> > >
> > > From the results above, we can see that even if in new stock markets with different dynamics, the learned reward from DJIA can still be used to generate policies in the new stock markets and the generated policies can still achieve similar cumulative ground truth reward with the expert in the new stock markets. This shows that the learned reward successfully captures the investor's risk tolerance.
> > >
> > > **For the comment (2)**, we will add more explanations about the problem formulation (2)-(3) as follows: (i) we will highlight that the method we use to learn a reward function is maximum likelihood IRL (ML-IRL) and provide the intuition of ML-IRL where we aim to learn a reward function that makes the demonstration most likely; (ii) we will show our connection to the standard IRL where our problem formulation (2)-(3) will reduce to the formulation of ML-IRL when the complete trajectory can be observed; (iii) we will discuss the motivation to learn a reward function incrementally.
> > >
> > > [C11] Qizhou Sun, X. Gong, and Y.-W. Si, "Transaction-aware inverse reinforcement learning for trading in stock markets," Applied Intelligence, 2023.
> > >
> > > [C12] Xiaoyang Liu, et al. "FinRL-Meta: Market environments and benchmarks for data-driven financial reinforcement learning." NeurIPS, 2022.

---

> > > > ### Author Response · Authors · 2024-08-12
> > > >
> > > > Dear reviewer,
> > > >
> > > > Thanks again for your review. We have addressed all your comments. Could you please reevlaute our paper based on our response?
> > > >
> > > > Best,
> > > > Authors.

---

> > > > > ### Comment · Reviewer_L1zi · 2024-08-12
> > > > >
> > > > > Dear authors, I had already read your comments. Although I appreciated the additional comments, I have decided to keep the score I assigned.

---

### Official Review · Reviewer_cmJ6 · 2024-07-12

**Soundness:** 4
**Presentation:** 4
**Contribution:** 3
**Rating:** 8
**Confidence:** 4

**Summary:**

The paper "Learn Incrementally from An Ongoing Trajectory: A Provable In-Trajectory Inverse Reinforcement Learning Framework" proposes an innovative approach to inverse reinforcement learning. The authors introduce an online learning algorithm to address the IRL problem with incomplete expert demonstrations. A thorough theoretical analysis guarantees convergence, and a comprehensive experimental section validates the approach.

**Strengths:**

1.Very interesting problem and strong theoretical analysis.


2.The paper introduces a practical algorithm that appears easy to reproduce and yields good results.

**Weaknesses:**

1.In the experiments, I don’t understand why different algorithms have different initial points? When t=0, are you using a same random reward function or the prior-meta reward function to train your policy?

**Questions:**

1. Can the author explain why comparing the suffixes \(\{s_i', a_i'\}\) and \(\{s_i'', a_i''\}\) encourages the reward function to predict the future? Since you are completing the incomplete trajectory using the same policy, can reward function can learn something even if both are generated by the same policy.


2. Just wondering how you complete the incomplete trajectories in MuJoCo using saved states, since the state in MuJoCo only includes partial information about the simulator.

**Limitations:**

The authors address the limitations of this work.

---

> ### Author Rebuttal · Authors · 2024-08-06
>
> Thanks for your insightful reviews. We believe that our discussion will lead a better paper. We address your comments below:
>
> **Weakness 1**: In the experiments, I don’t understand why different algorithms have different initial points? When t=0, are you using a same random reward function or the prior-meta reward function to train your policy?
> **Answer**: Thanks for mentioning the different initial points. There are two different initial points in the experiment. One initial point is the meta-prior reward and the other initial point is random reward. There are four algorithms where the algorithms with meta-regularization (i.e., MERIT-IRL and naive MERIT-IRL) use the same meta-prior reward function as the initialization and the algorithms without meta-regularization (i.e., IT-IRL and naive IT-IRL) use the same random reward function as the initialization. The different initial points show the benefit of meta-regularization where the algorithms with meta-regularization have higher initial cumulative reward than the algorithms without meta-regularization.
>
> **Question 1**: Can the author explain why comparing the suffixes $\\{s_i', a_i'\\}$ and $\\{s_i'', a_i''\\}$ encourages the reward function to predict the future? Since you are completing the incomplete trajectory using the same policy, can reward function learn something even if both are generated by the same policy.
> **Answer**: The suffices $\\{s\_i',a\_i'\\}\_{i\geq t}$ and $\\{s\_i'',a\_i''\\}\_{i\geq t}$ are both generated by the learned policy but they start from different initial state-action pairs where $\\{s\_i'',a\_i''\\}\_{i\geq t}$ starts from the expert state-action pair $(s_t^E,a_t^E)$ and $\\{s\_i',a\_i'\\}\_{i\geq t}$ starts from the learner state-action pair $(s_t',a_t')$. Consider the standard IRL where we have a complete expert trajectory $\\{s\_i^E,a\_i^E\\}\_{i\geq 0}$ to guide the learned trajectory $\\{s\_i',a\_i'\\}\_{i\geq 0}$. The best option to predict the future is to use the expert future (i.e., expert suffix $\\{s\_i^E,a\_i^E\\}\_{i\geq t}$) to guide the learner suffix $\\{s\_i',a\_i'\\}\_{i\geq t}$. However, in our setting, we do not have the future expert trajectory $\\{s\_i^E,a\_i^E\\}\_{i\geq t}$ so that we use the suffix $\\{s\_i'',a\_i''\\}\_{i\geq t}$ as an approximation of the future expert trajectory $\\{s\_i^E,a\_i^E\\}\_{i\geq t}$ to guide the learner suffix $\\{s\_i',a\_i'\\}\_{i\geq t}$. If the approximation suffix $\\{s\_i'',a\_i''\\}\_{i\geq t}$ is a good approximation of the expert suffix $\\{s\_i^E,a\_i^E\\}\_{i\geq t}$, the learned reward can be accurate for the entire horizon.
>
> Now the question left is whether the approximation suffix $\\{s\_i'',a\_i''\\}\_{i\geq t}$ is a good approximation of $\\{s\_i^E,a\_i^E\\}\_{i\geq t}$. The answer is that the approximation error between $\\{s\_i'',a\_i''\\}\_{i\geq t}$ and $\\{s\_i^E,a\_i^E\\}\_{i\geq t}$ diminishes when $t$ increases. This is mathematically justified by the sub-linear regret in Theorems 1 and 2 because the regret will be at least linear if the approximation error is always above a certain constant. The intuition behind this is that when $t$ increases, we observe more portion of the expert trajectory and thus the learned reward $r_{\theta_t}$ improves. A better $r_{\theta_t}$ will lead to a better learned policy $\pi_t$ that generates trajectories closer to the expert trajectory, and thus the new approximation suffix $\\{s\_i'',a\_i''\\}\_{i\geq t+1}$ will be closer to the expert suffix $\\{s\_i^E,a\_i^E\\}\_{i\geq t+1}$, i.e., the approximation error becomes smaller. Therefore, by comparing the learner suffix $\\{s\_i',a\_i'\\}\_{i\geq t+1}$ and the approximation suffix $\\{s\_i'',a\_i''\\}\_{i\geq t+1}$, the learner suffix $\\{s\_i',a\_i'\\}\_{i\geq t+1}$ gets closer to the expert suffix $\\{s\_i^E,a\_i^E\\}\_{i\geq t+1}$.
>
> **Question 2**: Just wondering how you complete the incomplete trajectories in MuJoCo using saved states, since the state in MuJoCo only includes partial information about the simulator.
> **Answer**: In MuJoCo, the learned policy usually maps the observation to a distribution of actions. Given the current observation and current action, the simulator tells us the next observation. We agree that the observation only includes partial information about the simulator. However, the observation is not the physical state of the simulated robot. The physical state in MuJoCo is characterized by two vectors, called "qpos" and "qvel". The vector "qpos" stores all the current positions and orientations of the joints and bodies and the vector "qvel" stores all the current linear and angular velocities of the joints and bodies. The vectors "qpos" and "qvel" include the complete information of the simulated robot. Therefore, we save "qpos" and "qvel" instead of the observation to save a physical state of the simulated robot. In practice, we use the command "env.data.qpos" and "env.data.qvel" to obtain the "qpos" and "qvel" vectors of the current simulated robot and save these vectors. When we want to set the robot's physical state to a saved (qpos,qvel), we use the command "env.set\_state(qpos,qvel)". These commands work for gym-v4.

---

> > ### Comment · Reviewer_cmJ6 · 2024-08-08
> >
> > I thank the authors for responding to my comments and providing these clarifications. I am happy to increase my score, assuming you can incorporate the global clarifications into the text.

---

### Official Review · Reviewer_MWk1 · 2024-07-12

**Soundness:** 3
**Presentation:** 3
**Contribution:** 3
**Rating:** 7
**Confidence:** 4

**Summary:**

The authors consider a new problem setting, called in-trajectory IRL, where a reward function and a corresponding policy need to be learned from an ongoing trajectory. The authors propose a novel reward update mechanism specially designed for this scenario and incorporate a meta-regularization strategy to embed prior knowledge and avoid overfitting. Theoretical analysis is provided for the convergence of the algorithm towards stationary points (and global under linear reward parametrization). MuJoCo benchmark and a real-world stock market example are used for experiments.

**Strengths:**

1.	The topic is novel to IRL community. The new problem setting of learning the reward function and policy on the fly aligns with many real-world scenarios. In this new situation, standard IRL algorithms are not suitable anymore.
2.	The special reward update design of predicting the future trajectory is reasonable and interesting for this ongoing trajectory setting.
3.	The meta-regularization is novel to IRL, which helps to better find a reward from limited data.
4.	The theoretical analysis is solid and tackles the issue of non i.i.d. input data. The theoretical distinction from literature is well clarified. The experiment includes real-world stock market data.

**Weaknesses:**

1.	Assumption 1 assumes that the parameterized reward is smooth, which can be a strong assumption, especially for neural networks with non-smooth activation functions such as ReLU.
2.	The meta-regularization part seems to be important to the performance. However, the meta-learning needs to pre-collect data for many related tasks, which can be difficult in real world.

**Questions:**

How the authors collect data for the meta-regularization in the experiment?

Other questions please refer to the Weaknesses.

**Limitations:**

The authors have discussed the limitations in Appendix F.

---

> ### Author Rebuttal · Authors · 2024-08-06
>
> Thanks for your constructive review. We believe that our discussion will lead to a stronger paper. We address your comments below:
>
> **Weakness 1**: Assumption 1 assumes that the parameterized reward is smooth, which can be a strong assumption, especially for neural networks with non-smooth activation functions such as ReLU.
> **Answer**: Thanks for mentioning Assumption 1. We agree that this assumption could be strong when the neural networks use ReLU activation. However, the smoothness assumption can be satisfied by using some smooth activation functions such as tanh and using linearly parameterized models. In our experiment, we use tanh as the activation function. Moreover, as mentioned in the paragraph under Assumption 1 (i.e., line 255), this assumption is widely adopted in RL [A1] and IRL [A2,A3]. It is interesting to relax this assumption, and we will explore it in the future.
>
> **Weakness 2**: The meta-regularization part seems to be important to the performance. However, the meta-learning needs to pre-collect data for many related tasks, which can be difficult in real world.
> **Answer**: Meta-learning requires to collect data for related tasks before training, however, data collection is not difficult in many real-world problems. In fact, meta-learning has been applied to many real-world problems, such as velocity modelling for self-driving where the data of related tasks (i.e., velocity trajectories in different roads) can be obtained from public data sets [A4], and object grasping of robotic arms where the data of related tasks (i.e., demonstrations of grasping objects from different locations) is generated by solving RL problems to grasp the objects [A5].
> In Section 5.2, a real-world stock market example is used to evaluate our algorithm. Data collection is not difficult for this application. In particular, the demonstration data of related tasks is the investment trajectories of other investors with different preferences of taking risks. We can collect such information from public records [A6]. Moreover, our method can also be applied to [A4,A5]. Take [A5] as an example, we can first use their data sets to learn the meta-prior. Then, given a new demonstration trajectory of grasping an object from a new location, we can sequantially reveal this trajectory to the learner in an ongoing fashion.
>
> **Question 1**: How the authors collect data for the meta-regularization in the experiment?
> **Answer**: For the MuJoCo experiment, the different training tasks have different target velocity from 0 to 3 (line 328) and the  reward function is $-|v-v_{\text{target}}|$ where $v$ is the current velocity and $v_{\text{target}}$ is the target velocity (line 326). Therefore, for each training task, we randomly sample the target velocity from 0 to 3, design the reward function correspondingly, and use the reward function to train an RL policy to generate demonstration trajectories. For the stock market environment, different tasks have different preferences of taking risks, which is captured by different turbulence threshold (line 362). The reward function is $p_1-p_2$ where $p_1$ quantifies the profit and $p_2$ quantifies the amount of trading stocks that are above the turbulence threshold. Therefore, for each training task, we randomly sample turbulence threshold from 30 to 60 (line 365), design the corresponding $p_2$, and train an RL agent under the current $p_1-p_2$ to generate investment demonstrations. For the active shooting experiment in Appendix D.5, different training tasks have different goal locations. For each training task, we randomly sample an $1\times 1$ goal area from the state space and train an RL algorithm to reach the goal to generate demonstrations.
>
> [A1] Kaiqing Zhang ,Alec Koppel, Hao Zhu, and Tamer Basar. "Global convergence of policy gradient methods to (almost) locally optimal policies." SIAM Journal on Control and Optimization, 2020.
>
> [A2] Siliang Zeng, Chenliang Li, Alfredo Garcia, and Mingyi Hong, “Maximum-likelihood inverse reinforcement learning with finite-time guarantees,” in Advances in Neural Information Processing Systems, 2022.
>
> [A3] Ziwei Guan, Tengyu Xu, and Yingbin Liang. "When will generative adversarial imitation learning algorithms attain global convergence." In International Conference on Artificial Intelligence and Statistics, 2021.
>
> [A4] Bo Yu, Xiangyu Feng, You Kong, Yuren Chen, Zeyang Cheng, and Shan Bao. "Using meta-learning to establish a highly transferable driving speed prediction model from the visual road environment." Engineering Applications of Artificial Intelligence, 2024.
>
> [A5] Chelsea Finn, Tianhe Yu, Tianhao Zhang, Pieter Abbeel, and Sergey Levine. "One-shot visual imitation learning via meta-learning." In Conference on robot learning, 2017.
>
> [A6] Xiaoyang Liu et al. "FinRL-Meta: Market environments and benchmarks for data-driven financial reinforcement learning." Advances in Neural Information Processing Systems, 2022.

---

> > ### Comment · Reviewer_MWk1 · 2024-08-08
> >
> > Thanks for the detailed response. The authors have done a great job answering my questions. I have no concerns nor questions about the assumption and the meta-regularization part now. I thus increase my rating. The meta-regularization is novel to IRL and the in-trajectory learning is an intereting new problem setting. I recommend acceptance of this paper.

---

### Author Rebuttal · Authors · 2024-08-06

**How can we extract a meaningful reward even though we have not observed the expert’s policy for
the entire horizon?**

The reason is that our reward update approximates the entire expert horizon and learns from the approximate entire expert trajectory (detailed in lines 188-219). We include a figure (Figure 4) in the uploaded one-page PDF to visualize this process.

Consider the standard IRL (visualized in Figure 4a) where we can compare the entire expert trajectory $\\{s\_i^E, a\_i^E\\}\_{i \geq 0}$ (the green one in Figure 4) and entire learner trajectory $\\{s\_i', a\_i'\\}\_{i \geq 0}$ (the red one in Figure 4). In this standard IRL case, the learned reward considers the entire horizon. In our case, at time $t$, we only observe expert trajectory prefix $\\{s\_i^E, a\_i^E\\}\_{0 \leq i \leq t}$. To ensure that the learned reward still considers the entire horizon, we use the learned policy to complete the expert trajectory by generating the suffix $\\{s\_i'', a\_i''\\}\_{i \geq t}$ that starts from $(s\_t^E, a\_t^E)$ (the yellow one in Figure 4b). Now we have a combined entire trajectory whose prefix is $\\{s\_i^E, a\_i^E\\}\_{0 \leq i \leq t}$ and suffix is $\\{s\_i'', a\_i''\\}\_{i \geq t}$. This combined trajectory approximates the entire expert trajectory $\\{s\_i^E, a\_i^E\\}\_{i \geq 0}$ where the suffix $\\{s\_i'', a\_i''\\}\_{i \geq t}$ approximates the expert suffix $\\{s\_i^E, a\_i^E\\}\_{i \geq t}$. We compare this combined entire trajectory with the entire learner trajectory $\\{s\_i', a\_i'\\}\_{i \geq 0}$ to enable the learned reward to consider the entire horizon. Moreover, this comparison encourages the learner suffix $\\{s\_i', a\_i'\\}\_{i \geq t}$ to approach the approximation suffix $\\{s\_i'', a\_i''\\}\_{i \geq t}$. If the approximation suffix $\\{s\_i'', a\_i''\\}\_{i \geq t}$ is a good approximation of the expert suffix $\\{s\_i^E, a\_i^E\\}\_{i \geq t}$, the learned reward can be accurate for the entire horizon.

Now the question left is whether the approximation suffix $\\{s\_i'', a\_i''\\}\_{i \geq t}$ is a good approximation of $\\{s_i^E, a_i^E\\}\_{i \geq t}$. The answer is that the approximation error between $\\{s\_i'',a\_i''\\}\_{i\geq t}$ and $\\{s\_i^E,a\_i^E\\}\_{i\geq t}$ diminishes when $t$ increases. This is mathematically justified by the sub-linear regret in Theorems 1 and 2 because the regret will be at least linear if the approximation error does not diminish or is always above a certain constant. The intuition behind this is that when $t$ increases, we observe more portion of the expert trajectory and thus the learned reward $r_{\theta_t}$ improves. A better $r_{\theta_t}$ will lead to a better learned policy $\pi_t$ that generates trajectories closer to the expert trajectory, and thus the new approximation suffix $\\{s\_i'',a\_i''\\}\_{i\geq t+1}$ will be closer to the expert suffix $\\{s\_i^E,a\_i^E\\}\_{i\geq t+1}$, i.e., the approximation error becomes smaller (visualized in Figure 4c). Therefore, by comparing the learner suffix $\\{s\_i',a\_i'\\}\_{i\geq t+1}$ and the approximation suffix $\\{s\_i'',a\_i''\\}\_{i\geq t+1}$, the learner suffix $\\{s\_i',a\_i'\\}\_{i\geq t+1}$ gets closer to the expert suffix $\\{s\_i^E,a\_i^E\\}\_{i\geq t+1}$.

---

### Decision · Program_Chairs · 2024-09-25

**Decision:**

Accept (poster)

**Comment:**

This paper considers in-trajectory IRL, where a reward function and a corresponding policy need to be learned from an ongoing trajectory. The authors introduce an online learning algorithm to address this IRL problem with incomplete expert demonstrations. The paper includes a thorough theoretical analysis of the proposed solution, as well as a convincing empirical analysis on simulated data and some toy problems. A reviewer raised the issue regarding the identifiability of reward functions, but other reviewers think that this is a minor issue in the context of the other contributions of this work.